# Minimax Optimal Estimation of Approximate Differential Privacy on Neighboring Databases

**Xiyang Liu**    **Sewoong Oh**
Allen School of Computer Science and Engineering,
University of Washington
{xiyangl, sewoong}@cs.washington.edu

## Abstract

Differential privacy has become a widely accepted notion of privacy, leading to the introduction and deployment of numerous privatization mechanisms. However, ensuring the privacy guarantee is an error-prone process, both in designing mechanisms and in implementing those mechanisms. Both types of errors will be greatly reduced, if we have a data-driven approach to verify privacy guarantees, from a black-box access to a mechanism. We pose it as a property estimation problem, and study the fundamental trade-offs involved in the accuracy in estimated privacy guarantees and the number of samples required. We introduce a novel estimator that uses polynomial approximation of a carefully chosen degree to optimally trade-off bias and variance. With $n$ samples, we show that this estimator achieves performance of a straightforward plug-in estimator with $n \ln n$ samples, a phenomenon known as sample size amplification. The minimax optimality of the estimator is proved by comparing it to a matching fundamental lower bound.

## 1   Introduction

Differential privacy is gaining popularity as an agreed upon measure of privacy leakage, widely used by the government to publish Census statistics [1], Google to aggregate user's choices in web-browser features [2, 3], Apple to aggregate mobile user data [4], and smart meters in telemetry [5]. As increasing number of privatization mechanisms are introduced and deployed in the wild, it is critical to have countermeasures to check the fidelity of those mechanisms. Such techniques will allow us to hold accountable the deployment of privatization mechanisms if the claimed privacy guarantees are not met, and help us find and fix bugs in implementations of those mechanisms.

A user-friendly tool for checking privacy guarantees is necessary for several reasons. Writing a program for a privatization mechanism is error-prone, as it involves complex probabilistic computations. Even with customized languages for differential privacy, checking the end-to-end privacy guarantee of an implementation remains challenging [6, 7]. Furthermore, even when the implementation is error-free, there have been several cases where the mechanism designers have made errors in calculating the privacy guarantees, and falsely reported higher level of privacy [8, 9]. This is evidence of an alarming issue that analytically checking the proof of a privacy guarantee is a challenging process even for an expert. An automated and data-driven algorithm for checking privacy guarantees will significantly reduce such errors in the implementation and the design. On other cases, we are given very limited information about how the mechanism works, like Apple's white paper [4]. The users are left to trust the claimed privacy guarantees.

To address these issues, we propose a data-driven approach to estimate how much privacy is guaranteed, from a black-box access to a purportedly private mechanism. Our approach is based on an optimal polynomial approximation that gracefully trades off bias and variance. We study the

fundamental limit of how many samples are necessary to achieve a desired level of accuracy in the estimation, and show that the proposed approach achieves this fundamental bound.

**Problem formulation.** Differential privacy (DP) introduced in [10] is a formal mathematical notion of privacy that is widespread, due to several key advantages. It gives one of the strongest guarantees, allows for precise mathematical analyses, and is intuitive to explain even to non-technical end-users. When accessing a database through a query, we say the query output is private if the output did not reveal whether a particular person's entry is in the database or not. Formally, we say two databases are *neighboring* if they only differ in one entry (one row in a table, for example). Let $P_{Q,D}$ denote the distribution of the randomized output to a query $Q$ on a database $D$. We consider discrete valued mechanisms taking one of $S$ values, i.e. the response to a query is in $[S] = \{1, \ldots, S\}$ for some integer $S$. We say a mechanism guarantees $(\varepsilon, \delta)$-DP, [10], if the following holds

$$P_{Q,D}(E) \leq e^{\varepsilon} P_{Q,D'}(E) + \delta , \tag{1}$$

for some $\varepsilon \geq 0$, $\delta \in [0, 1]$, and all subset $E \subseteq [S]$ and for all neighboring databases $D$ and $D'$. When $\delta = 0$, $(\varepsilon, 0)$-DP is referred to as (pure) differential privacy, and the general case of $\delta \geq 0$ is referred to as *approximate* differential privacy. For pure DP, the above condition can be relaxed as

$$P_{Q,D}(x) \leq e^{\varepsilon} P_{Q,D'}(x) , \tag{2}$$

for all *output symbol* $x \in [S]$, and for all neighboring databases $D$ and $D'$. This condition can now be checked, one symbol $x$ at a time from $[S]$, without having to enumerate all subsets $E \subseteq [S]$. This naturally leads to the following algorithm.

For a query $Q$ and two neighboring databases $D$ and $D'$ of interest, we need to verify the condition in Eq. (2). As we only have a black-box access to the mechanism, we collect $n$ responses from the mechanism on the two databases. We check the condition on the empirical distribution of those collected samples, for each $x \in [S]$. If it is violated for any $x$, we assert the mechanism to be not $(\varepsilon, 0)$-DP and present $x$ as an evidence. Focusing only on pure DP, [11] proposed an approach similar to this, where they also give guidelines for choosing the databases $D$ and $D'$ to test. However, their approach is only evaluated empirically, no statistical analysis is provided, and a more general case of approximate DP is left as an open question, as the condition in Eq. (1) cannot be decoupled like Eq. (2) when $\delta > 0$.

We propose an alternative approach from first principles to check the general approximate DP guarantees, and prove its minimax optimality. Given two probability measures $P = [p_1, \ldots, p_S]$ and $Q = [q_1, \ldots, q_S]$ over $[S] = \{1, \ldots, S\}$, we define the following *approximate DP divergence* with respect to $\varepsilon$ as

$$d_{\varepsilon}(P\|Q) \triangleq \sum_{i=1}^{S} [p_i - e^{\varepsilon} q_i]^+ = \mathbb{E}_{x \sim P}\left[ \left[ 1 - e^{\varepsilon} \frac{q_x}{p_x} \right]^+ \right] . \tag{3}$$

where $[x]^+ = \max\{x, 0\}$. The last representation indicates that this metric falls under a broader class of metrics known as $f$-divergences, with a special choice of $f(x) = [1 - e^{\varepsilon} x]^+$. From the definition of DP, it follows that a mechanism is $(\varepsilon, \delta)$-DP if and only if $d_{\varepsilon}(P_{Q,D}\|P_{Q,D'}) \leq \delta$ for all neighboring databases $D$ and $D'$. We propose estimating this divergence $d_{\varepsilon}(P_{Q,D}\|P_{Q,D'})$ from samples, and comparing it to the target $\delta$. This only requires number of operations scaling as $S \ln n$ where $n$ is the sample size.

In this paper, we suppose there is a specific query $Q$ of interest, and two neighboring databases $D$ and $D'$ have been already selected either by a statistician who has some side information on the structure of the mechanism or by some algorithm, such as those from [11, 12, 13]. Without exploiting the structure (such as symmetry, exchangeability, or invariance to the entries of the database conditioned on the true output of the query), one cannot avoid having to check all possible combinations of neighboring databases. As a remedy, [12] proposes checking randomly selected databases. This in turn ensures a relaxed notion of privacy known as random differential privacy. Similarly, [13] proposed checking the *typical* databases, assuming there we have access to a prior distribution over the databases. Our framework can be seamlessly incorporated with such higher-level routines to select databases.

**Contributions.** We study the problem of estimating the approximate differential privacy guaranteed by a mechanism, from a black-box access where we can sample from the mechanism output given a

query $\mathcal{Q}$, a database $\mathcal{D}$, and a target $(\varepsilon, \delta)$. We first show that a straightforward plug-in estimator of $d_\varepsilon(P\|Q)$ achieves mean squared error scaling as $(e^\varepsilon S)/n$, where $S$ is the size of the alphabet and $n$ is the number of samples used (Section 2.1.1).

In the regime where we fix $S$ and increase the sample size, this achieves the parametric rate of $1/n$, and cannot be improved upon. However, in many cases of practical interest where $S$ is comparable to $n$, we show that this can be improved upon with a more sophisticated estimator. To this end, we introduce a novel estimator of $d_\varepsilon(P\|Q)$. The main idea is to identify the regimes of non-smoothness in $[p_i - e^\varepsilon q_i]^+$ where the plug-in estimator has a large bias. We replace it by the uniformly best polynomial approximation of the non-smooth regime of the function, and estimate those polynomial from samples. By selecting appropriate degree of the polynomial, we can optimally trade off the bias and variance. We provide an upper bound on the error scaling as $(e^\varepsilon S)/(n \ln n)$, when $S$ and $n$ are comparable. We prove that this is the best one can hope for, by providing a matching lower bound.

We first show this for the case when we know $P$ and sample from $Q$ in Section 2.1, to explain the main technical insights while maintaining simple exposition. Then, we consider the practical scenario where both $P$ and $Q$ are accessed via samples, and provide an minimax optimal estimator in Section 2.2. This phenomenon is referred to as *effective sample size amplification*; one can achieve with $n$ samples a desired error rate, that would require $n \ln n$ samples for a plug-in estimator. We present numerical experiments supporting our theoretical predictions in Section 3.

**Related work.** A formal investigation into verifying DP guarantees of a given mechanism was addressed in [13]. DP condition is translated into a certain Lipschitz condition on $P_{\mathcal{Q},\mathcal{D}}$ over the databases $\mathcal{D}$, and a Lipschitz tester is proposed to check the conditions. However, this approach is not data driven, as it requires the knowledge of the distribution $P_{\mathcal{Q},\mathcal{D}}$ and no sampling of the mechanism outputs is involved. [12] analyzes tradeoffs involved in testing DP guarantees. It is shown that one cannot get accurate testing without sacrificing the privacy of the databases used in the testing. Hence, when testing DP guarantees, one should not use databases that contain sensitive data. We compare some of the techniques involved in Section 2.1.1.

Our techniques are inspired by a long line of research in property estimation of a distribution from samples. There has been significant recent advances for high-dimensional estimation problems, starting from entropy estimation in [14, 15, 16]. The general recipe is to identify the regime where the property to be estimated is not smooth, and use functional approximation to estimate a smoothed version of the property. This has been widely successful in support recovery [17], density estimation with $\ell_1$ loss [18], and estimating Renyi entropy [19]. More recently, this technique has been applied to estimate certain divergences between two unknown distributions, for Kullback-Leibler divergence [20], total variation distance [21], and identity testing [22]. With carefully designed estimators, these approximation-based approaches can achieve improvement over typical parametric rate of $1/n$ error rate, sometimes referred to as *effective sample size amplification*.

**Notations.** We let the alphabet of a discrete distribution be $[S] = \{1, \ldots, S\}$ for some positive integer $S$ denoting the size of the alphabet. We let $\mathcal{M}_S$ denote the set of probability distributions over $[S]$. We use $f(n) \gtrsim g(n)$ to denote that $\sup_n f(n)/g(n) \geq C$ for some constant $C$, and $f(n) \lesssim g(n)$ is analogously defined. $f(n) \asymp g(n)$ denotes that $f(n) \gtrsim g(n)$ and $f(n) \lesssim g(n)$.

## 2 Estimating differential privacy guarantees from samples

We want to estimate $d_\varepsilon(P\|Q)$ from a blackbox access to the mechanism outputs accessing two databases, i.e. $P = P_{\mathcal{Q},\mathcal{D}}$ and $Q = P_{\mathcal{Q},\mathcal{D}'}$. We first consider a simpler case, where $P = [p_1, \ldots, p_S]$ is known and we observe samples from an unknown distribution $Q = [q_1, \ldots, q_S]$ in Section 2.1. We cover this simpler case first to demonstrate the main ideas on the algorithm design and analysis technique while maintaining the exposition simple. This paves the way for our main algorithmic and theoretical results in Section 2.2, where we only have access to samples from both $P$ and $Q$.

### 2.1 Estimating $d_\varepsilon(P\|Q)$ with known P

For a given budget $n$, representing an upper bound on the expected number of samples we can collect, we propose sampling a random number $N$ of samples from Poisson distribution with mean $n$, i.e. $N \sim \text{Poi}(n)$. Then, each sample $X_j \in [S]$ is drawn from $Q$ for $j \in \{1, \ldots, N\}$, and we let

$Q_n = [\hat{q}_1, \ldots, \hat{q}_S]$ denote the resulting histogram divided by $n$, such that $\hat{q}_i \triangleq |\{j \in [N] : X_i = j\}|/n$. Note that $Q_n$ is not the standard *empirical distribution*, as $\sum_i \hat{q}_i \neq 1$ with high probability. However, in this paper we refer to $Q_n$ as empirical distribution of the samples. The empirical distribution would have been divided by $N$ instead of $n$. Instead, $Q_n$ is the maximum likelihood estimate of the true distribution $Q$. This Poisson sampling, together with the MLE construction of $Q_n$, ensures independence among $\{\hat{q}_i\}_{i=1}^S$, making the analysis simpler.

### 2.1.1 Performance of the plug-in estimator

The following result shows that it is necessary and sufficient to have $n \approx e^\varepsilon S$ samples to achieve an arbitrary desired error rate, if we use this plug-in estimator $d_\varepsilon(P\|Q_n)$, under the worst-case $P$ and $Q$. Some assumption on $(P, Q)$ is inevitable as it is trivial to achieve zero error for any sample size, for example if $P$ and $Q$ have disjoint supports. Both $d_\varepsilon(P\|Q)$ and $d_\varepsilon(P\|Q_n)$ are 1 with probability one. We provide a proof in Appendix C.2. The bound in Eq. (4) also holds for $d_\varepsilon(P_n\|Q)$.

**Theorem 1.** *For any $\varepsilon \geq 0$ and support size $S \in \mathbb{Z}^+$, if $n \geq e^\varepsilon S$, then the plug-in estimator satisfies*

$$\sup_{P,Q \in \mathcal{M}_S} \mathbb{E}_Q\big[\, |d_\varepsilon(P\|Q_n) - d_\varepsilon(P\|Q)|^2 \,\big] \;\asymp\; \frac{e^\varepsilon S}{n} \;. \tag{4}$$

A similar analysis was done in [12], which gives an upper bound scaling as $e^{2\varepsilon}S/n$. We tighten the analysis by a factor of $e^\varepsilon$, and provide a matching lower bound.

### 2.1.2 Achieving optimal sample complexity with a polynomial approximation

We construct a minimax optimal estimator using techniques first introduced in [16, 15] and adopted in several property estimation problems including [18, 20, 19, 21, 22, 17].

---

**Algorithm 1** Differential Privacy (DP) estimator with known $P$

---

**Input:** target privacy $\varepsilon \in \mathbb{R}^+$, query $\mathcal{Q}$, neighboring databases $(\mathcal{D}, \mathcal{D}')$, pmf of $P_{\mathcal{Q},\mathcal{D}}$
  samples from $P_{\mathcal{Q},\mathcal{D}'}$, degree $K \in \mathbb{Z}^+$, constants $c_1, c_2 \in \mathbb{R}^+$, expected sample size $2n$

**Output:** estimate $\widehat{d}_{\varepsilon,K,c_1,c_2}(P\|Q_n)$ of $d_\varepsilon(P_{\mathcal{Q},\mathcal{D}}\|P_{\mathcal{Q},\mathcal{D}'})$

  $P \leftarrow P_{\mathcal{Q},\mathcal{D}}$
  Draw two independent sample sizes: $N_1 \leftarrow \mathrm{Poi}\,(n)$ and $N_2 \leftarrow \mathrm{Poi}\,(n)$
  Sample from $P_{\mathcal{Q},\mathcal{D}'}$: $\{X_{i,1}\}_{i=1}^{N_1} \in [S]^{N_1}$ and $\{X_{i,2}\}_{i=1}^{N_2} \in [S]^{N_2}$
  $\hat{q}_{i,j} \leftarrow \frac{|\{\ell \in [N_j] : X_{\ell,j} = i\}|}{n}$ for all $i \in [S]$ and $j \in \{1, 2\}$
  $Q_{n,1} \leftarrow [\hat{q}_{1,1}, \ldots, \hat{q}_{S,1}]$ and $Q_{n,2} \leftarrow [\hat{q}_{1,2}, \ldots, \hat{q}_{S,2}]$
  **for** $i = 1$ **to** $S$ **do**

$$\delta_i \leftarrow \begin{cases} 0 & , \text{ if } \hat{q}_{i,1} > U(p_i; c_1, c_2) \\ \tilde{D}_K(\hat{q}_{i,2}; p_i) & , \text{ if } \hat{q}_{i,1} \in U(p_i; c_1, c_2) \\ [p_i - e^\varepsilon \hat{q}_{i,2}]^+ & , \text{ if } \hat{q}_{i,1} < U(p_i; c_1, c_2) \end{cases} \quad \text{(defined in Appendix A)}$$

  **end for**
  $\widehat{d}_{\varepsilon,K,c_1,c_2}(P\|Q_n) \leftarrow 0 \vee (1 \wedge \sum_{i=1}^S \delta_i)$

---

To simplify the analysis, we split the samples randomly into two partitions, each having an independent and identical distribution of $\mathrm{Poi}\,(n)$ samples from the multinomial distribution $Q$. We let $Q_{n,1} = [\hat{q}_{1,1}, \ldots, \hat{q}_{S,1}]$ denote the count of the first set of $N_1 \sim \mathrm{Poi}\,(n)$ samples (normalized by $n$), and $Q_{n,2} = [\hat{q}_{1,1}, \ldots, \hat{q}_{S,1}]$ the second set of $N_2 \sim \mathrm{Poi}\,(n)$ samples. See Algorithm 1 for a formal definition. Note that for the analysis we are collecting $2n$ samples in total on average. In all the experiments, however, we apply our estimator without partitioning the samples. A major challenge in achieving the minimax optimality is in handling the non-smoothness of the function $f(\hat{q}_i; p_i) \triangleq [p_i - e^\varepsilon \hat{q}_i]^+$ at $p_i \simeq e^\varepsilon \hat{q}_i$. We use one set of samples to identify whether an outcome $i \in [S]$ is in the smooth regime ($\hat{q}_{i,1} \notin U(p_i; c_1, c_2)$) or not ($\hat{q}_{i,1} \in U(p_i; c_1, c_2)$), with an appropriately defined set function:

$$U(p; c_1, c_2) \triangleq \begin{cases} [0, \frac{(c_1+c_2)\ln n}{n}] & , \text{ if } p \leq \frac{c_1 e^\varepsilon \ln n}{n} \;, \\ \left[ e^{-\varepsilon}p - \sqrt{\frac{c_2 e^{-\varepsilon}p \ln n}{n}}, e^{-\varepsilon}p + \sqrt{\frac{c_2 e^{-\varepsilon}p \ln n}{n}} \right] & , \text{ otherwise,} \end{cases} \tag{5}$$

for $c_1 \geq c_2 > 0$ and $p \in [0,1]$. The scaling of the interval is chosen carefully such that $(a)$ it is large enough for the probability of making a mistake on the which regime $(p_i, q_i)$ falls into to vanishes (Lemma 13); and $(b)$ it is small enough for the variance of the polynomial approximation in the non-smooth regime to match that of the other regimes (Lemma 14). In the smooth regime, we use the plug-in estimator. In the non-smooth regime, we can improve the estimation error by using the best polynomial approximation of $f(x;p) = [p - e^\varepsilon x]^+$, which has a smaller bias:

$$D_K(x;p) \triangleq \underset{P \in \mathrm{poly_K}}{\arg\min} \ \underset{\tilde{x} \in U(p;c_1,c_1)}{\max} \left| [p - e^\varepsilon \tilde{x}]^+ - P(\tilde{x}) \right|, \tag{6}$$

where $\mathrm{poly_K}$ is the set of polynomial functions of degree at most $K$, and we approximate $f(x;p)$ in an interval $U(p;c_1,c_1) \supset U(p;c_1,c_2)$ for any $c_1 > c_2$. Having this slack of $c_1 > c_2$ in the approximation allows us to guarantee the approximation quality, even if the actual $q$ is not exactly in the non-smooth regime $U(p;c_1,c_2)$. Once we have the polynomial approximation, we estimate this polynomial function $D_K(x;p)$ from samples, using the *uniformly minimum variance unbiased estimator (MVUE)*.

There are several advantages that makes this two-step process attractive. As we use an unbiased estimate of the polynomial, the *bias* is exactly the polynomial approximation error of $D_K(x;p)$, which scales as $(1/K)\sqrt{(p_i \ln n)/n}$. Larger degree $K$ reduces the approximation error, and larger $n$ reduces the support of the domain we apply the approximation to in $U(p;c_1,c_1)$ (Lemma 14). The *variance* is due to the sample estimation of the polynomial $D_K(x;p)$, which scales as $(B^K p_i \ln n)/n$ for some universal constant $B$ (Lemma 14). Larger degree $K$ increases the variance. We prescribe choosing $K = c_3 \ln n$ for appropriate constant $c_3$ to optimize the bias-variance tradeoff in Algorithm 1. The methods of constructing the polynomial approximation $D_K(x;p)$ and corresponding unbiased estimator $\tilde{D}_K(x;p)$ are described in details at Appendix A.

**Theorem 2.** *Suppose $\ln n \leq C'(\ln S - \varepsilon)$ for some constants $C'$, then there exist constants $c_1, c_2$ and $c_3$ that only depends on $C'$ and $\varepsilon$ such that*

$$\sup_{P,Q \in \mathcal{M}_S} \mathbb{E}_Q \left[ \left| \widehat{d}_{\varepsilon,K,c_1,c_2}(P\|Q_n) - d_\varepsilon(P\|Q) \right|^2 \right] \ \lesssim \ \frac{e^\varepsilon S}{n \ln n} \ . \tag{7}$$

*for $K = c_3 \ln n$ and where $\widehat{d}_{\varepsilon,K,c_1,c_2}$ is defined in Algorithm 1.*

We provide a proof in Appendix C.3, and a matching lower bound in Theorem 3. Note that the plug-in estimator in Theorem 1 achieves the parametric rate of $1/n$. In the low-dimensional regime, where we fix $S$ and grow $n$, this cannot be improved upon. To go beyond the parametric rate, we need to consider a high-dimensional regime, where $S$ grows with $n$. Hence, a condition similar to $\ln n \leq C' \ln S$ is necessary, although it might be possible to further relax it.

### 2.1.3 Matching minimax lower bound

In the high-dimensional regime, where $S$ grows with $n$ sufficiently fast, we can get a tighter lower bound then Theorem 1, that matches the upper bound in Theorem 2. Again, supremum over $Q$ is necessary as there exists $(P,Q)$ where it is trivial to achieve zero error, for any sample size (see Section 2.1.1 for an example). For any given $P$ we provide a minimax lower bound in the following. A proof is provided in Appendix C.4.

**Theorem 3.** *Suppose $S \geq 2$ and there exists constants $c, C_1, C_2 > 0$ such that $C_1 \ln S \leq \ln n \leq C_2 \ln S$ and $n \geq c(e^\varepsilon S)/\ln S$, then*

$$\sup_{P \in \mathcal{M}_S} \ \inf_{\widehat{d}_\varepsilon(P\|Q_n)} \ \sup_{Q \in \mathcal{M}_S} \mathbb{E}_Q \left[ \left| \widehat{d}_\varepsilon(P\|Q_n) - d_\varepsilon(P\|Q) \right|^2 \right] \ \gtrsim \ \frac{e^\varepsilon S}{n \ln n} \ . \tag{8}$$

*where the infimum is taken over all possible estimators.*

### 2.2 Estimating $d_\varepsilon(P\|Q)$ from samples

We now consider the general case where $P = P_{\mathcal{Q},\mathcal{D}}$ and $Q = P_{\mathcal{Q},\mathcal{D}'}$ are both unknown, and we access them through samples. We propose sampling a random number of samples $N_1 \sim \mathrm{Poi}(n)$ and $N_2 \sim \mathrm{Poi}(n)$ from each distribution, respectively. Define the empirical distributions $P_n =$

$[\hat{p}_1, \ldots, \hat{p}_S]$ and $Q_n = [\hat{q}_1, \ldots, \hat{q}_S]$ as in the previous section. From the proof of Theorem 1, we get the same sample complexity for the plug-in estimator: If $n \geq e^\varepsilon S$ and $S \geq 2$, we have

$$\sup_{P,Q \in \mathcal{M}_S} \mathbb{E}_Q \left[ \left| d_\varepsilon(P_n \| Q_n) - d_\varepsilon(P \| Q) \right|^2 \right] \; \asymp \; \frac{e^\varepsilon S}{n} . \tag{9}$$

Using the same two-step process, we construct an estimator that improves upon this parametric rate of plug-in estimator.

### 2.2.1 Estimator for $d_\varepsilon(P \| Q)$

We present an estimator using similar techniques as in Algorithm 1, but there are several challenges in moving to a multivariate case. The multivariate function $f(x, y) = [x - e^\varepsilon y]^+$ is non-smooth in a region $x = e^\varepsilon y$. We first define a two-dimensional non-smooth set $U(c_1, c_2) \subset [0,1] \times [0, e^\varepsilon]$ as

$$U(c_1, c_2) \;=\; \left\{ (p, e^\varepsilon q) : |p - e^\varepsilon q| \leq \sqrt{\frac{(c_1 + c_2) \ln n}{n}} (\sqrt{p} + \sqrt{e^\varepsilon q}), \; p \in [0,1], \; q \in [0,1] \right\} , \tag{10}$$

where $0 < c_2 < c_1$. As before, the plug-in estimator is good enough in the smooth regime, i.e. $(p, e^\varepsilon q) \notin U(c_1, c_2)$.

We construct a polynomial approximation of this function with order $K$, in this non-smooth regime. We will set $K = c_3 \ln n$ again to achieve the optimal tradeoff. We split the samples randomly into four partitions, each having an independent and identical distribution of $\text{Poi}(n)$ samples, two from the multinomial distributions $P$ and other two from $Q$. See Algorithm 2 for a formal definition. We use one set of samples to identify the regime, and the other for estimation. We give a full description and justification of the algorithm in the longer version of this paper [23].

---

**Algorithm 2** Differential Privacy (DP) estimator

**Input:** target privacy $\varepsilon \in \mathbb{R}^+$, query $\mathcal{Q}$, neighboring databases $(\mathcal{D}, \mathcal{D}')$,
  samples from $P_{\mathcal{Q},\mathcal{D}}$ and $P_{\mathcal{Q},\mathcal{D}'}$, degree $K \in \mathbb{Z}^+$, constants $c_1, c_2 \in \mathbb{R}^+$, expected sample size $2n$
**Output:** estimate $\widehat{d}_{\varepsilon,K,c_1,c_2}(P_n \| Q_n)$ of $d_\varepsilon(P_{\mathcal{Q},\mathcal{D}} \| P_{\mathcal{Q},\mathcal{D}'})$
  $P \leftarrow P_{\mathcal{Q},\mathcal{D}}, Q \leftarrow P_{\mathcal{Q},\mathcal{D}'}$
  Draw four independent sample sizes: $N_{1,1}, N_{1,2}, N_{2,1}, N_{2,2} \sim \text{Poi}(n)$
  Sample from $P_{\mathcal{Q},\mathcal{D}}$: $\{X_{i,1}\}_{i=1}^{N_{1,1}} \in [S]^{N_{1,1}}$ and $\{X_{i,2}\}_{i=1}^{N_{1,2}} \in [S]^{N_{1,2}}$
  Sample from $P_{\mathcal{Q},\mathcal{D}'}$: $\{Y_{i,1}\}_{i=1}^{N_{2,1}} \in [S]^{N_{2,1}}$ and $\{Y_{i,2}\}_{i=1}^{N_{2,2}} \in [S]^{N_{2,2}}$
  $\hat{p}_{i,j} \leftarrow \frac{|\{\ell \in [N_{1,j}] : X_{\ell,j} = i\}|}{n}$ and $\hat{q}_{i,j} \leftarrow \frac{|\{\ell \in [N_{2,j}] : Y_{\ell,j} = i\}|}{n}$ for all $i \in [S]$ and $j \in \{1,2\}$
  $P_{n,1} \leftarrow [\hat{p}_{1,1}, \ldots, \hat{p}_{S,1}]$, $P_{n,2} \leftarrow [\hat{p}_{1,2}, \ldots, \hat{p}_{S,2}]$, $Q_{n,1} \leftarrow [\hat{q}_{1,1}, \ldots, \hat{q}_{S,1}]$ and $Q_{n,2} \leftarrow [\hat{q}_{1,2}, \ldots, \hat{q}_{S,2}]$
  **for** $i = 1$ **to** $S$ **do**
$$\delta_i \leftarrow \begin{cases} 0 & , \text{ if } \hat{p}_{i,1} - e^\varepsilon \hat{q}_{i,1} < -\sqrt{\frac{(c_1+c_2)\ln n}{n}}(\sqrt{\hat{p}_{i,1}} + \sqrt{e^\varepsilon \hat{q}_{i,1}}) \\ \hat{p}_{i,2} - e^\varepsilon \hat{q}_{i,2} & , \text{ if } \hat{p}_{i,1} - e^\varepsilon \hat{q}_{i,1} > \sqrt{\frac{(c_1+c_2)\ln n}{n}}(\sqrt{\hat{p}_{i,1}} + \sqrt{e^\varepsilon \hat{q}_{i,1}}) \\ \tilde{D}_K^{(1)}(\hat{p}_{i,2}, \hat{q}_{i,2}) & , \text{ if } \hat{p}_{i,1} + e^\varepsilon \hat{q}_{i,1} < \frac{c_1 \ln n}{n} \\ \tilde{D}_K^{(2)}(\hat{p}_{i,2}, \hat{q}_{i,2}; \hat{p}_{i,1}, \hat{q}_{i,1}) & , \text{ if } (\hat{p}_{i,1}, e^\varepsilon \hat{q}_{i,1}) \in U(c_1, c_2), \; \hat{p}_{i,1} + e^\varepsilon \hat{q}_{i,1} \geq \frac{c_1 \ln n}{n} \end{cases}$$
  **end for**
  $\widehat{d}_{\varepsilon,K,c_1,c_2}(P_n \| Q_n) \leftarrow 0 \vee (1 \wedge \sum_{i=1}^S \delta_i)$

---

**case 1:** For $(x, e^\varepsilon y) \in [0, (2c_1 \ln n)/n]^2$. A straightforward polynomial approximation of $[x - e^\varepsilon y]^+$ on $[0, (2c_1 \ln n)/n]^2$ cannot achieve approximation error smaller than $(1/K)((2c_1 \ln n)/n)$. As $K = c_3 \ln n$, this gives a bias of $1/n$ for each symbol in $[S]$, resulting in total bias of $S/n$. This requires $n \gg S$ to achieve arbitrary small error, as opposed to $n \gg S/\ln S$ which is what we are targeting. This is due to the fact that we are requiring multivariate approximation, and the bias is dominated by the worst case $y$ for each $x$. If $y$ is fixed, as in the case of univariate approximation in Lemma 14, the bias would have been $(1/K)\sqrt{(e^\varepsilon y 2c_1 \ln n)/n}$, with $y = q_i$, where total bias scales as $\sqrt{S/n \ln n}$ when summed over all symbols $i$.

Our strategy is to use the decomposition $[x - e^\varepsilon y]^+ = (\sqrt{x} + \sqrt{e^\varepsilon y})[\sqrt{x} - \sqrt{e^\varepsilon y}]^+$. Each function can be approximated up to a bias of $(1/K)\sqrt{(\ln n)/n}$, and the dominant term in the bias becomes $(1/K)\sqrt{(e^\varepsilon q_i \ln n)/n}$. This gives the desired bias. Concretely, we use two bivariate polynomials $u_K(x,y)$ and $v_K(x,y)$ to approximate $\sqrt{x} + \sqrt{y}$ and $[\sqrt{x} - \sqrt{y}]^+$ in $[0,1]^2$, respectively. Namely,

$$\sup_{(x,y)\in[0,1]^2} \left| u_K(x,y) - (\sqrt{x} + \sqrt{y}) \right| = \inf_{P\in\mathrm{poly}_K^2} \sup_{(x',y')\in[0,1]^2} \left| P(x',y') - (\sqrt{x'} + \sqrt{y'}) \right| , \text{ and } \quad (11)$$

$$\sup_{(x,y)\in[0,1]^2} \left| v_K(x,y) - [\sqrt{x} - \sqrt{y}]^+ \right| = \inf_{P\in\mathrm{poly}_K^2} \sup_{(x',y')\in[0,1]^2} \left| P(x',y') - [\sqrt{x'} - \sqrt{y'}]^+ \right| . \quad (12)$$

Denote $h_{2K}(x,y) = u_K(x,y)v_K(x,y) - u_K(0,0)v_K(0,0)$. Define

$$D_K^{(1)}(x,y) = \frac{2c_1 \ln n}{n} h_{2K}\left(\frac{xn}{2c_1 \ln n}, \frac{e^\varepsilon yn}{2c_1 \ln n}\right) , \quad (13)$$

for $(x, e^\varepsilon y) \in [0, (2c_1 \ln n)/n]^2$. In practice, one can use the best Chebyshev polynomial expansion to achieve the same uniform error rate, efficiently [24].

**case 2: For** $(x, e^\varepsilon y) \in U(c_1, c_1)$ **and** $x + e^\varepsilon y \geq (c_1 \ln n)/2n$**.** We utilize the best polynomial approximation of $|t|$ on $[-1, 1]$ with order $K$. Denote it as $R_K(t) = \sum_{j=0}^K r_j t^j$. Define

$$D_K^{(2)}(x,y;\hat{p}_{i,1}, \hat{q}_{i,1}) = \frac{1}{2}\sum_{j=0}^K r_j W^{-j+1}(e^\varepsilon y - x)^j + \frac{x - e^\varepsilon y}{2} , \quad (14)$$

where $W = \sqrt{(8c_1 \ln n)/n}\left(\sqrt{\hat{p}_{i,1}} + e^\varepsilon \hat{q}_{i,1}\right)$. Finally, we use second part of samples to construct unbiased estimator for $D_K^{(1)}(x,y)$ and $D_K^{(2)}(x,y;\hat{p}_{i,1}, \hat{q}_{i,1})$ by Lemma 11 and 12 . Namely,

$$\mathbb{E}\left[\tilde{D}_K^{(1)}(\hat{p}_{i,2}, \hat{q}_{i,2})\right] = D_K^{(1)}(p,q) , \text{ and} \quad (15)$$

$$\mathbb{E}\left[\tilde{D}_K^{(2)}(\hat{p}_{i,2}, \hat{q}_{i,2}; \hat{p}_{i,1}, \hat{q}_{i,1})|\hat{p}_{i,1}, \hat{q}_{i,1}\right] = D_K^{(2)}(p,q;\hat{p}_{i,1}, \hat{q}_{i,1}) . \quad (16)$$

The formula for the unbiased estimators can be found in the Appendix in Eqs. (180) and (187).

### 2.2.2 Minimax optimal upper bound

We provide an upper bound on the error achieved by the proposed estimator. The analysis uses similar techniques as the proof of Theorem 2. We provide a proof in Appendix C.5.

**Theorem 4.** *Suppose there exists a constant $C > 0$ such that $\ln n \leq C \ln S$. Then there exist constants $c_1, c_2$ and $c_3$ that only depends on $C$ and $\varepsilon$ such that*

$$\sup_{P,Q\in\mathcal{M}_S} \mathbb{E}_{P\times Q}\left[\left|\widehat{d}_{\varepsilon,K,c_1,c_2}(P_n\|Q_n) - d_\varepsilon(P\|Q)\right|^2\right] \lesssim \frac{e^\varepsilon S}{n\ln n} , \quad (17)$$

*for $K = c_3 \ln n$ where $\widehat{d}_{\varepsilon,K,c_1,c_2}$ is defined in Algorithm 2.*

It follows from the proof of Theorem 3 that

$$\inf_{\widehat{d}_\varepsilon(P_n\|Q_n)} \sup_{P,Q\in\mathcal{M}_S} \mathbb{E}_{P\times Q}\left[\left|\widehat{d}_\varepsilon(P_n\|Q_n) - d_\varepsilon(P\|Q)\right|^2\right] \gtrsim \frac{e^\varepsilon S}{n\ln n} .$$

Together, the above upper and lower bounds prove that the proposed estimator in Algorithm 2 is minimax optimal and cannot be improved upon in terms of sample complexity. We want to emphasize that we do not require to know the size of the support $S$, as opposed to exiting methods in [11], which requires collecting enough samples to identify the support. Comparing it to the error rate of plug-in estimator in Theorem 1, this minimax rate of $e^\varepsilon S/(n\ln n)$ demonstrates the *effective sample size amplification* holds; with $n$ samples, a sophisticated estimator can achieve the error rate equivalent to a plug-in estimator with $n\ln n$ samples.

# 3 Experiments

We present the experiment details in Appendix B and the code to reproduce our experiments at `https://github.com/xiyangl3/adp-estimator`. Figure 1 (a) illustrates the Mean Square Error (MSE) for estimating $d_\varepsilon(P\|Q)$ between uniform distribution $P$ and Zipf distribution $Q$, where the support size is fixed to be $S = 100$, $\text{Zipf}(\alpha) \propto 1/i^\alpha$, and $\alpha = -0.6$ for $i \in [S]$. The $\varepsilon$ is fixed to be $\varepsilon = 0.4$. This suggests that the Algorithm 2 consistently improves upon the plug-in estimator, as predicted by Theorem 4.

We demonstrate how we can use Algorithm 2 to detect mechanisms with false claim of DP guarantees on four types of mechanisms: Report Noisy Max [25], Histogram [26], Sparse Vector Technique [8] and Mixture of Truncated Geometric Mechanism. We closely following the experimental set-up of [11], and the settings and discussions are provided in Appendix B.2.

In [11], the test query and databases defining $(\mathcal{Q}, \mathcal{D}, \mathcal{D}')$ are chosen by some heuristics. Figure 1 (b) and (c) show $(\varepsilon, \delta)$ regions for the variations of noisy max mechanisms for privacy budget $\varepsilon_0 = 0.3$. From the figures, one can easily confirm that *RNM+Lap* and *RNM+Exp* have $\hat{\delta} \gg 0$ at $\varepsilon = 0.3$ (blue lines), confirming that these two mechanisms do not guarantee the claimed $(0.3, 0)$-DP, as known in the literature [11]. For those faulty mechanisms, Algorithm 2 also provides a certificate in the form of a set $T \subseteq [S]$ such that $[P(T) - e^\varepsilon Q(T)]^+ - \delta > 0$. With the setting of privacy budget $\varepsilon_0 = 0.5$, Figure 1 (d) shows that the incorrect histogram with incorrect $Lap(\varepsilon_0)$ noise is likely to be $(1/\varepsilon_0, 0)$-DP as known from [11]. Both mechanisms claim $(0.5, 0)$-DP, but the figure shows that the incorrect mechanism ensures $(1/0.5, 0)$-DP instead. Figure 1 (e) shows that *SVT* is likely to be $(\varepsilon_0, 0)$-DP with $\varepsilon_0 = 0.5$. However, *iSVT1*, *iSVT2*, and *iSVT3* do not meet the claimed $(0.5, 0)$-DP. Figure 1 (f) confirms that *MTGM* satisfied the claimed $(\varepsilon_0, \delta_0)$ differential privacy.

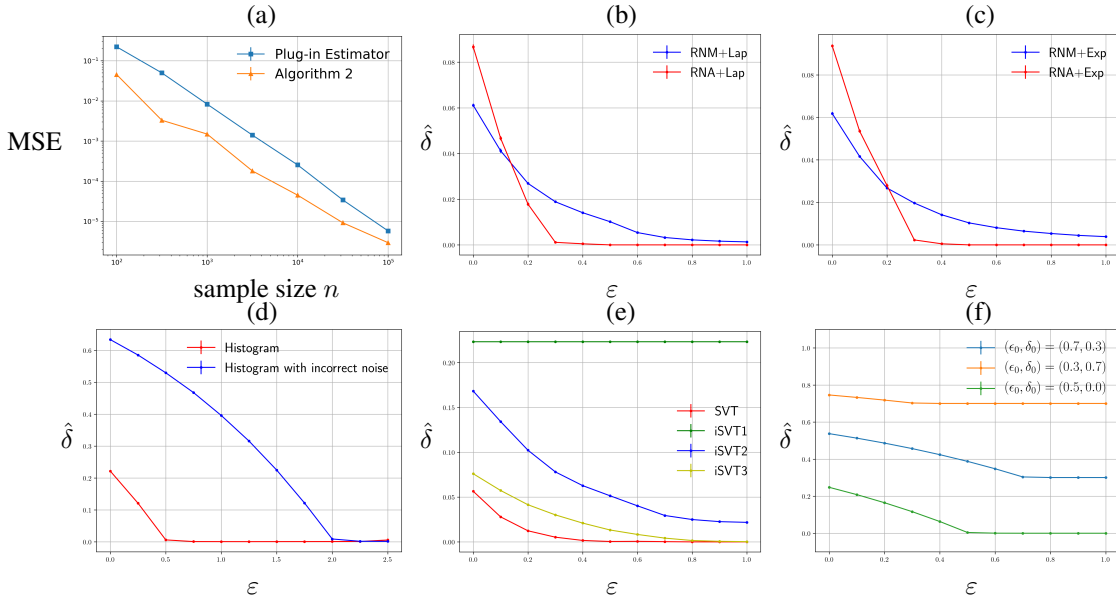

Figure 1: (a) shows the proposed minimax optimal estimator in Algorithm 2 consistently improves upon the plug-in estimator on synthetic data. Each data point represents 100 random trials, with standard error (SE) error bars smaller than the plot marker. (b), (c), (d), (e), (f) estimate $\hat{\delta}$ from Algorithm 2 of $\delta$ given $\varepsilon$, and privacy budget $\varepsilon_0$ for DP mechanisms. Each point is showing an average over 10 random trials with standard error. The red lines represent the original correct mechanisms. Algorithm 2 allows us to detect violation of claimed DP guarantees.

# 4 Conclusion

We investigate the fundamental trade-off between accuracy and sample size in estimating differential privacy guarantees from a black-box access to a purportedly private mechanism. Such a data-driven

approach to verifying privacy guarantees will allow us to hold accountable the mechanisms in the wild that are not faithful to the claimed privacy guarantees, and help find and fix bugs in either the design or the implementation. To this end, we propose a polynomial approximation based approach to estimate the differential privacy guarantees. We show that in the high-dimensional regime, the proposed estimator achieves *sample size amplification* effect. Compared to the parametric rate achieved by the plug-in estimator, we achieve a factor of $\ln n$ gain in the sample size. A matching lower bound proves the minimax optimality of our approach. Here, we list important remaining challenges that are outside the scope of this paper.

Since the introduction of differential privacy, there have been several innovative notions of privacy, such as pufferfish, concentrated DP, zCDP, and Renyi DP, proposed in [27, 28, 29, 30]. Our estimator builds upon the fact that differential privacy guarantee is a divergence between two random outputs. This is no longer true for the other notions of privacy, which makes it more challenging.

Characterizing the fundamental tradeoff for continuous mechanisms is an important problem, as several popular mechanisms output continuous random variables, such as Laplacian and Gaussian mechanisms. One could use non-parametric estimators such as $k$-nearest neighbor methods and kernel methods, popular for estimating information theoretic quantities and divergences [31, 32, 33, 34]. Further, when the output is a mixture of discrete and continuous variables, recent advances in estimating mutual information for mixed variables provide a guideline for such complex estimation process [35].

There is a fundamental connection between differential privacy and ROC curves, as investigated in [30, 36, 37]. Binary hypothesis testing and ROC curves provide an important measure of performance in generative adversarial networks (GAN) [38]. This fundamental connection between differential privacy and GAN was first investigated in [39], where it was used to provide an implicit bias for mitigating mode collapse, a fundamental challenge in training GANs. A DP estimator, like the one we proposed, provides valuable tools to measure performance of GANs. The main challenge is that GAN outputs are extremely high-dimensional (popular examples being $1,024 \times 1,024, \times 3$ dimensional images). Non-parametric methods have exponential dependence in the dimension, rendering them useless. Even some recent DP approaches have output dimensions that are equally large [40]. We need fundamentally different approach to deal with such high dimensional continuous mechanisms.

We considered a setting where we create synthetic databases $\mathcal{D}$ and $\mathcal{D}'$ and test the guarantees of a mechanism of interest. Instead, [12] assumes we do not have such a control, and the privacy of the real databases used in the testing needs to also be preserved. It is proven that one cannot test the privacy guarantee of a mechanism without revealing the contents of the test databases. Such fundamental limits suggest that the samples used in estimating DP needs to be destroyed after the estimation. However, the estimated $d_\varepsilon(P_{\mathcal{Q},\mathcal{D}} \| P_{\mathcal{Q},\mathcal{D}'})$ still leaks some information about the databases used, although limited. This is related to a challenging task of designing mechanisms with $(\varepsilon, \delta)$-DP guarantees when $(\varepsilon, \delta)$ also depends on the databases. Without answering any queries, just publishing the guarantee of the mechanism on a set of databases reveal something about the database. Detection and estimation under such complicated constraints is a challenging open question.

## Acknowledgement

This work is partially supported by NSF awards CNS-1527754, CNS-1705007, CCF-1927712, RI-1929955 and generous gift from Google.

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
