[Supplementary Material · nips_supp.pdf]

## Appendix

## A   Construction of $D_K(x; p)$ and corresponding estimator

We want to construct polynomial approximation of function $f(x; p_i) = [p_i - e^\varepsilon x]^+$ on $[0, 1]$ under smooth regimes. The resulting two-step polynomial estimator, has two characterizations, depending on where $p_i$ is. Let $\Delta = (c_1 \ln n)/n$.

**Case 1:** $p_i \le e^\varepsilon \Delta$ **and** $\hat{q}_i \in U(p_i; c_1, c_1) = [0, 2\Delta]$**.** We consider the function $g(y) = [p_i - e^\varepsilon 2\Delta y]^+$ by substituting $2\Delta y = x$ into $f(x; p_i) = [p_i - e^\varepsilon x]^+$. Let $H_K(y)$ be the best polynomial approximation of $g(y) \in C[0, 1]$ with order $K$, i.e. $H_K(y) = \arg\min_{P \in \text{poly}_K} \max_{y' \in [0,1]} |g(y') - P(y')|$ and denote it as $H_K(y) = \sum_{j=0}^{K} a_j y^j$. Then $D_K(x; p_i) = H_K(x/(2\Delta)) = \sum_{j=0}^{K} a_j(2\Delta)^{-j} x^j$. Once we have the polynomial approximation, we estimate with the uniformly minimum variance unbiased estimator (MVUE) to estimate $D_K(\hat{q}_i; p_i)$.

$$\tilde{D}_K(\hat{q}_i; p_i) = \sum_{j=0}^{K} a_j(2\Delta)^{-j} \prod_{k=0}^{j-1} (\hat{q}_i - \frac{k}{n}) . \tag{18}$$

Computing the $a_j$'s can be challenging, and we discuss this for the general case when $P$ is not known in Section 2.2.

**Case 2:** $p_i > e^\varepsilon \Delta$ **and** $\hat{q}_i \in [e^{-\varepsilon} p_i - \sqrt{e^{-\varepsilon} p_i \Delta}, e^{-\varepsilon} p_i + \sqrt{e^{-\varepsilon} p_i \Delta}]$**.** In this regime, the best polynomial approximation $D_K(x; p_i)$ of $[p_i - e^\varepsilon x]^+$ is given by

$$D_K(x; p_i) = \frac{e^\varepsilon}{2} \sum_{j=0}^{K} r_j \left(\sqrt{e^{-\varepsilon} p_i \Delta}\right)^{-j+1} (x - e^{-\varepsilon} p_i)^j + \frac{p_i - e^\varepsilon x}{2} ,$$

where $r_j$'s are defined from the best polynomial approximation $R_K(y)$ of $g(y) = |y|$ on $[-1, 1]$ with order $K$: $R_K(y) = \sum_{j=0}^{K} r_j y^j$. The unique uniformly minimum variance unbiased estimator (MVUE) for $(q_i - p_i)^j$ is

$$g_{j,p_i}(\hat{q}_i) \triangleq \sum_{k=0}^{j} \binom{j}{k} (-p_i)^{j-k} \prod_{h=0}^{k-1} \left(\hat{q}_i - \frac{h}{n}\right) ,$$

shown in Lemma 11. Hence,

$$\tilde{D}_K(x; p_i) = \frac{e^\varepsilon}{2} \left( \sum_{j=0}^{K} r_j \left(\sqrt{e^{-\varepsilon} p_i \Delta}\right)^{-j+1} g_{j,e^{-\varepsilon} p_i}(\hat{q}_i) + g_{1,e^{-\varepsilon} p_i}(\hat{q}_i) \right) .$$

The coefficients $r_j$'s only depend on $K$ and can be pre-computed and stored in a table.

## B   Details of Experiments

We implemented both plug-in estimator and Algorithm 2. Note that the coefficients of bivariate polynomial $u_K(x, y)$, $v_K(x, y)$ on $[0, 1]^2$ and $R_K(t)$ on $[-1, 1]$ are independent of data and $\varepsilon$. We pre-compute the coefficients and look them up from a table when running our estimators. Using numerical computation provided by Chebfun toolbox [41], we obtained the coefficients of $R_K$ by Remez algorithm, and the coefficients of bivariate polynomial $u_K$, $v_K$ by lowpass filtered Chebyshev expansion [24]. The time complexity of Algorithm 2 is $O(n \ln^2 n)$.

We compare the proposed Algorithm 2 and the plug-in estimator on synthetic data (Appendix B.1), and demonstrate the $(\varepsilon, \delta)$ regions for some of the popular differential privacy mechanisms and their variations (Appendix B.2). All experiments are done on a Macbook Pro with Intel® Core™ i5 processor and 8 GB memory. Our estimator is implemented in Python 3.6. We provide the code as a supplementary material.

## B.1 Synthetic experiments

Although in the analysis we make conservative choices of the constants $c_1$, $c_2$ and $c_3$, Algorithm 2 is not sensitive to those choices in practice and we fix them to be $c_1 = 4$, $c_2 = 0.1$ and $c_3 = 1.5$ in the experiments. The degree of polynomial approximation is chosen as $K = \lfloor c_3 \ln(n) \rfloor$. Figure 1 (a) illustrates the Mean Square Error (MSE) for estimating $d_\varepsilon(P\|Q)$ between uniform distribution $P$ and Zipf distribution $Q$, where the support size is fixed to be $S = 100$, $\text{Zipf}(\alpha) \propto 1/i^\alpha$, and $\alpha = -0.6$ for $i \in [S]$. The $\varepsilon$ is fixed to be $\varepsilon = 0.4$. Figure 2 shows results for a different choice of $\varepsilon = 0.2$ (left) and different distributions of Zipf and mixture of uniform (right). Each data point represents 100 random trials, with standard error (SE) error bars smaller than the plot marker. This suggests that the Algorithm 2 consistently improves upon the plug-in estimator, as predicted by Theorem 4.

Figure 2: (a) and (b) show the proposed minimax optimal estimator in Algorithm 2 consistently improves upon the plug-in estimator on synthetic data. Each data point represents 100 random trials, with standard error (SE) error bars smaller than the plot marker.

## B.2 Detecting violation of differential privacy with Algorithm 2

We demonstrate how we can use Algorithm 2 to detect mechanisms with false claim of DP guarantees on four types of mechanisms: Report Noisy Max [25], Histogram [26], Sparse Vector Technique [8] and Mixture of Truncated Geometric Mechanism. Following the experimental set-up of [11], the test query and databases defining $(\mathcal{Q}, \mathcal{D}, \mathcal{D}')$ are chosen by some heuristics, shown in Table 1. However, unlike the approach from [11], we do not require to know the size of the support $S$, we don't have to specify candidate bad events $E \subseteq [S]$, and we can estimate general approximate DP with $\delta > 0$. Throughout all the experiments, we fix $c_1 = 4$, $c_2 = 0.1$, $c_3 = 0.9$, and the mean of number of samples $n = 100000$. For the examples in Report Noisy Max, Histogram, and Sparse Vector Technique, we compose 5 and 10 queries together to form a one giant query. There are several categories of queries and databases to be tested, each represented by the true answer of the 5 queries in the table. When testing, we test all categories, and report the largest estimate $\hat{\delta}$ for each given $\varepsilon$.

Table 1: Database categories and samples [11]

| Category | $[\mathcal{Q}_1(\mathcal{D}), \ldots, \mathcal{Q}_5(\mathcal{D})]$ | $[\mathcal{Q}_1(\mathcal{D}'), \ldots, \mathcal{Q}_5(\mathcal{D}')]$ |
|---|---|---|
| One Above | [1, 1, 1, 1, 1] | [2, 1, 1, 1, 1] |
| One Below | [1, 1, 1, 1, 1] | [0, 1, 1, 1, 1] |
| One Above Rest Below | [1, 1, 1, 1, 1] | [2, 0, 0, 0, 0] |
| One Below Rest Above | [1, 1, 1, 1, 1] | [0, 2, 2, 2, 2] |
| Half Half | [1, 1, 1, 1, 1] | [0, 0, 0, 2, 2] |
| All Above & All Below | [1, 1, 1, 1, 1] | [2, 2, 2, 2, 2] |
| X Shape | [1, 1, 1, 1, 1] | [0, 0, 1, 1, 1] |

**Report Noisy Max.** For privacy budget $\varepsilon_0$, *Report Noisy Argmax with Laplace noise (RNA+Lap)* adds independent $Lap(2/\varepsilon_0)$ noise to query answers $\mathcal{Q}(\mathcal{D})$ and return the index of the largest noisy

query answer. *Report Noisy Argmax with Exponential noise (RNA+Exp)* adds $Exp(2/\varepsilon_0)$ noise instead of Laplace noise. Their claimed level of $(\varepsilon_0)$-DP is correctly guaranteed [25, Claim 3.9 and Theorem 3.10]. On the other hand, *Report Noisy Max with Laplace noise (RNM+Lap)* or *Report Noisy Max with Exponential noise (RNM+Exp)* return the largest noisy answer itself, instead of its index. This reveals more information than intended, leading to violation of claimed $(\varepsilon_0, 0)$-DP. Figure 1 (b) and (c) show $(\varepsilon, \delta)$ regions for the above variations of noisy max mechanisms for privacy budget $\varepsilon_0 = 0.3$. As expected, for each $\varepsilon_0$, *RNA+Lap* and *RNA+Exp* satisfy $(\varepsilon_0, 0)$-DP, whereas *RNM+Lap* and *RNM+Exp* have $\hat{\delta} > 0$.

**Histogram.** For privacy budget $\varepsilon_0$, *Histogram* takes histogram queries as input, adds independent $Lap(1/\varepsilon_0)$ noise to each query answers, and output the randomized query answers directly, which is proved to be $(\varepsilon_0, 0)$-DP [25]. As a comparison, *Histogram with incorrect noise* adds incorrect noise $Lap(\varepsilon_0)$. Note that as we use histogram queries, we require $\mathcal{Q}(\mathcal{D})$ and $\mathcal{Q}(\mathcal{D}')$ to be different in at most one element, which is tested on One Above and One Below samples as shown in Table 1. With the setting $\varepsilon_0 = 0.5$, Figure 1 (d) shows that the incorrect histogram is likely to be $(1/\varepsilon_0, 0)$-DP.

**Sparse Vector Technique (SVT).** We consider original sparse vector technique mechanism *SVT* [8], and its variations *iSVT1* [42], *iSVT2* [43], and *iSVT3* [44]. They are discussed in Section 1 and also studied and tested in [11]. Figure 1 (e) shows that *SVT* is likely to be $(\varepsilon_0, 0)$-DP. However, *iSVT1* and *iSVT2* are not likely to be pure differentially private for $\varepsilon \in [0, 1]$ with budget $\varepsilon_0 = 0.5$. As discussed in [8], *iSVT3* is in fact $(\frac{(1+6N)}{4}\varepsilon_0, 0)$-DP, where $N$ is the bound of number of trues in the output Boolean vector and set as $N = 1$ in this experiment. Figure 1 (e) shows that with $\hat{\delta} = 0$, $\varepsilon$ is likely to be in the range $[0.8, 0.9]$, which verifies the theoretic guarantee.

**Mixture of Truncated Geometric Mechanism (MTGM)** With privacy budget $\varepsilon_0$, *Truncated Geometric Mechanism (TGM)* proposed by [45] is provably to be $(\varepsilon_0, 0)$-DP. With probability privacy budget $\varepsilon_0$ and $\delta_0 \in [0, 1]$, *Mixture of Truncated Geometric Mechanism (MTGM)* outputs the original query answer with probability $\delta_0$, and outputs the randomized query answer with probability $1 - \delta_0$. *MTGM* can be proved to be $(\varepsilon_0, \delta_0)$-DP by composition theorem. Note that *TGM* and *MTGM* both take single counting query as query function $\mathcal{Q}$. In the experiment, we consider the single counting query with range $\{0, 1, 2, 3\}$. Figure 1 (f) shows that *MTGM* is likely to be $(\varepsilon_0, \delta_0)$ differentially private.

# C  Proofs

## C.1  Auxiliary lemmas

### C.1.1  Lemmas on Poisson distribution

**Lemma 1** ([46, Exercise 4.7]). *If $X \sim \text{Poi}(\lambda)$, then for any $\delta > 0$, we have*

$$\mathbb{P}\Big(X \geq (1+\delta)\lambda\Big) \quad \leq \quad \Big(\frac{e^\delta}{(1+\delta)^{1+\delta}}\Big)^\lambda \quad \leq \quad e^{-\delta^2\lambda/3} \vee e^{-\delta\lambda/3}, \tag{19}$$

$$\mathbb{P}\Big(X \leq (1-\delta)\lambda\Big) \quad \leq \quad \Big(\frac{e^{-\delta}}{(1-\delta)^{1-\delta}}\Big)^\lambda \quad \leq \quad e^{-\delta^2\lambda/2}. \tag{20}$$

**Lemma 2** ([46, Exercise 4.14]). *Suppose $X \sim \text{Poi}(\lambda_1)$, $Y \sim \text{Poi}(\lambda_2)$, and $Z = \alpha X + Y$, where $\alpha > 1$ is a constant. Then $\mathbb{E}[Z] = \alpha\lambda_1 + \lambda_2$, and for any $\delta > 0$, we have*

$$\mathbb{P}\Big(Z \geq (1+\delta)(\alpha\lambda_1 + \lambda_2)\Big) \quad \leq \quad e^{-\delta^2(\alpha\lambda_1+\lambda_2)/3} \vee e^{-\delta(\alpha\lambda_1+\lambda_2)/3}, \tag{21}$$

$$\mathbb{P}\Big(Z \leq (1-\delta)(\alpha\lambda_1 + \lambda_2)\Big) \quad \leq \quad e^{-\delta^2(\alpha\lambda_1+\lambda_2)/2}. \tag{22}$$

**Lemma 3.** *Suppose $n\hat{q} \sim \text{Poi}(nq)$, then*

$$\mathbb{E}\big[[\hat{q} - q]^+\big] \in \begin{cases} q\,e^{-nq} & , 0 \leq q \leq \frac{1}{n} \\ \Big[\sqrt{\frac{q}{4n}}, \sqrt{\frac{q}{2n}}\Big] & , q \geq \frac{1}{n} \end{cases} \tag{23}$$

*Hence,*

$$\frac{1}{2}\left(q \wedge \sqrt{\frac{q}{n}}\right) \;\leq\; \mathbb{E}[[\hat{q}-q]^+] \;\leq\; \left(q \wedge \sqrt{\frac{q}{n}}\right). \tag{24}$$

*Proof.* Let $\lambda = nq$, then

$$\begin{aligned}
\mathbb{E}\big[\,[\hat{q}-q]^+\,\big] &= \frac{1}{n}\sum_{k=\lfloor\lambda\rfloor+1}^{\infty}\frac{\lambda^k e^{-\lambda}}{k!}(k-\lambda) \\
&= \frac{1}{n}\sum_{k=\lfloor\lambda\rfloor+1}^{\infty}\frac{\lambda^k e^{-\lambda}}{k!}k \;-\; \frac{1}{n}\sum_{k=\lfloor\lambda\rfloor+1}^{\infty}\frac{\lambda^k e^{-\lambda}}{k!}\lambda \\
&= \frac{1}{n}\lambda\sum_{k=\lfloor\lambda\rfloor}^{\infty}\frac{\lambda^k e^{-\lambda}}{k!} \;-\; \frac{1}{n}\lambda\sum_{k=\lfloor\lambda\rfloor+1}^{\infty}\frac{\lambda^k e^{-\lambda}}{k!} \\
&= \frac{\lambda^{\lfloor\lambda\rfloor+1}e^{-\lambda}}{n\,\lfloor\lambda\rfloor!}
\end{aligned}$$

For $\lambda = nq \leq 1$, this is $qe^{-nq}$. For $\lambda \geq 1$, we use Stirling's approximation to get

$$\frac{\lambda^{\lfloor\lambda\rfloor+1}e^{-\lambda}}{n\,\lfloor\lambda\rfloor!} \;\in\; \left[\frac{1}{e},\frac{1}{\sqrt{2\pi}}\right] \times \frac{\lambda^{\lfloor\lambda\rfloor+1}\,e^{-\lambda}}{n\,\lfloor\lambda\rfloor^{\lfloor\lambda\rfloor+\frac{1}{2}}\,e^{-\lfloor\lambda\rfloor}}\;.$$

As $\dfrac{\lambda^{\lfloor\lambda\rfloor+\frac{1}{2}}\,e^{-\lambda}}{\lfloor\lambda\rfloor^{\lfloor\lambda\rfloor+\frac{1}{2}}\,e^{-\lfloor\lambda\rfloor}}$ is in $[1,\,1.12]$ for $\lambda \geq 1$, this gives the desired bound. $\qquad\square$

**Lemma 4.** *Suppose $n\hat{q} \sim \mathrm{Poi}\,(nq)$, then*

$$\mathbb{E}\big[\,[p-e^{\varepsilon}\hat{q}]^+ - [p-e^{\varepsilon}q]^+\,\big] \;\leq\; e^{\varepsilon}\min\left\{q, e^{-\varepsilon}p, \sqrt{\frac{q}{n}}, \sqrt{\frac{e^{-\varepsilon}p}{n}}\right\}. \tag{25}$$

*Proof.*

$$\begin{aligned}
\mathbb{E}\big[\,[p-e^{\varepsilon}\hat{q}]^+ - [p-e^{\varepsilon}q]^+\,\big] &= \mathbb{E}\Big[\frac{(p-e^{\varepsilon}\hat{q})+|p-e^{\varepsilon}\hat{q}|}{2} - \frac{(p-e^{\varepsilon}q)+|p-e^{\varepsilon}q|}{2}\Big] \\
&= \frac{1}{2}\Big(\mathbb{E}\big[\,|p-e^{\varepsilon}\hat{q}|\,\big] - |p-e^{\varepsilon}q|\Big)\;.
\end{aligned}$$

If $p \geq e^{\varepsilon}q$, then

$$\begin{aligned}
\frac{1}{2}\Big(\mathbb{E}\big[\,|p-e^{\varepsilon}\hat{q}|\,\big] - |p-e^{\varepsilon}q|\Big) &= \frac{1}{2}\Big(\mathbb{E}\big[\,|p-e^{\varepsilon}\hat{q}| - (p-e^{\varepsilon}q)\,\big]\Big) \\
&= \frac{1}{2}\Big(\mathbb{E}\big[\,(p-e^{\varepsilon}\hat{q})-(p-e^{\varepsilon}q)+2[e^{\varepsilon}\hat{q}-p]^+\,\big]\Big) \\
&= \mathbb{E}\big[\,[e^{\varepsilon}\hat{q}-p]^+\,\big] \\
&\leq e^{\varepsilon}\mathbb{E}\big[\,[\hat{q}-q]^+\,\big] \\
&\leq e^{\varepsilon}\Big(q \wedge \sqrt{\frac{q}{n}}\Big)\;,
\end{aligned}$$

where the second equality is because of the fact that $x = [x]^+ - [-x]^+$ and $|x| = [x]^+ + [-x]^+$, the first inequality follows from the fact that $[x]^+$ is monotone, and the last inequality follows from Lemma 3.

If $p < e^{\varepsilon}q$, then

$$\begin{aligned}
\frac{1}{2}\Big(\mathbb{E}\big[\,|p-e^{\varepsilon}\hat{q}|\,\big] - |p-e^{\varepsilon}q|\Big) &= \frac{1}{2}\Big(\mathbb{E}\big[\,|p-e^{\varepsilon}\hat{q}| - (e^{\varepsilon}q-p)\,\big]\Big) \\
&= \frac{1}{2}\Big(\mathbb{E}\big[\,(e^{\varepsilon}\hat{q}-p)-(e^{\varepsilon}q-p)+2[p-e^{\varepsilon}\hat{q}]^+\,\big]\Big) \\
&= \mathbb{E}\big[\,[p-e^{\varepsilon}\hat{q}]^+\,\big]\;,
\end{aligned}$$

Now we construct new random variable $\hat{p}$ by $n\hat{q} = ne^{-\varepsilon}\hat{p} + Z$, where $Z$ is independent of $\hat{p}$ and $Z \sim \mathrm{Poi}\,(n(q - e^{-\varepsilon}p))$. Hence, $e^{-\varepsilon}\hat{p} \leq \hat{q}$ with probability one. And the marginal distribution satisfies $ne^{-\varepsilon}\hat{p} \sim \mathrm{Poi}\,(ne^{-\varepsilon}p)$. We have

$$
\begin{aligned}
\mathbb{E}\big[\,[e^{-\varepsilon}p - \hat{q}]^+\,\big] &\leq \mathbb{E}\big[\,[e^{-\varepsilon}p - e^{-\varepsilon}\hat{p}]^+\,\big] \\
&\leq \Big(e^{-\varepsilon}p \wedge \sqrt{\frac{e^{-\varepsilon}p}{n}}\Big),
\end{aligned}
$$

where the first inequality follows from that fact that $[x]^+$ is monotone, and the last inequality follows from Lemma 3. $\qquad\square$

**Lemma 5.** *Suppose $n\hat{q} \sim \mathrm{Poi}\,(nq)$, then for any $p$ and $q$, we have*

$$
\mathrm{Var}\big(\,[p - e^{\varepsilon}\,\hat{q}]^+\,\big) \quad\lesssim\quad \frac{e^{\varepsilon}p}{n}\,. \tag{26}
$$

*Proof.* **Case 1: If $nq < 1$ and $e^{-\varepsilon}p \geq q$ or if $nq \geq 1$ and $ne^{-\varepsilon}p > \lfloor nq\rfloor - 1$:**

$$
\begin{aligned}
\mathrm{Var}\big(\,[p - e^{\varepsilon}\hat{q}]^+\,\big) &= \inf_a \mathbb{E}\big(\,[p - e^{\varepsilon}\hat{q}]^+ - a\,\big)^2 \\
&\leq \mathbb{E}\big(\,[p - e^{\varepsilon}\hat{q}]^+ - [p - e^{\varepsilon}q]^+\,\big)^2 \\
&= \mathbb{E}\big(\frac{(p - e^{\varepsilon}\hat{q}) - (p - e^{\varepsilon}q)}{2} + \frac{|p - e^{\varepsilon}\hat{q}| - |p - e^{\varepsilon}q|}{2}\big)^2 \\
&\leq \mathbb{E}\big(\frac{e^{\varepsilon}|\hat{q} - q|}{2} + \frac{\big||p - e^{\varepsilon}\hat{q}| - |p - e^{\varepsilon}q|\big|}{2}\big)^2 \\
&\leq e^{2\varepsilon}\mathbb{E}\big(\frac{|\hat{q} - q|}{2} + \frac{|\hat{q} - q|}{2}\big)^2 \\
&= e^{2\varepsilon}\mathbb{E}\big(\hat{q} - q\big)^2 \\
&= \frac{e^{2\varepsilon}q}{n} \\
&\leq \frac{e^{\varepsilon}p}{n}\,,
\end{aligned}
$$

where the last step follows from the assumption that $e^{-\varepsilon}p \geq q$.

**Case 2: If $nq < 1$ and $e^{-\varepsilon}p < q$ or if $nq \geq 1$ and $ne^{-\varepsilon}p < 1$ :**

In both cases, $ne^{-\varepsilon}p < 1$, and we have

$$
[ne^{-\varepsilon}p - n\hat{q}]^+ = \begin{cases} ne^{-\varepsilon}p & \text{w.p.} \quad e^{-nq} \\ 0 & \text{w.p.} \quad 1 - e^{-nq} \end{cases}\,,
$$

which is a Bernoulli random variable. The variance of it is

$$
\begin{aligned}
\mathrm{Var}\big([p - e^{\varepsilon}\hat{q}]^+\big) &= \frac{e^{2\varepsilon}}{n^2}\mathrm{Var}\big([ne^{-\varepsilon}p - n\hat{q}]^+\big) \\
&= p^2(1 - e^{-nq})e^{-nq} \\
&\leq p^2 < \frac{e^{\varepsilon}p}{n}\,,
\end{aligned}
$$

where we used the assumption that $ne^{-\varepsilon}p < 1$.

**Case 3: If $nq \geq 1$ and $1 \leq ne^{-\varepsilon}p \leq \lfloor nq\rfloor - 1$:**

Let $\lambda = nq$, and denote the random variable $X \sim \mathrm{Poi}\,(\lambda)$.

If $1 \le ne^{-\varepsilon}p \le \lfloor nq \rfloor - 1$, we know from Lemma 1 that

$$\mathbb{P}(X \le ne^{-\varepsilon}p) \le e^{-D_{\mathrm{KL}}(ne^{-\varepsilon}p \| nq)}, \tag{27}$$

where $D_{\mathrm{KL}}(k\|m) \triangleq m - k + k \ln(k/m)$.

We have

$$
\begin{aligned}
\mathbb{E}\big( [p - e^{\varepsilon}\hat{q}]^{+} \big)^2 &= \sum_{k=0}^{\lfloor ne^{-\varepsilon}p \rfloor} \Big( p - e^{\varepsilon}\frac{k}{n} \Big)^2 \Big( \frac{\lambda^k e^{-\lambda}}{k!} \Big) \\
&= p^2 \mathbb{P}(X \le ne^{-\varepsilon}p) - 2e^{\varepsilon}pq\mathbb{P}(X \le ne^{-\varepsilon}p - 1) \\
&\quad + \frac{e^{2\varepsilon}q}{n}\mathbb{P}(X \le ne^{-\varepsilon}p - 1) + e^{2\varepsilon}q^2\mathbb{P}(X \le ne^{-\varepsilon}p - 2) \\
&= p^2 \mathbb{P}(X \le ne^{-\varepsilon}p) - 2e^{\varepsilon}pq\mathbb{P}(X \le ne^{-\varepsilon}p) + 2e^{\varepsilon}pq\mathbb{P}(X = \lfloor ne^{-\varepsilon}p \rfloor) \\
&\quad + \frac{e^{2\varepsilon}q}{n}\mathbb{P}(X \le ne^{-\varepsilon}p) - \frac{e^{2\varepsilon}q}{n}\mathbb{P}(X = \lfloor ne^{-\varepsilon}p \rfloor) \\
&\quad + e^{2\varepsilon}q^2\mathbb{P}(X \le ne^{-\varepsilon}p) - e^{2\varepsilon}q^2\mathbb{P}(X = \lfloor ne^{-\varepsilon}p \rfloor) - e^{2\varepsilon}q^2\mathbb{P}(X = \lfloor ne^{-\varepsilon}p \rfloor - 1) \\
&= \Big( p^2 - 2e^{\varepsilon}pq + \frac{e^{2\varepsilon}q}{n} + e^{2\varepsilon}q^2 \Big)\mathbb{P}(X \le ne^{-\varepsilon}p) \\
&\quad + \Big( 2e^{\varepsilon}pq - \frac{e^{2\varepsilon}q}{n} - e^{2\varepsilon}q^2 - \frac{e^{2\varepsilon}q\lfloor ne^{-\varepsilon}p \rfloor}{n} \Big)\mathbb{P}(X = \lfloor ne^{-\varepsilon}p \rfloor) \\
&\lesssim \Big( (e^{\varepsilon}q - p)^2 \vee \frac{e^{2\varepsilon}q}{n} \Big)e^{-D_{\mathrm{KL}}(ne^{-\varepsilon}p \| nq)}.
\end{aligned}
$$

Let $y = q/(e^{-\varepsilon}p)$, as $1 \le ne^{-\varepsilon}p \le \lfloor nq \rfloor - 1$, we know $y > 1$.

We have

$$
\begin{aligned}
\frac{e^{2\varepsilon}q}{n}e^{-D_{\mathrm{KL}}(ne^{-\varepsilon}p \| nq)} &= \frac{e^{\varepsilon}py}{n}e^{-ne^{-\varepsilon}p(y-1-\ln y)} \\
&\lesssim \frac{e^{\varepsilon}p}{n},
\end{aligned}
$$

where we used the assumption that $ne^{-\varepsilon}p \ge 1$ and the inequality that $ye^{-(y-1-\ln y)}$ is bounded by some constant for $y > 1$.

We have

$$
\begin{aligned}
(e^{\varepsilon}q - p)^2 e^{-D_{\mathrm{KL}}(ne^{-\varepsilon}p \| nq)} &= (y-1)^2 p^2 e^{-ne^{-\varepsilon}p(y-1-\ln y)} \\
&\lesssim \frac{e^{\varepsilon}p}{n},
\end{aligned}
$$

where we used the assumption that $ne^{-\varepsilon}p \ge 1$ and the inequality that $(y-1)^2 e^{-x(y-1-\ln y)} \lesssim 1/x$ for $x, y > 1$.

It suffices to show that $f_y(x) = \ln x - x(y - 1 - \ln y) + 2\ln(y-1)$ is bounded by some constant for $x, y > 1$. Indeed, when $y - 1 - \ln y > 1$, $f_y'(x) = 1/x - (y - 1 - \ln y) < 0$. $f_y(x)$ is monotonically decreasing and $f_y(x) < f_y(1) = 2\ln(y-1) - y + 1 + \ln y$, which is bounded. When $y - 1 - \ln y < 1$, $f_y(x)$ attains maximum at $x = 1/(y - 1 - \ln y)$. In this case, we have $f_y(x) \le 2\ln(y-1) - \ln(y - 1 - \ln y) - 1$, which is also bounded.

$\square$

**Lemma 6.** *Suppose $n\hat{q} \sim \mathrm{Poi}(nq)$. Then,*

$$\mathbb{P}\big( e^{\varepsilon}\hat{q} \notin U(e^{2\varepsilon}q; c_1, c_1) \big) \le \frac{2}{n^{-c_1 e^{-\varepsilon}/3}} \le \frac{2}{n^{-c_1/3}}, \tag{28}$$

*and*

$$\mathbb{P}\big( \hat{q} \notin U(e^{\varepsilon}q; c_1, c_1) \big) \le \frac{2}{n^{-c_1/3}}. \tag{29}$$

*Proof.* The second inequality is exactly [21, Lemma 1]. We now prove the first inequality.

If $q \leq \frac{c_1 e^{-\varepsilon} \ln n}{n}$, we have

$$
\begin{aligned}
\mathbb{P}\left(e^\varepsilon \hat{q} \notin U(e^{2\varepsilon} q : c_1, c_1)\right) &= \mathbb{P}\left(n\hat{q} \geq 2c_1 e^{-\varepsilon} \ln n\right) \\
&\leq \mathbb{P}\left(\mathrm{Poi}\left(c_1 e^{-\varepsilon} \ln n\right) \geq 2c_1 e^{-\varepsilon} \ln n\right) \\
&\leq e^{-\frac{c_1 e^{-\varepsilon} \ln n}{3}},
\end{aligned}
$$

where we applied Lemma 1 in the last inequality.

If $q > \frac{c_1 e^{-\varepsilon} \ln n}{n}$, we have

$$
\begin{aligned}
\mathbb{P}\left(e^\varepsilon \hat{q} \notin U(e^{2\varepsilon} q; c_1, c_1)\right) &= \mathbb{P}\left(e^\varepsilon \hat{q} > e^\varepsilon q + \sqrt{\frac{c_1 e^\varepsilon q \ln n}{n}}\right) + \mathbb{P}\left(e^\varepsilon \hat{q} < e^\varepsilon q - \sqrt{\frac{c_1 e^\varepsilon q \ln n}{n}}\right) \\
&\leq \mathbb{P}\left(\mathrm{Poi}\left(nq\right) > nq + \sqrt{c_1 q e^{-\varepsilon} n \ln n}\right) + \mathbb{P}\left(\mathrm{Poi}\left(nq\right) < nq - \sqrt{c_1 q e^{-\varepsilon} n \ln n}\right) \\
&\leq e^{-\frac{c_1 e^{-\varepsilon} \ln n}{nq} \frac{nq}{3}} + e^{-\frac{c_1 e^{-\varepsilon} \ln n}{nq} \frac{nq}{2}} \\
&\leq \frac{2}{n^{c_1 e^{-\varepsilon}/3}}.
\end{aligned}
$$

$\square$

### C.1.2 Lemmas on the best polynomial approximation

The first-order and second-order symmetric difference with function $\varphi(x) = \sqrt{x(1-x)}$ are defined as

$$
\Delta_{h\varphi} f(x) \triangleq f(x + \frac{h\varphi(x)}{2}) - f(x - \frac{h\varphi(x)}{2}), \tag{30}
$$

and

$$
\Delta_{h\varphi}^2 f(x) = f(x + h\varphi(x)) - 2f(x) + f(x - h\varphi(x)), \tag{31}
$$

respectively.

For function $f(x)$ with domain $[0,1]$, the first-order Ditzian-Totik modulus of smoothness is defined as

$$
\omega_\varphi^1(f, t) \triangleq \sup_{0 < h \leq t} \|\Delta_{h\varphi}^1 f(x)\|_\infty, \tag{32}
$$

and the second-order Ditzian-Totik modulus of smoothness is defined as

$$
\omega_\varphi^2(f, t) \triangleq \sup_{0 < h \leq t} \|\Delta_{h\varphi}^2 f(x)\|_\infty. \tag{33}
$$

The following lemma upper bounds the best polynomial approximation error by the Ditzian-Totik moduli.

**Lemma 7** ([47, Theorem 7.2.1 and 12.1.1]). *There exists a constant $M(r) > 0$ such that for any function $f \in C[0,1]$,*

$$
\Delta_L[f; [0,1]] \leq M(r)\omega_\varphi^r(f, \frac{1}{L}), \quad L > r, \tag{34}
$$

*where $\Delta_L[f; I]$ denotes the distance of the function $f$ to the space $\mathrm{poly}_L$ in the uniform norm $\|\cdot\|_{\infty, I}$ on $I \subset \mathbb{R}$. Moreover, if $f(x) : [0,1]^2 \mapsto \mathbb{R}$, we have*

$$
\Delta_L[f; [0,1]^2] \leq M\omega_{[0,1]^2}^r(f, \frac{1}{L}), \quad L > r, \tag{35}
$$

*where $M$ is independent of $f$ and $L$, and $\Delta_L[f; [0,1]^2]$ denotes the distance of the function $f$ to the space $\mathrm{poly}_L^2$ in the uniform norm on $[0,1]^2$.*

**Lemma 8.** *For $f(x) = [p - e^\varepsilon 2x\Delta]^+$, for some $\Delta > 0$, $p \in [0, 2e^\varepsilon \Delta]$, $x \in [0, 1]$, and any integer $K \geq 1$,*

$$
\omega_\varphi^2(f, K^{-1}) = \begin{cases} p & \frac{p}{2e^\varepsilon \Delta} \leq \frac{1}{1+K^2} \\ \frac{\sqrt{p(2e^\varepsilon \Delta - p)}}{K} & \frac{1}{1+K^2} \leq \frac{p}{2e^\varepsilon \Delta} \leq \frac{K^2}{1+K^2} \\ e^\varepsilon \Delta - p & \frac{K^2}{1+K^2} \leq \frac{p}{2e^\varepsilon \Delta} \leq 1 \end{cases} \lesssim \min\left\{ p, \frac{\sqrt{p(2e^\varepsilon \Delta - p)}}{K}, e^\varepsilon \Delta - p \right\},
$$

*where $\omega_\varphi^2(f, t)$ is defined in Eq. (33).*

*Proof.* Let $g(x) := |p - 2e^\varepsilon \Delta x|$.

$$
\begin{aligned}
\Delta_{h\varphi}^2 f(x) &= [p - 2e^\varepsilon \Delta(x + h\varphi(x))]^+ - 2[p - 2e^\varepsilon \Delta x]^+ + [p - 2e^\varepsilon \Delta(x - h\varphi(x))]^+ \\
&= \frac{(p - 2e^\varepsilon \Delta(x + h\varphi(x))) - 2(p - 2e^\varepsilon \Delta x) + (p - 2e^\varepsilon \Delta(x - h\varphi(x)))}{2} + \\
&\quad \frac{\left|p - 2e^\varepsilon \Delta(x + h\varphi(x))\right| - 2\left|p - 2e^\varepsilon \Delta x\right| + \left|p - 2e^\varepsilon \Delta(x - h\varphi(x))\right|}{2} \\
&= \frac{1}{2}\Delta_{h\varphi}^2 g(x)
\end{aligned}
$$

Hence,

$$
\omega_\varphi^2(f, t) = \frac{1}{2}\omega_\varphi^2(g, t) \tag{36}
$$

It follows from [21, Lemma 12] that, for some $\Delta > 0$, $p \in [0, 2e^\varepsilon \Delta]$, $x \in [0, 1]$ and any integer $K \geq 1$, we have:

$$
\omega_\varphi^2(g, K^{-1}) = \begin{cases} 2p & \frac{p}{2e^\varepsilon \Delta} \leq \frac{1}{1+K^2} \\ \frac{2\sqrt{p(2e^\varepsilon \Delta - p)}}{K} & \frac{1}{1+K^2} \leq \frac{p}{2e^\varepsilon \Delta} \leq \frac{K^2}{1+K^2} \\ 2e^\varepsilon \Delta - p & \frac{K^2}{1+K^2} \leq \frac{p}{2e^\varepsilon \Delta} \leq 1 \end{cases},
$$

which implies the desired bound.

$\square$

**Lemma 9.** *Suppose $f(x) = [\sqrt{x} - \sqrt{a}]^+$, $x \in [0, 1]$ and $a \in [0, 1]$. Then*

$$
\omega_\varphi^1(f, t) \leq \frac{t}{\sqrt{2}}. \tag{37}
$$

*Similarly, suppose $f(x) = [\sqrt{a} - \sqrt{x}]^+$, $x \in [0, 1]$ and $a \in [0, 1]$. Then*

$$
\omega_\varphi^1(f, t) \leq \frac{t}{\sqrt{2}}. \tag{38}
$$

*Proof.* Let $g(x) = |\sqrt{x} - \sqrt{a}|$.

$$
\begin{aligned}
\Delta_{h\varphi}^1 f(x) &= \left| f\left(x + \frac{h\varphi(x)}{2}\right) - f\left(x - \frac{h\varphi(x)}{2}\right) \right| \\
&= \left| \left[ \sqrt{x + \frac{h\varphi(x)}{2}} - \sqrt{a} \right]^+ - \left[ \sqrt{x - \frac{h\varphi(x)}{2}} - \sqrt{a} \right]^+ \right| \\
&= \left| \frac{1}{2}\left( \left| \sqrt{x + \frac{h\varphi(x)}{2}} - \sqrt{a} \right| - \left| \sqrt{x - \frac{h\varphi(x)}{2}} - \sqrt{a} \right| \right) + \right. \\
&\qquad \left. \frac{\sqrt{x + \frac{h\varphi(x)}{2}} - \sqrt{x - \frac{h\varphi(x)}{2}}}{2} \right| \\
&\leq \frac{1}{2}\nabla_{h\varphi}^1 g(x) + \frac{1}{2}\left| \sqrt{x + \frac{h\varphi(x)}{2}} - \sqrt{x - \frac{h\varphi(x)}{2}} \right| \\
&= \frac{1}{2}\nabla_{h\varphi}^1 g(x) + \frac{1}{2}\left| \frac{h\varphi(x)}{\sqrt{x + h\varphi(x)/2} + \sqrt{x - h\varphi(x)/2}} \right| \\
&\leq \frac{1}{2}\nabla_{h\varphi}^1 g(x) + \frac{1}{2}\left| \frac{h\varphi(x)}{\sqrt{x + h\varphi(x)/2 + x - h\varphi(x)/2}} \right| \\
&\leq \frac{1}{2}\nabla_{h\varphi}^1 g(x) + \frac{h\sqrt{1-x}}{2\sqrt{2}} \\
&\leq \frac{t}{\sqrt{2}},
\end{aligned}
$$

where we used the fact that $\sqrt{x} + \sqrt{y} \leq x + y$ and [21, Lemma 11] for smoothness of $g$. $\qquad\square$

**Lemma 10** ([20, Lemma 27])**.** *Let $p_n(x) = \sum_{v=0}^{n} a_v x^v$ be a polynomial of degree at most $n$ such that $|p_n(x)| \leq A$ for $x \in [a, b]$. Then*

1. *If $a + b \neq 0$, then*

$$
|a_v| \leq 2^{7n/2} A \left| \frac{a+b}{2} \right|^{-v} \left( \left| \frac{b+a}{b-a} \right|^n + 1 \right), \qquad v = 0, 1, \cdots, n. \tag{39}
$$

2. *If $a + b = 0$, then*

$$
|a_v| \leq A b^{-v}(\sqrt{(2)} + 1)^n, \qquad v = 0, 1, \cdots, n. \tag{40}
$$

### C.1.3   Lemmas on the uniformly unbiased minimum variance unbiased estimator

**Lemma 11** ([21, Lemma 18])**.** *Suppose $nX \sim \mathrm{Poi}(np)$, $p \geq 0$, $q \geq 0$. Then, the estimator*

$$
g_{j,q}(X) \triangleq \sum_{k=0}^{j} \binom{j}{k}(-q)^{j-k} \prod_{h=0}^{k-1}\left( X - \frac{h}{n} \right) \tag{41}
$$

*is the unique uniformly minimum variance unbiased estimator for $(p-q)^j$, $j \geq 0$, $j \in \mathbb{N}$, and its second moment is given by*

$$
\mathbb{E}\left[ \left(g_{j,q}(X)\right)^2 \right] = \sum_{k=0}^{j} \binom{j}{k}^2 (p-q)^{2(j-k)} \frac{p^k k!}{n^k} = j! \left(\frac{p}{n}\right)^j L_j\left( -\frac{n(p-q)^2}{p} \right) \text{ assuming } p > 0, \tag{42}
$$

*where $L_m(x)$ stands for the Laguerre polynomial with order $m$, which is defined as:*

$$
L_m(x) = \sum_{k=0}^{m} \binom{m}{k}\frac{(-x)^k}{k!}. \tag{43}
$$

*If $M \geq \max\left\{\frac{n(p-q)^2}{p}, j\right\}$, we have*

$$\mathbb{E}\left[\left(g_{j,q}(X)\right)^2\right] \leq \left(\frac{2Mp}{n}\right)^j . \tag{44}$$

*When $k = 0$, $\prod_{h=0}^{k-1}\left(X - \frac{h}{n}\right) \triangleq 1$. When $p = 0$, $g_{j,q}(X) \equiv (-q)^j$, $\mathbb{E}\left[g_{j,q}(X)\right]^2 \equiv q^{2j}$.*

**Lemma 12.** *Suppose $(n\hat{p}, n\hat{q}) \sim \mathrm{Poi}(np) \times \mathrm{Poi}(nq)$. Then the following estimator using $(\hat{p}, \hat{q})$ is the unique uniformly minimum unbiased estimator for $(e^\varepsilon q - p)^j$, $j \geq 0$, $j \in \mathbb{Z}$:*

$$\hat{A}_j(\hat{p}, \hat{q}) = \sum_{k=0}^{j} \binom{j}{k} \prod_{i=0}^{k-1}\left(\hat{q} - \frac{i}{n}\right) e^{\varepsilon k}(-1)^{j-k} \prod_{m=0}^{j-k}\left(\hat{p} - \frac{m}{n}\right) . \tag{45}$$

*Furthermore,*

$$\mathbb{E}\hat{A}_j^2 \leq \left(2(e^\varepsilon q - p)^2 \vee \frac{8j(e^{2\varepsilon}q \vee p)}{n}\right)^j . \tag{46}$$

*Proof.* It follows from [21, Lemma 19] and binomial theorem that $\hat{A}_j(\hat{p}, \hat{q})$ is the unique uniformly minimum variance unbiased estimator for $(e^\varepsilon q - p)^j$. Now we show $\mathbb{E}\hat{A}_j^2$ is bounded.

It follows from binomial theorem again that for any fixed $r > 0$,

$$(e^\varepsilon q - p)^j = (e^\varepsilon q - r + r - p)^j \tag{47}$$

$$= \sum_{k=0}^{j} \binom{j}{k}(e^\varepsilon q - r)^k(-1)^{j-k}(p-r)^{j-k} . \tag{48}$$

The following estimator is also unbiased for estimating $(e^\varepsilon q - p)^j$,

$$\sum_{k=0}^{j} e^{\varepsilon k} g_{k, \frac{r}{e^\varepsilon}}(\hat{q})(-1)^{j-k} g_{j-k, r}(\hat{p}) , \tag{49}$$

where $g_{i,q}(\hat{p})$ is defined in Lemma 11.

Define $M_1 = \frac{n(q - \frac{r}{e^\varepsilon})^2}{p} \vee j$, $M_2 = \frac{n(p-r)^2}{p} \vee j$, $M = 2(e^\varepsilon q - p)^2 \vee \frac{8j(e^{2\varepsilon}q \vee p)}{n}$ and set $r = \frac{e^\varepsilon q + p}{2}$.

Denote $\|X\|_2 = \sqrt{\mathbb{E}(X - \mathbb{E}X)^2}$ for random variable $X$. It follows from Lemma 11 that

$$
\begin{aligned}
\|\hat{A}_j\|_2 &\leq \sum_{k=0}^{j} \binom{j}{k} e^{\varepsilon k} \|g_{k, \frac{r}{e^\varepsilon}}(\hat{q})\|_2 \cdot \|g_{j-k, r}(\hat{p})\|_2 \\
&\leq \sum_{k=0}^{j} \binom{j}{k} e^{\varepsilon k} \left(\frac{2M_1 q}{n}\right)^{k/2} \left(\frac{2M_2 p}{n}\right)^{(j-k)/2} \\
&= \left(e^\varepsilon \sqrt{\frac{2M_1 q}{n}} + \sqrt{\frac{2M_2 p}{n}}\right)^j \\
&= \left(\sqrt{\frac{(e^\varepsilon q - p)^2}{2} \vee \frac{2je^{2\varepsilon}q}{n}} + \sqrt{\frac{(e^\varepsilon q - p)^2}{2} \vee \frac{2jp}{n}}\right)^j \\
&= M^{j/2} .
\end{aligned}
$$

$\square$

## C.2 Proof of Theorem 1

In this section we prove a more general statement in the following theorem, in which case Theorem 1 follows as a corollary.

**Theorem 5.** *For any $\varepsilon \geq 0$, support size $S \in \mathbb{Z}^+$, and distribution $P \in \mathcal{M}_S$, the plug-in estimator satisfies*

$$\sup_{Q \in \mathcal{M}_S} \mathbb{E}_Q \left[ |d_\varepsilon(P\|Q_n) - d_\varepsilon(P\|Q)|^2 \right] \lesssim \left( \sum_{i=1}^{S} p_i \wedge \sqrt{\frac{e^\varepsilon p_i}{n}} \right)^2 + \frac{e^\varepsilon}{n}, \tag{50}$$

*with expected number of samples $n$. If $S \geq 2$, we can also lower bound the worst case mean squared error as*

$$\sup_{Q \in \mathcal{M}_S} \mathbb{E}_Q \left[ |d_\varepsilon(P\|Q_n) - d_\varepsilon(P\|Q)|^2 \right] \gtrsim \left( \sum_{i=1}^{S} p_i \wedge \sqrt{\frac{e^\varepsilon p_i}{n}} \right)^2. \tag{51}$$

Note that

$$\mathbb{E}_Q \left[ |d_\varepsilon(P\|Q_n) - d_\varepsilon(P\|Q)|^2 \right] = \left( \sum_{i=1}^{S} \mathbb{E}_Q \left[ [p_i - e^\varepsilon \hat{q}_i]^+ \right] - [p_i - e^\varepsilon q_i]^+ \right)^2 + \mathrm{Var}\left( d_\varepsilon(P\|Q_n) \right). \tag{52}$$

We first claim the following upper bound for all $P$:

$$\sum_{i=1}^{S} \mathbb{E}_Q \left[ [p_i - e^\varepsilon \hat{q}_i]^+ \right] - [p_i - e^\varepsilon q_i]^+ \leq \sum_{i=1}^{S} p_i \wedge \sqrt{\frac{e^\varepsilon p_i}{n}}, \tag{53}$$

where the inequality follows from Lemma 4.

For the upper bound of the variance term in Eq. (52), we have

$$\mathrm{Var}\left( d_\varepsilon(P_n\|Q) \right) = \sum_{i=1}^{S} \mathrm{Var}\left( [p_i - e^\varepsilon \hat{q}_i]^+ \right) \lesssim \sum_{i=1}^{S} \frac{e^\varepsilon q_i}{n} = \frac{e^\varepsilon}{n}, \tag{54}$$

where the inequality follows from Lemma 5.

We next construct $Q$ to get the lower bound. Let

$$q_i = \begin{cases} e^{-\varepsilon} p_i & , i \in S_+ \\ \frac{1 - Q(S_+)}{S - |S_+|} & , i \in S_- \end{cases} \tag{55}$$

where $S_+$ is a set of indices satisfying $Q(S_+) = \sum_{i \in S_+} q_i \leq e^{-\varepsilon}$.

Note that each term in the bias of Eq. (52) is non-negative via Jensen's inequality, which gives

$$\sum_{i=1}^{S} \mathbb{E}_Q \left[ [p_i - e^\varepsilon \hat{q}_i]^+ \right] - [p_i - e^\varepsilon q_i]^+ \geq e^\varepsilon \sum_{i \in S_+} \mathbb{E}_Q \left[ [q_i - \hat{q}_i]^+ \right] \tag{56}$$

$$\gtrsim \sum_{i \in S_+} \left\{ p_i \wedge \sqrt{\frac{e^\varepsilon p_i}{n}} \right\}, \tag{57}$$

where we used Lemma 3. Note that we can choose $Q$ such that $|S_+| = S/3$. This implies the desired lower bound when plugged into Eq. (52).

## C.3 Proof of Theorem 2

In this section we prove a more general statement in the following theorem, in which case Theorem 2 follows as a corollary.

**Theorem 6.** *For any $P$, suppose $c \ln S \leq \ln n \leq C \ln(e^{-\varepsilon} \sum_{i=1}^{S} \sqrt{e^{\varepsilon} p_i} \wedge p_i \sqrt{n \ln n})$ for some constants $c$ and $C$, then there exist constants $c_1, c_2$ and $c_3$ that only depends on $c$, $C$ and $\varepsilon$ such that*

$$\sup_{Q \in \mathcal{M}_S} \mathbb{E}_Q \big[ \, \big| \widehat{d}_{\varepsilon,K,c_1,c_2}(P\|Q_n) - d_\varepsilon(P\|Q) \big|^2 \, \big] \; \lesssim \; \Big( \sum_{i=1}^{S} p_i \wedge \sqrt{\frac{e^{\varepsilon} p_i}{n \ln n}} \Big)^2 , \tag{58}$$

*for $K = c_3 \ln n$ and where $\widehat{d}_{\varepsilon,K,c_1,c_2}$ is defined in Algorithm 1.*

Define good events, where our choice of the regimes are correct as

$$E \;\triangleq\; \Big\{ \{i : \hat{q}_{i,1} > U(p_i; c_1, c_2)\} \subseteq S^+ \Big\} \cap \Big\{ \{i : \hat{q}_{i,1} < U(p_i; c_1, c_2)\} \subseteq S^- \Big\}$$
$$\cap \Big\{ \{i : \hat{q}_{i,1} \in U(p_i, c_1, c_2) \subseteq \{i : q_i \in U(p_i, c_1, c_1)\}\} \Big\} , \tag{59}$$

where $S^+ = \{i : e^\varepsilon q_i \geq p_i\}$ and $S^- = \{i : e^\varepsilon q_i \leq p_i\}$. Decompose the error under the good events as

$$\mathcal{E}_1 \;\triangleq\; \sum_{i \in I_1} \big\{ [p_i - e^\varepsilon \hat{q}_{i,2}]^+ - [p_i - e^\varepsilon q_i]^+ \big\} , \tag{60}$$

$$\mathcal{E}_2 \;\triangleq\; \sum_{i \in I_2} \big\{ \tilde{D}_K(\hat{q}_{i,2}; p_i) - [p_i - e^\varepsilon q_i]^+ \big\} . \tag{61}$$

where the indices of those regimes under the good event are

$$I_1 \;\triangleq\; \{i : \hat{q}_{i,1} < U(p_i; c_1, c_2), e^\varepsilon q_i \leq p_i\} \tag{62}$$
$$I_2 \;\triangleq\; \{i : \hat{q}_{i,1} \in U(p_i; c_1, c_2), q_i \in U(p_i; c_1, c_1)\} . \tag{63}$$

We can bound the squared error as

$$\begin{aligned} \mathbb{E}\big[ \big(\widehat{d}_{\varepsilon,K,c_1,c_2}(P\|Q_n) - d_\varepsilon(P\|Q)\big)^2 \big] &\leq& \mathbb{E}\big[ \big(\widehat{d}_{\varepsilon,K,c_1,c_2}(P\|Q_n) - d_\varepsilon(P\|Q)\big)^2 \mathbb{I}(E) \big] + \mathbb{P}(E^c) \\ &\leq& \mathbb{E}[(\mathcal{E}_1 + \mathcal{E}_2)^2] + \mathbb{P}(E^c) \\ &\leq& 2\mathbb{E}[(\mathcal{E}_1)^2] + 2\mathbb{E}[(\mathcal{E}_2)^2] + \mathbb{P}(E^c) , \end{aligned} \tag{64}$$

The last term on the bad event is bounded by $3S/n^\beta$ as shown in the following lemma, and a proof is provided in Appendix C.3.1. This is a direct consequence of standard concentration inequality for Poisson variables.

**Lemma 13.** *Let $\beta = \min\{\frac{c_2^2}{3c_1}, \frac{(c_1 - c_2)^2}{4c_1}, \frac{(\sqrt{c_1} - \sqrt{c_2})^2}{3}\}$, then for the good event $E$ defined in (59),*

$$\mathbb{P}(E^c) \;\leq\; \frac{3S}{n^\beta} . \tag{65}$$

The first term is bounded by $e^\varepsilon/n$, as

$$\mathbb{E}[(\mathcal{E}_1)^2] \;=\; \mathbb{E}[\mathrm{Var}(\mathcal{E}_1|I_1) + (E[\mathcal{E}_1|I_1])^2] \;=\; \mathbb{E}[\mathrm{Var}(\mathcal{E}_1|I_1)] \;\leq\; \sum_{i=1}^{S} \frac{e^\varepsilon p_i}{n} \;\leq\; \frac{e^\varepsilon}{n} \tag{66}$$

where we used Lemma 5 and the fact that $\mathbb{E}[\mathcal{E}_1|I_1] = 0$ with probability one.

The second term is bounded by the following lemma, with a proof in Appendix C.3.2.

**Lemma 14.** *For $n\hat{q} \sim \mathrm{Poi}(nq)$ and $q \in U(p; c_1, c_1)$, there exists a universal constant $B > 0$ such that*

$$\big| \mathbb{E}[\tilde{D}_K(\hat{q}; p)] - [p - e^\varepsilon q]^+ \big| \;\lesssim\; p \wedge \frac{1}{K} \sqrt{\frac{e^\varepsilon p c_1 \ln n}{n}} , \text{ and} \tag{67}$$

$$\mathrm{Var}\big(\tilde{D}_K(\hat{q}; p)\big) \;\lesssim\; \frac{B^K e^\varepsilon c_1 \ln n}{n}(p + e^\varepsilon q) , \tag{68}$$

*where $\tilde{D}_K(\hat{p}; q)$ is the uniformly minimum variance unbiased estimate (MVUE) defined in Eq. (18), $U(q, c_1)$ is defined in Eq. (5), and $K = c_3 \ln n$ for some $c_3 < c_1$.*

We have

$$\mathbb{E}[(\mathcal{E}_2)^2] \quad \lesssim \quad \sum_{i=1}^{S} \frac{B^K e^\varepsilon c_1 \ln n}{n}(p_i + e^\varepsilon q_i) + \Big(\sum_{i=1}^{S} p_i \wedge \frac{1}{K}\sqrt{\frac{e^\varepsilon p_i c_1 \ln n}{n}}\Big)^2 \tag{69}$$

$$\lesssim \quad \frac{c_1 \ln n}{n^{1-c_3 \ln B}} e^\varepsilon(e^\varepsilon + 1) + \Big(\sum_{i=1}^{S} p_i \wedge \sqrt{\frac{e^\varepsilon p_i c_1}{c_3^2 n \ln n}}\Big)^2 . \tag{70}$$

Substituting bounds (70), (66) and (65), we get that

$$\mathbb{E}\big[\big(\widehat{d}_{\varepsilon,K,c_1,c_2}(P\|Q_n) - d_\varepsilon(P\|Q)\big)^2\big] \quad \lesssim \quad \frac{c_1 \ln n}{n^{1-c_3 \ln B}} e^{2\varepsilon} + \Big(\sum_{i=1}^{S} p_i \wedge \sqrt{\frac{e^\varepsilon p_i c_1}{c_3^2 n \ln n}}\Big)^2 + \frac{S}{n^\beta} \tag{71}$$

where we use the fact that $\frac{e^\varepsilon}{n} \lesssim \frac{c_1 \ln n}{n^{1-c_3 \ln B}} e^\varepsilon(e^\varepsilon + 1)$.

As $\ln n \gtrsim \ln S$, one may choose $c_1$ large enough to and let $c_2 = c_1/2$ to ensure that $\frac{S}{n^\beta} \lesssim \frac{c_1 \ln n}{n^{1-c_3 \ln B}} e^{2\varepsilon}$. As $\ln n \lesssim \ln\big(e^{-\varepsilon}\sum_{i=1}^{S}\sqrt{e^\varepsilon p_i} \wedge p_i\sqrt{n \ln n}\big)$, one may choose $c_3$ small enough to ensure $\frac{c_1 \ln n}{n^{1-c_3 \ln B}} e^{2\varepsilon} \lesssim \big(\sum_{i=1}^{S} p_i \wedge \sqrt{\frac{e^\varepsilon p_i c_1}{c_3^2 n \ln n}}\big)^2$. The worst case of $P$ result is proved upon noting

$$\sum_{i=1}^{S} p_i \wedge \sqrt{\frac{e^\varepsilon p_i}{n \ln n}} \quad \leq \quad \sum_{i=1}^{S} \sqrt{\frac{e^\varepsilon p_i}{n \ln n}} \quad \leq \quad \sqrt{\frac{e^\varepsilon S}{n \ln n}} . \tag{72}$$

Note that in the worst case of $P$, we do not require $\ln n \gtrsim \ln S$, as we can take $c_1$ large enough and $c_2 = c_1/2$ to ensure $\frac{S}{n^\beta} \lesssim \frac{e^\varepsilon S}{n \ln n}$.

### C.3.1   Proof of Lemma 13

Let $E_1 = \Big\{\{i : \hat{q}_{i,1} > U(p_i; c_1, c_2)\} \subseteq S^+\Big\}$, $E_2 = \Big\{\{i : \hat{q}_{i,1} < U(p_i; c_1, c_2)\} \subseteq S^-\Big\}$ and $E_3 = \Big\{\{i : \hat{q}_{i,1} \in U(p_i, c_1, c_2) \subseteq \{i : q_i \in U(p_i, c_1, c_1)\}\}\Big\}$. We first show $\mathbb{P}(E_1^c) \leq Sn^{-\beta}$ for $\beta \leq (c_2)^2/(3c_1)$.

$$\begin{aligned}
\mathbb{P}(E_1^c) \quad &= \quad \mathbb{P}\Big(\bigcup_{i=1}^{S}\{\hat{q}_{i,1} > U(p_i; c_1, c_2), e^\varepsilon q_i < p_i\}\Big) \\
&\leq \quad S \max_{i \in S} \mathbb{P}(\{\hat{q}_{i,1} > U(p_i; c_1, c_2), e^\varepsilon q_i < p_i\}) \\
&= \quad S \max_{i \in S} \mathbb{P}(\{\mathrm{Poi}\,(nq_i) > nU(p_i; c_1, c_2), q_i < e^{-\varepsilon}p_i\}) \\
&\leq \quad S \max_{i \in S} \mathbb{P}(\{\mathrm{Poi}\,(ne^{-\varepsilon}p_i) > nU(p_i; c_1, c_2)\})
\end{aligned}$$

If $p_i \leq (c_1 e^\varepsilon \ln n)/n$, it follows from Lemma 1 that

$$\begin{aligned}
\mathbb{P}(\{\mathrm{Poi}\,(ne^{-\varepsilon}p_i) > nU(p_i; c_1, c_2)\}) \quad &= \quad \mathbb{P}(\{\mathrm{Poi}\,(ne^{-\varepsilon}p_i) > (c_1 + c_2)\ln n\}) \\
&\leq \quad \mathbb{P}(\{\mathrm{Poi}\,(c_1 \ln n) > (c_1 + c_2)\ln n\}) \\
&\leq \quad e^{-\frac{c_2^2}{3c_1}\ln n} .
\end{aligned}$$

If $p_i > (c_1 e^\varepsilon \ln n)/n$, it follows from Lemma 1 that

$$\begin{aligned}
\mathbb{P}(\{\mathrm{Poi}\,(ne^{-\varepsilon}p_i) > nU(p_i; c_1, c_2)\}) \quad &= \quad \mathbb{P}(\{\mathrm{Poi}\,(ne^{-\varepsilon}p_i) > ne^{-\varepsilon}p_i + \sqrt{c_2 e^{-\varepsilon}p_i n \ln n}\}) \\
&\leq \quad e^{-\frac{c_2 \ln n}{3}} .
\end{aligned}$$

Together, these bounds imply that $\mathbb{P}(E_1^c) \leq Sn^{-\beta}$.

Next, we show $\mathbb{P}(E_2^c) \leq Sn^{-\beta}$, for positive constant $\beta \leq c_2^2/(3c_1)$. Recall that

$$\mathbb{P}(E_2^c) \quad = \quad \mathbb{P}\Big(\bigcup_{i=1}^{S}\{\hat{q}_{i,1} < U(p_i; c_1, c_1), e^\varepsilon q_i \geq p_i\}\Big) .$$

If $p_i \leq (c_1 e^\varepsilon \ln n)/n$, $\mathbb{P}(\{\hat{q}_{i,1} < U(p_i; c_1, c_2), e^\varepsilon p_i \geq q_i\}) = 0$. If $p_i > (c_1 e^\varepsilon \ln n)/n$, it follows from Lemma 1 that

$$
\begin{aligned}
\mathbb{P}(\{\hat{q}_{i,1} < U(p_i; c_1, c_2), e^\varepsilon q_i \geq p_i\}) &= \mathbb{P}(\{\mathrm{Poi}\,(nq_i) < ne^{-\varepsilon}p_i - \sqrt{c_2 e^{-\varepsilon}p_i n \ln n}, q_i \geq e^{-\varepsilon}p_i\}) \\
&\leq \mathbb{P}(\{\mathrm{Poi}\,(ne^{-\varepsilon}p_i) < ne^{-\varepsilon}p_i - \sqrt{c_2 e^{-\varepsilon}p_i n \ln n}, q_i \geq e^{-\varepsilon}p_i\}) \\
&\leq e^{-\frac{c_2 \ln n}{2}} \;.
\end{aligned}
$$

As $c_2/2 \geq c_2^2/(3c_1)$, we have $\mathbb{P}(E_2^c) \leq Sn^{-\beta}$.

Finally, we show that $\mathbb{P}(E_3^c) \leq Sn^{-\beta}$ for $\beta \leq \min\{(c_1 - c_2)^2/(4c_1), (\sqrt{c_1} - \sqrt{c_2})^2/3\}$. Recall

$$
\mathbb{P}(E_3^c) = \mathbb{P}(\bigcup_{i=1}^{S}\{\hat{q}_{i,1} \in U(p_i; c_1, c_2), q_i \notin U(p_i; c_1, c_1)\}) \;.
$$

If $p_i \leq (c_1 e^\varepsilon \ln n)/n$,

$$
\begin{aligned}
\mathbb{P}(\{\hat{q}_{i,1} \in U(p_i; c_1, c_2), q_i \notin U(p_i; c_1, c_1)\}) &= \mathbb{P}(\{\mathrm{Poi}\,(nq_i) \leq (c_1 + c_2)\ln n, nq_i \geq 2c_1 \ln n\}) \\
&\leq \mathbb{P}(\{\mathrm{Poi}\,(2c_1 \ln n) \leq (c_1 + c_2)\ln n\}) \\
&\leq e^{-\frac{(c_1 - c_2)^2}{4c_1}\ln n} \;.
\end{aligned}
$$

If $p_i > (c_1 e^\varepsilon \ln n)/n$,

$$
\begin{aligned}
&\mathbb{P}(\{\hat{q}_{i,1} \in U(p_i; c_1, c_2), q_i > U(p_i; c_1, c_1)\}) \\
&= \mathbb{P}(\{\mathrm{Poi}\,(nq_i) \leq ne^{-\varepsilon}p_i + \sqrt{c_2 e^{-\varepsilon}p_i n \ln n}, nq_i > ne^{-\varepsilon}p_i + \sqrt{c_1 e^{-\varepsilon}p_i n \ln n}\}) \\
&\leq \mathbb{P}(\{\mathrm{Poi}\,(ne^{-\varepsilon}p_i + \sqrt{c_1 e^{-\varepsilon}p_i n \ln n}) \leq ne^{-\varepsilon}p_i + \sqrt{c_2 e^{-\varepsilon}p_i n \ln n}\}) \\
&\leq e^{-\left(\frac{(\sqrt{c_1} - \sqrt{c_2})\sqrt{e^{-\varepsilon}p_i n \ln n}}{ne^{-\varepsilon}p_i + \sqrt{c_1 e^{-\varepsilon}p_i n \ln n}}\right)^2 \frac{1}{2}(ne^{-\varepsilon}p_i + \sqrt{c_1 e^{-\varepsilon}p_i n \ln n})} \\
&\leq e^{-\frac{(\sqrt{c_1} - \sqrt{c_2})^2 \ln n}{4}} \;,
\end{aligned}
$$

Similarly, we can show that $\mathbb{P}(\{\hat{q}_{i,1} \in U(p_i; c_1, c_2), q_i < U(p_i; c_1, c_1)\}) \leq e^{-(\sqrt{c_1} - \sqrt{c_2})^2 \ln n/3}$.

### C.3.2 Proof of Lemma 14

Let $\Delta = (c_1 \ln n)/n$. We divide the analysis into two regimes.

**Case 1: $p \leq e^\varepsilon \Delta$ and $q \in U(p, c_1, c_1) = [0, 2\Delta]$.**

First we analyze the bias. As we apply the universally minimum variance unbiased estimator (MVUE) to $D_K(q; p)$, the bias is entirely due to the functional approximation. Recall that we consider the best polynomial approximation $H_K(y)$ of function $g(y) = [p - e^\varepsilon 2\Delta y]^+$ on $[0, 1]$ with order $K$, i.e. $H_K(y) = \arg\min_{P \in \mathrm{poly}_K} \max_{y' \in [0,1]} |g(y') - P(y')|$. Denote it as $H_K(y) = \sum_{j=0}^{K} a_j y^j$. Then $D_K(x; p) = H_K(x/(2\Delta))$. It follows from Lemma 8 that there exists a universal constant $M \geq 0$ such that for all $K \geq 1$,

$$
\begin{aligned}
\sup_{x \in [0, 2\Delta]} |D_K(x; p) - [p - e^\varepsilon x]^+| &= \sup_{y \in [0,1]} |D_K(2\Delta y; p) - [p - e^\varepsilon 2\Delta y]^+| \\
&= \sup_{y \in [0,1]} |H_K(y) - g(y)| \\
&\leq M\left(p \wedge \frac{1}{K}\sqrt{\frac{e^\varepsilon c_1 p \ln n}{n}}\right) \;.
\end{aligned} \tag{73}
$$

Next to analyze the variance, we upper bound the magnitude of the coefficients in $\tilde{D}_K$ using Lemma 10, and upper bound the second moment of the unique MVUE using the tail bound of Poisson distribution in Lemma 11. As the universal MVUE is of the form $\sum_{j=0}^{K} a_j (2\Delta)^{-j} \prod_{k=0}^{j-1}(\hat{p} - $

$k/n)$ as shown in Appendix C.1.3, The variance is upper bounded by

$$
\begin{aligned}
\mathrm{Var}(\,\tilde{D}_K(\hat{q};p)\,) \quad &= \quad \mathrm{Var}\Big(\sum_{j=0}^{K} a_j (2\Delta)^{-j} \prod_{k=0}^{j-1}(\hat{q} - \tfrac{k}{n})\Big) \\
&\leq \quad \Big(\sum_{j=0}^{K} |a_j|(2\Delta)^{-j}\Big(\mathrm{Var}(\prod_{k=0}^{j-1}(\hat{q} - \tfrac{k}{n}))\Big)^{\frac{1}{2}}\Big)^2 \\
&\leq \quad \max_{0\leq j'\leq K} |a_{j'}|^2 \Big(\sum_{j=1}^{K}(2\Delta)^{-j}\big(4\Delta q\big)^{\frac{j}{2}}\Big)^2 \qquad\qquad (74) \\
&= \quad \max_{0\leq j'\leq K} |a_{j'}|^2 \Big(\sum_{j=1}^{K}\big(\tfrac{q}{\Delta}\big)^{\frac{j}{2}}\Big)^2 \\
&= \quad \max_{0\leq j'\leq K} |a_{j'}|^2 \tfrac{q}{\Delta} \Big(\sum_{j=0}^{K-1}\big(\tfrac{q}{\Delta}\big)^{\frac{j}{2}}\Big)^2 \\
&\leq \quad \max_{0\leq j'\leq K} |a_{j'}|^2 \tfrac{q}{\Delta} \Big(\sum_{j=0}^{K-1} 2^{\frac{j}{2}}\Big)^2 \\
&\lesssim \quad e^{2\varepsilon} B^K \Delta^2 \tfrac{q}{\Delta} \qquad\qquad\qquad\qquad\qquad\qquad (75) \\
&\lesssim \quad B^K \tfrac{e^\varepsilon c_1 \ln n}{n}(e^\varepsilon q)\;,
\end{aligned}
$$

where $B$ is some universal constant, we use $q \leq 2\Delta$ and $\Delta = (c_1 \ln n)/n$ in the last inequality, and (74) follows from Lemma 11, (75) from Lemma 10. Concretely, it follows from Lemma 11 that for $M_2 = \max\{2n\Delta, K\} = 2n\Delta$,

$$
\mathrm{Var}\Big(\prod_{k=0}^{j-1}(\hat{q} - \tfrac{k}{n})\Big) \;\leq\; \mathbb{E}\Big(\prod_{k=0}^{j-1}(\hat{q} - \tfrac{k}{n})\Big)^2 \;\leq\; \big(\tfrac{2M_2 q}{n}\big)^j \;=\; \big(4\Delta q\big)^j\;.
$$

To apply Lemma 10, we first transform the domain of the polynomial approximation to be symmetric around the origin by change of variables. We consider $H_K(z^2) = \sum_{j=0}^{K} a_j z^{2j}$, which is a polynomial with degree no more than $2K$ and satisfies

$$
\sup_{z\in[-1,1]} |H_K(z^2)| \leq M_1 e^\varepsilon \Delta\;.
$$

This bound follows from a triangular inequality applied to $\max_{y\in[0,1]} |g(y)| \lesssim e^\varepsilon \Delta$ and $\sup_{y\in[0,1]} |H_K(y) - g(y)| \lesssim e^\varepsilon \Delta$. It follows that there exists a universal constant $M_1 > 0$ such that $\sup_{y\in[0,1]} |H_K(y)| \leq M_1 e^\varepsilon \Delta$. It follows from Lemma 10 that for all $0 \leq j \leq K$,

$$
|a_j| \leq M_1 e^\varepsilon \Delta (\sqrt{2} + 1)^{2K}\;. \qquad\qquad (76)
$$

**Case 2:** $p > e^\varepsilon \Delta$ and $q \in [e^{-\varepsilon}p - \sqrt{e^{-\varepsilon}p\Delta}, e^{-\varepsilon}p + \sqrt{e^{-\varepsilon}p\Delta}]$.

First, we analyze the bias. In this regime, we claim that the best polynomial approximation $D_K(x;p)$ of $[p - e^\varepsilon x]^+$ is given by

$$
D_K(x;p) \quad=\quad \frac{e^\varepsilon}{2}\sum_{j=0}^{K} r_j\big(\sqrt{e^{-\varepsilon}p\Delta}\big)^{-j+1}(x - e^{-\varepsilon}p)^j + \frac{p - e^\varepsilon x}{2}\;, \qquad (77)
$$

where $r_j$'s are defined from the best polynomial approximation $R_K(y)$ of $g(y) = |y|$ on $[-1, 1]$ with order $K$: $R_K(y) = \sum_{j=0}^{K} r_j y^j$. And it is well known (e.g. [48, Chapter 9, Theorem 3.3])

that there exists a universal constant $M_3$ such that $|R_K(y) - |y|| \leq M_3/K$, for all $y \in [-1, 1]$. As $[a]^+ = (1/2)a + (1/2)|a|$, the optimality of $D_K(x; q)$ follows from

$$
\begin{aligned}
\left| D_K(x; p) - [p - e^{\varepsilon} x]^+ \right| &= \left| \frac{e^{\varepsilon} \sum_{j=0}^{K} r_j \left( \sqrt{e^{-\varepsilon} p \Delta} \right)^{-j+1} (x - e^{-\varepsilon} p)^j - |p - e^{\varepsilon} x|}{2} \right. \\
&\qquad + \left. \frac{(p - e^{\varepsilon} x) - (p - e^{\varepsilon} x)}{2} \right| \\
&= \frac{e^{\varepsilon}}{2} \left| \sum_{j=0}^{K} r_j \left( \sqrt{e^{-\varepsilon} p \Delta} \right)^{-j+1} (x - e^{-\varepsilon} p)^j - |e^{-\varepsilon} p - x| \right| \\
&= \frac{e^{\varepsilon} \sqrt{e^{-\varepsilon} p \Delta}}{2} \left| R_k \left( \frac{x - e^{-\varepsilon} p}{\sqrt{e^{-\varepsilon} p \Delta}} \right) - \left| \frac{e^{-\varepsilon} p - x}{\sqrt{e^{-\varepsilon} p \Delta}} \right| \right| \\
&= \frac{\sqrt{e^{\varepsilon} p \Delta}}{2} \left| R_k(y) - |y| \right| ,
\end{aligned}
$$

where we let $x = e^{-\varepsilon} p + y \sqrt{e^{-\varepsilon} p \Delta}$, and we want small approximation error in $y \in [-1, 1]$. This gives the desired bound on the bias:

$$
|D_K(x; p) - [p - e^{\varepsilon} x]^+| = \frac{\sqrt{e^{\varepsilon} p \Delta}}{2} \left| R_k(y) - |y| \right| \leq \frac{M_3 \sqrt{e^{\varepsilon} p \Delta}}{K} \lesssim \frac{1}{K} \sqrt{\frac{e^{\varepsilon} p c_1 \ln n}{n}}
$$

Next, we analyze the variance. Recall from Lemma 11 that $g_{j,c}(\hat{q})$ defined as

$$
g_{j,c}(\hat{q}) \triangleq \sum_{k=0}^{j} \binom{j}{k} (-c)^{j-k} \prod_{h=0}^{k-1} \left( \hat{q} - \frac{h}{n} \right) \tag{78}
$$

is the unique uniformly minimum variance unbiased estimator (MVUE) for $(q - c)^j$, $j \geq 0$, $j \in \mathbb{N}$. Hence,

$$
\tilde{D}_K(x; p) = \frac{e^{\varepsilon}}{2} \left( \sum_{j=0}^{K} r_j \left( \sqrt{e^{-\varepsilon} p \Delta} \right)^{-j+1} g_{j,e^{-\varepsilon} p}(\hat{q}) + g_{1,e^{-\varepsilon} p}(\hat{q}) \right) \tag{79}
$$

Let $a_j = r_j$ for $j = 0, 2, 3, \ldots, K$ and $a_1 = r_1 - 1$ and we can write $\tilde{D}_K(x; p)$ as

$$
\tilde{D}_K(x; p) = \frac{e^{\varepsilon}}{2} \sum_{j=0}^{K} a_j \left( \sqrt{e^{-\varepsilon} p \Delta} \right)^{-j+1} g_{j,e^{-\varepsilon} p}(\hat{q}) .
$$

It is shown in [49, Lemma 2] that $|r_j| \leq 2^{3K}$, for $0 \leq j \leq K$. So we can safely say $\max_{0 \leq j \leq K} |a_j|^2 \leq 4 \cdot 2^{6K}$. It follows from Lemma 11 that for $M_4 = \max\{\frac{n(q - e^{-\varepsilon} p)^2}{q}, K\}$,

$$
\mathrm{Var}(g_{j,e^{-\varepsilon} p}(\hat{q})) \leq \mathbb{E} g_{j,e^{-\varepsilon} p}^2(\hat{q}) \leq \left( \frac{2M_4 q}{n} \right)^j .
$$

Note that if $q = 0$, the variance is $0$. We consider the case $q \neq 0$. The variance is

$$
\begin{aligned}
\mathrm{Var}(\tilde{D}_K(x;p)) &= \frac{e^{2\varepsilon}}{4}\mathrm{Var}\Big(\sum_{j=0}^{K} a_j \big(\sqrt{e^{-\varepsilon}p\Delta}\big)^{-j+1} g_{j,e^{-\varepsilon}p}(\hat{q})\Big) \\
&\leq \frac{e^{2\varepsilon}}{4}\Big(\sum_{j=0}^{K} |a_j| \big(\sqrt{e^{-\varepsilon}p\Delta}\big)^{-j+1} \mathrm{Var}^{\frac{1}{2}}\big(g_{j,e^{-\varepsilon}p}(\hat{q})\big)\Big)^2 \\
&\leq e^{2\varepsilon}2^{6K}e^{-\varepsilon}p\Delta\Big(\sum_{j=0}^{K} \big(\sqrt{e^{-\varepsilon}p\Delta}\big)^{-j}\big(\frac{2M_4 q}{n}\big)^{\frac{j}{2}}\Big)^2 \\
&= e^{\varepsilon}2^{6K}p\Delta\Big(\sum_{j=0}^{K} \big(\frac{2M_4 q}{ne^{-\varepsilon}p\Delta}\big)^{\frac{j}{2}}\Big)^2 \\
&\leq e^{\varepsilon}2^{6K}p\Delta\Big(\frac{c^{K+1}-1}{c-1}\Big)^2 && (80) \\
&\leq \frac{c^2}{(c-1)^2}(8c)^{2K}e^{\varepsilon}p\Delta \\
&\lesssim B^K \frac{c_1 \ln n}{n}e^{\varepsilon}p \,,
\end{aligned}
$$

where $c = \max\{\sqrt{2}, 2\sqrt{c_3/c_1}\}$, and $B > 0$ is some universal constant as $c_3 < c_1$. The inequality in (80) follows from $\sqrt{2M_4 q/(ne^{-\varepsilon}p\Delta)} \leq c$, which follows from

$$
\sqrt{\frac{2Kq}{ne^{-\varepsilon}p\Delta}} \leq \sqrt{\frac{2K(e^{-\varepsilon}p + \sqrt{e^{-\varepsilon}p\Delta})}{ne^{-\varepsilon}p\Delta}} \leq \sqrt{\frac{2K \cdot 2e^{-\varepsilon}p}{ne^{-\varepsilon}p\Delta}} = \sqrt{\frac{4c_3}{c_1}} \,, \text{ and}
$$

$$
\sqrt{\frac{2q}{ne^{-\varepsilon}p\Delta} \cdot \frac{n(q-e^{-\varepsilon}p)^2}{q}} = \sqrt{\frac{2(q-e^{-\varepsilon}p)^2}{e^{-\varepsilon}p\Delta}} \leq \sqrt{\frac{2e^{-\varepsilon}p\Delta}{e^{-\varepsilon}p\Delta}} = \sqrt{2} \,.
$$

### C.4  Proof of Theorem 3

In this section we prove a more general statement in the following theorem, in which case Theorem 3 follows as a corollary.

**Theorem 7.** *Suppose $S \geq 2$ and there exists a constant $C > 0$ such that $\ln n \geq C \ln S$. Then for any $P$, there exists a constant $C'$ that only depends on $C$ such that if $\sum_{j=1}^{S} p_j \wedge \sqrt{e^{\varepsilon}p_j/(n \ln n)} \geq C'\left(\sqrt{(e^{\varepsilon}\ln n)/n} + (e^{\varepsilon}\sqrt{S}\ln n)/n\right)$, then*

$$
\inf_{\widehat{d}_\varepsilon(P\|Q_n)} \sup_{Q \in \mathcal{M}_S} \mathbb{E}_Q\big[\,|\,\widehat{d}_\varepsilon(P\|Q_n) - d_\varepsilon(P\|Q)\,|^2\,\big] \gtrsim \Big(\sum_{i=1}^{S} p_i \wedge \sqrt{\frac{e^{\varepsilon}p_i}{n \ln n}}\Big)^2 \,, \qquad (81)
$$

*where the infimum is taken over all possible estimator.*

Note that $d_\varepsilon(P\|Q) = \sum_{i=1}^{S}[p_i - e^{\varepsilon}q_i]^+$ is well defined even if $Q$ does not sum to exactly one. Define a set of such approximate probability vectors as

$$
\mathcal{M}_S(\zeta) = \Big\{Q : \Big|\sum_{i=1}^{S} q_i - 1\Big| \leq \zeta\Big\} \,. \qquad (82)
$$

Later in this section, we use the method of two fuzzy hypotheses from [50] to show that for some $\chi \gtrsim \sum_{j=1}^{S} p_j \wedge \sqrt{e^{\varepsilon}p_j/(n \ln n)}$ and $\chi \leq e^{\varepsilon}$, the estimation error exceeds $\chi/4$ with a strictly positive probability, under a minimax setting over the approximate probability class $\mathcal{M}_S(\zeta)$ with $\zeta = \chi/(10e^{\varepsilon})$:

$$
\inf_{\widehat{d}_\varepsilon(P\|Q_n)} \sup_{Q \in \mathcal{M}_S(\chi/(10e^{\varepsilon}))} P\left(\big|\widehat{d}_\varepsilon(P\|Q_n) - d_\varepsilon(P\|Q)\big| \geq \frac{\chi}{4}\right) \geq \frac{1}{3} \,, \qquad (83)
$$

for a sufficiently large $n$, where we extend the definition of $Q_n$ to be Poisson sampling each alphabet with the appropriate rate. This gives a lower bound on the minimax risk for $\zeta = \chi/(10e^\varepsilon)$:

$$
\begin{aligned}
R(S, n, P, \zeta) \quad &\triangleq \quad \inf_{\widehat{d}_\varepsilon(P\|Q_n)} \sup_{Q \in \mathcal{M}_S(\zeta)} \mathbb{E}_Q\left[ \left( \widehat{d}_\varepsilon(P\|Q_n) - d_\varepsilon(P\|Q) \right)^2 \right] && (84) \\
&\geq \quad \frac{\chi^2}{16} \left\{ \inf_{\widehat{d}_\varepsilon(P\|Q_n)} \sup_{Q \in \mathcal{M}_S(\zeta)} Q\left( \left| \widehat{d}_\varepsilon(P\|Q_n) - d_\varepsilon(P\|Q) \right| \geq \frac{\chi}{4} \right) \right\} \\
&\geq \quad \frac{\chi^2}{48} \;,
\end{aligned}
$$

As our goal is to prove a lower bound on the minimax error, which is $R(S, n, P, 0)$, we use the following lemma. We provide a proof in Appendix C.4.1.

**Lemma 15.** *For any $S, n \in \mathbb{N}_+$, $0 < \zeta < 1$, any distribution $P \in \mathcal{M}_S$, and any $\varepsilon > 0$ that defines the quantity $d_\varepsilon(\cdot\|\cdot)$ used in the definition of $R(\cdot)$ in (84), we have*

$$
R(S, n(1-\zeta)/4, P, 0) \quad \geq \quad \frac{1}{4} R(S, n, P, \zeta) - \frac{1}{2} e^{-n(1-\zeta)/8} - \frac{1}{2} e^{2\varepsilon} \zeta^2 \;. \tag{85}
$$

This implies that for our choice of $\zeta = \chi/(10e^\varepsilon)$,

$$
\begin{aligned}
R(S, n(1-\zeta)/4, P, 0) \quad &\geq \quad \frac{1}{4} R(S, n, P, \zeta) - \frac{1}{2} e^{-n(1-\zeta)/8} - \frac{1}{2} e^{2\varepsilon} \zeta^2 \\
&\geq \quad \frac{\chi^2}{192} - \frac{1}{2} e^{-n(1-\chi/(10e^\varepsilon))/8} - \frac{\chi^2}{200} \\
&\gtrsim \quad \chi^2 \\
&\gtrsim \quad \left( \sum_{j=1}^S p_j \wedge \sqrt{\frac{e^\varepsilon p_j}{n \ln n}} \right)^2 \;,
\end{aligned}
$$

where $\zeta \leq 1/10$, which follows from $\chi \leq e^\varepsilon$, this proves the desired theorem.

Now, we are left to prove Eq. (83), by applying the following Lemma from [50]. The idea is to construct two fuzzy hypotheses, such that they are sufficiently close to each other (as measured by total variation) to be challenging, while sufficiently separated in $d_\varepsilon$. Translating the theorem into our context, we get the following corollary.

**Lemma 16** (Corollary of [50, Theorem 2.15])**.** *For any $s > 0$, $\zeta > 0$, $0 \leq \beta_0, \beta_1 < 1$, $\lambda \in \mathbb{R}$, if there exists two distributions $\sigma_0$ and $\sigma_1$ on $Q = [q_1, \ldots, q_S] \in \mathcal{M}_S(\zeta)$ such that*

$$
\begin{aligned}
\sigma_0(Q : d_\varepsilon(P\|Q) \leq \lambda - s) \quad &\geq \quad 1 - \beta_0 \;, && (86) \\
\sigma_1(Q : d_\varepsilon(P\|Q) \geq \lambda + s) \quad &\geq \quad 1 - \beta_1 \;, && (87)
\end{aligned}
$$

*and $D_{\mathrm{TV}}(F_1, F_0) \leq \eta < 1$, then*

$$
\inf_{\widehat{d}_\varepsilon(P\|Q_n)} \sup_{Q \in \mathcal{M}_S(\zeta)} \mathbb{P}_Q\left( \left| \widehat{d}_\varepsilon(P\|Q_n) - d_\varepsilon(P\|Q) \right| \geq s \right) \quad \geq \quad \frac{1 - \eta - \beta_0 - \beta_1}{2} \;, \tag{88}
$$

*where $F_i$ is the marginal distribution of $Q_n$ given the prior $\sigma_i$ for $i \in \{0, 1\}$.*

We construct two hypotheses, satisfying the assumptions with choices of $s = \chi/4 \gtrsim \sum_{j=1}^S p_j \wedge \sqrt{e^\varepsilon p_j/(n \ln n)}$ and $\eta, \beta_0, \beta_1 = o(1)$ such that Eq. (83) follows. We will first introduce the construction, check the separation conditions in Eqs. (86) and (87), and check the total variation condition.

**Constructing two prior distributions.** Fix the distribution $P \in \mathcal{M}_S$, and assume $p_S = \min_{1 \leq i \leq S} p_i$. Let $\boldsymbol{\mu}_0$, $\boldsymbol{\mu}_1$ be two prior distributions on the parameter $Q$ where $Q_n$ will be drawn from, and set

$$
\begin{aligned}
\boldsymbol{\mu}_0 \quad &= \quad \mu_0^{(p_1)} \otimes \mu_0^{(p_2)} \otimes \ldots \otimes \mu_0^{(p_{S-1})} \otimes \delta_{1-\gamma} \;, && (89) \\
\boldsymbol{\mu}_1 \quad &= \quad \mu_1^{(p_1)} \otimes \mu_1^{(p_2)} \otimes \ldots \otimes \mu_1^{(p_{S-1})} \otimes \delta_{1-\gamma} \;, && (90)
\end{aligned}
$$

where

$$\gamma = \sum_{j:p_j \leq \frac{ce^\varepsilon \ln n}{n}} \frac{p_j}{e^\varepsilon D} + \sum_{j:p_j > \frac{ce^\varepsilon \ln n}{n}} e^{-\varepsilon}p_j \,, \tag{91}$$

and $c \in (0,1)$ is a constant, $D$ is the universal constant in Lemma 20. Note that this does not produce a valid probability distribution, as it will not sum to one almost surely. However, this is sufficient as we can bound the difference in the minimax rate between exact and approximate probability distributions using Lemma 15. This choice of $\gamma$ ensures that the sum $\sum_{j=1}^{S} q_j$ concentrates around one. For a $p \in (0,1)$, we construct $\mu_i^{(p)}$, $i \in \{0,1\}$ depending on $p$ in two separate cases. Our goal is to construct two prior distributions, which match in the first $L$ degree moments (such that the marginal total variation distance is sufficiently small), but at the same time sufficiently different in estimation of $d_\varepsilon$, such that they differ approximately as much as the resolution of the best polynomial function approximation.

**Case 1:** $p > (ce^\varepsilon \ln n)/n$**, for some constant** $c \in (0,1)$**.** Define function $g(x) = e^{-\varepsilon}p + (\sqrt{(ce^{-\varepsilon}p \ln n)/n})x$, where $x \in [-1,1]$. Let $\nu_i, i = 0,1$ be two measures constructed in Lemma 17.

**Lemma 17** ( [49, Lemma 1])**.** *For any positive integer $L > 0$, there exists two probability measure $\nu_0$ and $\nu_1$ on $[-1,1]$ such that*

1. $\int t^l \nu_1(dt) = \int t^l \nu_0(dt)$, *for all $l = 0,1,2,\ldots,L$;*

2. $\int [-t]^+ \nu_1(dt) - \int [-t]^+ \nu_0(dt) = \Delta_L[[-t]^+; [-1,1]]$,

*where $\Delta_L[[-t]^+; [-1,1]]$ is the distance in the uniform norm on $[-1,1]$ from the function $[-t]^+$ to the space of polynomial functions of degree $L$: $\mathrm{poly}_L$.*

We define two new measures $\mu_i^{(p)}, i = 0,1$ on $[e^{-\varepsilon}p - \sqrt{\frac{ce^{-\varepsilon}p \ln n}{n}}, e^{-\varepsilon}p + \sqrt{\frac{ce^{-\varepsilon}p \ln n}{n}}]$ by $\mu_i^{(p)}(A) = \nu_i(g^{-1}(A))$. Note that we need the lower bound on $p$ to ensure that this is non-negative. Let $L = d_2 \ln n, d_2 > 1$. It follows that

1.

$$\int t\mu_1^{(p)}(dt) = \int t\mu_0^{(p)}(dt) = e^{-\varepsilon}p \,; \tag{92}$$

2.

$$\int t^l \mu_1^{(p)}(dt) = \int t^l \mu_0^{(p)}(dt), \quad \forall l = 2\ldots, L+1; \tag{93}$$

3.

$$\int [p - e^\varepsilon t]^+ \mu_1^{(p)}(dt) - \int [p - e^\varepsilon t]^+ \mu_0^{(p)}(dt) = \sqrt{\frac{ce^\varepsilon p \ln n}{n}} \Delta_L[[-t]^+; [-1,1]]$$

$$\gtrsim p \wedge \sqrt{\frac{ce^\varepsilon p}{d_2^2 n \ln n}} \,. \tag{94}$$

The last inequality follows from the following lemma, with a choice of $L = d_2 \ln n$ for some constant $d_2$.

**Lemma 18** ([51] )**.** *For $L > 1$,*

$$\Delta_L[[-t]^+; [-1,1]] = \Delta_L[|t|; [-1,1]] = \beta_* L^{-1}(1 + o(1)) \asymp \frac{1}{L} \,, \tag{95}$$

*where $\beta_* \approx 0.2802$ is the Bernstein constant.*

**Case 2:** $0 < p \leq (ce^\varepsilon \ln n)/n$**, for some constant** $c \in (0,1)$**.** When $p$ is too close to zero, directly applying the above strategy only gives a lower bound on the difference in $d_\varepsilon$ under the two

hypotheses that scales only as $p/\ln n$, and not as $\sqrt{p/(n \ln n)}$ as desired. Instead, we construct an approximation of $f(x; a) = ([a - e^\varepsilon x]^+ - a)/(e^\varepsilon x)$.

Our strategy is to first construct two prior distributions $\tilde{\nu}_i^{\eta,a}$'s on $\{0\} \cup [\eta, 1]$ which are difference in estimating $f(x; a) = ([a - e^\varepsilon x]^+ - a)/(e^\varepsilon x)$ (instead of $[a - e^\varepsilon x]^+$). The non-smoothness of $f(x; a)$ near zero allows one to control the hardness of this estimation by choosing $\eta$, while ensuring the non-negativity of the resulting random variable $p$ drawn from $\mu_i$'s and also the expectation is close to $q$. Concretely, we let $\mu_i^{(p)}$ to be a measure on $\{0\} \cup [p/(De^\varepsilon), M]$, where $g(x) = Mx$ and $\mu_i^{(q)} = \tilde{\nu}_0^{\eta,a}(g^{-1}(A))$. We first construct two new probability measures $\tilde{\nu}_i^{\eta,a}$, $i = 0, 1$ from the two probability measures $\nu_i^{\eta,a}$, $i = 0, 1$ constructed in Lemma 19.

**Lemma 19.** *Let* $f(x; a) \triangleq ([a - e^\varepsilon x]^+ - a)/(e^\varepsilon x)$, $a \in [0, 1]$. *For any* $0 < \eta < 1$ *and positive integer* $L > 0$, *there exists two probability measure* $\nu_0^{\eta,a}$ *and* $\nu_1^{\eta,a}$ *on* $[\eta, 1]$ *such that*

1. $\int t^l \nu_1^{\eta,a}(dt) = \int t^l \nu_0^{\eta,a}(dt)$, *for all* $l = 0, 1, 2, \ldots, L$;

2. $\int f(t; a)\nu_1^{\eta,a}(dt) - \int f(t; a)\nu_0^{\eta,a}(dt) = \Delta_L[f(x; a); [\eta, 1]]$,

*where* $\Delta_L[f(x; a); [\eta, 1]]$ *is the distance in the uniform norm on* $[\eta, 1]$ *from the function* $f(x; a)$ *to the space of polynomial functions of degree* $L$: $\text{poly}_L$.

We construct $\tilde{\nu}_i^{\eta,a}$ by scaling down $\nu_i^{\eta,a}$ and putting the remaining probability mass on zero. This ensures that the restriction on $[\eta, 1]$ of $\tilde{\nu}_i^{\eta,a}$ is absolutely continuous with $\nu_i^{\eta,a}$, and we construct the Radon-Nikodym derivative to be

$$\frac{d\tilde{\nu}_i^{\eta,a}}{dnu_i^{\eta,a}} = \frac{\eta}{t} \le 1, \quad t \in [\eta, 1] , \tag{96}$$

and $\tilde{\nu}_i^{\eta,a}(\{0\}) = 1 - \tilde{\nu}_i^{\eta,a}([\eta, 1]) \ge 0$. This choice of scaling ensures that

It follows that $\tilde{\nu}_i^{\eta,a}$, $i = 0, 1$ are probability measures on $[0, 1]$ that satisfy the following properties

1. $\int t \tilde{\nu}_1^{\eta,a}(dt) = \int t \tilde{\nu}_0^{\eta,a}(dt) = \eta$;
2. $\int t^l \tilde{\nu}_1^{\eta,a}(dt) = \int t^l \tilde{\nu}_0^{\eta,a}(dt)$, *for all* $l = 2, \ldots, L + 1$;
3. $\int [a - e^\varepsilon t]^+ \tilde{\nu}_1^{\eta,a}(dt) - \int [a - e^\varepsilon t]^+ \tilde{\nu}_0^{\eta,a}(dt) = \eta e^\varepsilon \Delta_L[f(x; a); [\eta, 1]]$.

Define

$$L = d_2 \ln n, \quad \eta = \frac{a}{D}, \quad a = \frac{p}{e^\varepsilon M}, \quad M = \frac{2c \ln n}{n} , \tag{97}$$

where $D$ is a universal constant in Lemma 20 and $d_2 > 1$ is a constant.

**Lemma 20.** *Let* $f(x; a) := \frac{[a - e^\varepsilon x]^+ - a}{e^\varepsilon x}$, $a \in (0, \frac{1}{2}]$, $x \in [0, 1]$, *there exists universal constant* $D$ *such that*

$$\Delta_L \left[ f(x; a); [\frac{a}{D}, 1] \right] \gtrsim \left( 1 \wedge \frac{1}{L\sqrt{a}} \right) . \tag{98}$$

Let $g(x) = Mx$ and let $\mu_i^{(p)}$ be the measure on $[0, M]$ defined by $\mu_i^{(p)} = \tilde{\nu}_0^{\eta,a}(g^{-1}(A))$. Then we have $\mu_i^{(p)}(A) = M\tilde{\nu}_0^{\eta,a}(A)$. It then follows that

1.

$$\int t\mu_1^{(p)}(dt) = \int t\mu_0^{(p)}(dt) = \frac{p}{e^\varepsilon D} ; \tag{99}$$

2.

$$\int t^l \mu_1^{(p)}(dt) = \int t^l \mu_0^{(p)}(dt), \quad \forall l = 2 \ldots, L + 1; \tag{100}$$

3.

$$\int [p - e^\varepsilon t]^+ \mu_1^{(p)}(dt) - \int [p - e^\varepsilon t]^+ \mu_0^{(p)}(dt) \quad = \quad \eta\, e^\varepsilon\, M\, \Delta_L[f(x;a); [\tfrac{a}{D}, 1]] \quad (101)$$

$$\gtrsim \quad p \wedge \sqrt{\frac{ce^\varepsilon p}{d_2^2 n \ln n}} \;. \qquad (102)$$

**Separation conditions.** In both cases, since we set $q_S = 1 - \gamma$, which is defined in Eq. (91), it follows from Eq. (99) and (92) that

$$\mathbb{E}_{\boldsymbol{\mu}_0}\left[\sum_{j=1}^n q_j\right] \quad = \quad \mathbb{E}_{\boldsymbol{\mu}_1}\left[\sum_{j=1}^n q_j\right] \quad = \quad 1 \;. \qquad (103)$$

Let

$$\chi \quad = \quad \mathbb{E}_{\boldsymbol{\mu}_1}[d_\varepsilon(P\|Q)] - \mathbb{E}_{\boldsymbol{\mu}_0}[d_\varepsilon(P\|Q)] \;, \qquad (104)$$

$$\zeta \quad = \quad \frac{\chi}{10 e^\varepsilon} \;. \qquad (105)$$

We know from Eq. (102) and (94) that, by construction, the estimates are separated in expectation:

$$\chi \quad \gtrsim \quad \sum_{j=1}^{S-1} p_j \wedge \sqrt{\frac{ce^\varepsilon p_j}{d_2^2 n \ln n}}$$

$$\geq \quad (1 - \frac{1}{S}) \sum_{j=1}^{S} p_j \wedge \sqrt{\frac{ce^\varepsilon p_j}{d_2^2 n \ln n}}$$

$$\gtrsim \quad \sum_{j=1}^{S} p_j \wedge \sqrt{\frac{ce^\varepsilon p_j}{d_2^2 n \ln n}} \;,$$

where the second inequality follows from the assumption that $p_S = \min_{1 \leq j \leq S} p_j$. To show concentration of $\mathbb{E}_{\boldsymbol{\mu}_i}[d_\varepsilon(P\|Q)]$ around its mean, for $i = 0, 1$, we introduce the events

$$E_i \quad = \quad \mathcal{M}_S(\zeta) \cap \left\{ Q : \left| d_\varepsilon(P\|Q) - \mathbb{E}_{\boldsymbol{\mu}_i}[d_\varepsilon(P\|Q)] \right| \leq \frac{\chi}{4} \right\} \;. \qquad (106)$$

Introduce

$$F(P) \quad = \quad \sum_{j: p_j \leq \frac{ce^\varepsilon \ln n}{n}} \left(\frac{2c\ln n}{n}\right)^2 + \sum_{j: p_j > \frac{ce^\varepsilon \ln n}{n}} \left(\frac{4ce^{-\varepsilon} p_j \ln n}{n}\right)$$

$$\leq \quad \frac{4c^2 S \ln^2 n}{n^2} + \frac{4ce^{-\varepsilon} \ln n}{n} \;.$$

It follows from the union bound and Hoeffding bound that

$$\boldsymbol{\mu}_i(E_i^c) \quad \leq \quad \boldsymbol{\mu}_i\left( \left| \sum_{j=1}^S q_j - 1 \right| > \zeta \right) + \boldsymbol{\mu}_i\left( \left| d_\varepsilon(P\|Q) - \mathbb{E}_{\boldsymbol{\mu}_i}[d_\varepsilon(P\|Q)] \right| > \frac{\chi}{4} \right)$$

$$\leq \quad 2\exp\left(-\frac{2\zeta^2}{F(P)}\right) + 2\exp\left(-\frac{\chi^2}{8F(P)}\right) \;.$$

Then we choose parameter $c$ such that $\boldsymbol{\mu}_i(E_i^c)$ can be made arbitrarily small. Since we assumed $c \in (0, 1)$ and $d_2 > 1$, and from the assumption that $\sum_{j=1}^{S} p_j \wedge \sqrt{\frac{e^\varepsilon p_j}{n \ln n}} \geq C'\left( \sqrt{\frac{e^\varepsilon \ln n}{n}} + \frac{e^\varepsilon \sqrt{S}\ln n}{n} \right)$,

we have

$$\zeta \asymp \frac{\chi}{e^\varepsilon}$$

$$\gtrsim \frac{1}{e^\varepsilon} \sum_{j=1}^{S} p_j \wedge \sqrt{\frac{ce^\varepsilon p_j}{d_2^2 n \ln n}}$$

$$\geq \frac{\sqrt{c}}{e^\varepsilon d_2} \sum_{j=1}^{S} p_j \wedge \sqrt{\frac{e^\varepsilon p_j}{n \ln n}}$$

$$\geq \frac{\sqrt{c}}{d_2} C' \left( \sqrt{\frac{e^{-\varepsilon} \ln n}{n}} + \frac{\sqrt{S} \ln n}{n} \right)$$

$$\gtrsim \frac{1}{d_2} C' \left( \sqrt{\frac{ce^{-\varepsilon} \ln n}{n}} + \frac{c\sqrt{S} \ln n}{n} \right)$$

$$\gtrsim \frac{1}{d_2} C' \sqrt{F(P)} .$$

Hence, it suffices to take $C'$ large enough to ensure $\boldsymbol{\mu}_i(E_i^c)$, is as small as we desire for $i = 0, 1$. So with this constant $C'$, we have

$$\mu_0 \left( d_\varepsilon(P\|Q) \leq \mathbb{E}_{\boldsymbol{\mu}_0}\left[ d_\varepsilon(P\|Q) \right] + \frac{\chi}{4} \right) \quad \geq \quad 1 - \beta_0 , \tag{107}$$

$$\mu_1 \left( d_\varepsilon(P\|Q) \geq \mathbb{E}_{\boldsymbol{\mu}_1}\left[ d_\varepsilon(P\|Q) \right] - \frac{\chi}{4} \right) \quad \geq \quad 1 - \beta_1 , \tag{108}$$

fo any constant $\beta_0$ and $\beta_1$, which satisfy the conditions of Lemma 16 and $s = \chi/4$.

**Total variation condition.** Let $G_i$ be marginal distribution of $(X_1, X_2, \ldots, X_S)$ under priors $\boldsymbol{\mu}_i$ for $i = 0, 1$. Denote by $\pi_i$ the probability measures defined as

$$\pi_i(A) = \frac{\boldsymbol{\mu}_i\left( E_i \cap A \right)}{\boldsymbol{\mu}_i\left( E_i \right)}, \quad i = 0, 1 . \tag{109}$$

Let $F_i$ be marginal distribution of $(X_1, X_2, \ldots, X_S)$ under priors $\pi_i$ for $i = 0, 1$.

Triangle inequality of total variation yields

$$\mathrm{TV}(F_0, F_1) \leq \mathrm{TV}(F_0, G_0) + \mathrm{TV}(G_0, G_1) + \mathrm{TV}(G_1, F_1)$$

$$= \sup_A \left| \boldsymbol{\mu}_0(A) - \frac{\boldsymbol{\mu}_0(A \cap E_0)}{\boldsymbol{\mu}_0(E_0)} \right| + \mathrm{TV}(G_0, G_1) + \sup_A \left| \boldsymbol{\mu}_1(A) - \frac{\boldsymbol{\mu}_1(A \cap E_1)}{\boldsymbol{\mu}_1(E_1)} \right|$$

$$= \mathrm{TV}(G_0, G_1) + \boldsymbol{\mu}_0(E_0^c) + \boldsymbol{\mu}_1(E_1^c) .$$

In view of fact that $\mathrm{TV}\left( \otimes_{i=1}^{S} P_i, \otimes_{i=1}^{S} Q_i \right) \leq \sum_{i=1}^{S} \mathrm{TV}\left( P_i, Q_i \right)$, we have

$$\mathrm{TV}(G_0, G_1) \leq \sum_{i=1}^{S-1} \mathrm{TV}(\mu_0^{(p_i)}, \mu_1^{(p_i)})$$

$$\leq \sum_{i=1}^{S-1} 2 \left( \frac{1}{2} \right)^{d_2 \ln n}$$

$$\leq \frac{2S}{n^{d_2 \ln 2}} ,$$

where in the second inequality we applied the following lemmas, assuming $d_2$ to be large enough.

**Lemma 21** ([16, Lemma 3] when $q_i \leq ce^\varepsilon \ln n/n$)**.** *Suppose $U_0$, $U_1$ are two random variables supported on $[0, M]$, where $M \geq 0$ is constant. Suppose $\mathbb{E}[U_0^j] = \mathbb{E}[U_1^j]$, $0 \leq j \leq L$. Denote the marginal distribution of $X$ where $X|\lambda \sim \mathrm{Poi}\,(\lambda)$, $\lambda \sim U_i$ as $F_i$ for $i = 0, 1$. If $L > 2eM$, then*

$$\mathrm{TV}(F_0, F_1) \leq \left( \frac{2eM}{L} \right)^L . \tag{110}$$

**Lemma 22** ([21, Lemma 32] when $q_i > ce^\varepsilon \ln n/n$). *Suppose $U_0$, $U_1$ are two random variables supported on $[a - M, a + M]$, where $a \geq M \geq 0$ are constants. Suppose $\mathbb{E}[U_0^j] = \mathbb{E}[U_1^j]$, $0 \leq j \leq L$. Denote the marginal distribution of $X$ where $X|\lambda \sim \mathrm{Poi}\,(\lambda)$, $\lambda \sim U_i$ as $F_i$ for $i = 0, 1$. If $L + 1 \geq (2eM)^2/a$, then*

$$\mathrm{TV}(F_0, F_1) \leq 2 \left( \frac{eM}{\sqrt{a(L+1)}} \right)^{L+1} . \tag{111}$$

Since there exists a constant $C > 0$ such that $\ln n \geq C \ln S$, $S \geq 2$, we can conclude that with chosen parameters, $T(F_0, F_1) = o(1)$.

### C.4.1   Proof of Lemma 15

We define minimax risk under the multinomial sampling model for a fixed $P$ as

$$R_B(S, n, P) \triangleq \inf_{\widehat{d}_\varepsilon(P\|Q_n)} \sup_{Q \in \mathcal{M}_S} \mathbb{E}_Q \left[ \left( \widehat{d}_\varepsilon(P\|Q_n) - d_\varepsilon(P\|Q) \right)^2 \right] . \tag{112}$$

Let $\hat{T} = \hat{T}(X_1, X_2, \ldots, X_S)$ be a near-minimax estimator under multinomial model such that for every sample size $n$,

$$\sup_{Q \in \mathcal{M}_S} \mathbb{E}_Q \left[ \left( \hat{T} - d_\varepsilon(P\|Q) \right)^2 \right] < R_B(S, n, P) + \xi , \tag{113}$$

where $\xi > 0$.

For any $Q \in \mathcal{M}_S(\zeta)$, let $\sum_{i=1}^S q_i = A$, we have

$$
\begin{aligned}
\left| d_\varepsilon \left( P \| \frac{Q}{\sum_{i=1}^S q_i} \right) - d_\varepsilon \left( P \| Q \right) \right| &\leq \sum_{i=1}^S \left| [p_i - e^\varepsilon q_i/A]^+ - [p_i - e^\varepsilon q_i]^+ \right| \\
&\leq \sum_{i=1}^S e^\varepsilon q_i |1/A - 1| \\
&= e^\varepsilon |1 - A| \\
&\leq e^\varepsilon \zeta .
\end{aligned}
$$

Now we consider risk of $\hat{T}$ for $Q \in \mathcal{M}_S(\zeta)$ under Poisson sampling model, where $X_i$ are mutually independent with marginal distributions $X_i \sim \mathrm{Poi}\,(nq_i)$. Let $n' = \sum_{i=1}^S X_i$, we know $n' \sim \mathrm{Poi}\,(n \sum_{i=1}^S q_i)$. In view of fact that conditioned on $n' = m$, $(X_1, X_2, \ldots, X_S)$ follows

multinomial distribution parameterized by $\left(m, \frac{Q}{\sum_{i=1}^{S} q_i}\right)$, we have

$$\mathbb{E}_Q\left[\left(\hat{T} - d_\varepsilon(P\|Q)\right)^2\right] \tag{114}$$

$$= \mathbb{E}_Q\left[\left(\hat{T} - d_\varepsilon\left(P\|\frac{Q}{\sum_{i=1}^{S} q_i}\right) + d_\varepsilon\left(P\|\frac{Q}{\sum_{i=1}^{S} q_i}\right) - d_\varepsilon(P\|Q)\right)^2\right] \tag{115}$$

$$\leq 2\mathbb{E}_Q\left[\left(\hat{T} - d_\varepsilon\left(P\|\frac{Q}{\sum_{i=1}^{S} q_i}\right)\right)^2\right] + 2\mathbb{E}_Q\left[\left(d_\varepsilon\left(P\|\frac{Q}{\sum_{i=1}^{S} q_i}\right) - d_\varepsilon(P\|Q)\right)^2\right] \tag{116}$$

$$\leq 2\mathbb{E}_Q\left[\left(\hat{T} - d_\varepsilon\left(P\|\frac{Q}{\sum_{i=1}^{S} q_i}\right)\right)^2\right] + 2e^{2\varepsilon}\zeta^2 \tag{117}$$

$$= 2\sum_{m=0}^{\infty} \mathbb{E}_Q\left[\left(\hat{T} - d_\varepsilon\left(P\|\frac{Q}{\sum_{i=1}^{S} q_i}\right)\right)^2\bigg| n' = m\right]\mathbb{P}(n' = m) + 2e^{2\varepsilon}\zeta^2 \tag{118}$$

$$\leq 2\sum_{m=0}^{\infty} R_B(S, m, P)\mathbb{P}(n' = m) + 2(e^{2\varepsilon}\zeta^2 + \xi) \tag{119}$$

$$\leq 2\left(1 \cdot \mathbb{P}\left(n' \leq n(1-\zeta)/2\right) + R_B\left(S, n(1-\zeta)/2, P\right)\mathbb{P}\left(n' \geq n(1-\zeta)/2\right)\right)$$
$$+ 2(e^{2\varepsilon}\zeta^2 + \xi) \tag{120}$$

$$\leq 2R_B\left(S, n(1-\zeta)/2, P\right) + 2\mathbb{P}\left(n' \leq n(1-\zeta)/2\right) + 2(e^{2\varepsilon}\zeta^2 + \xi) \tag{121}$$

$$\leq 2R_B\left(S, n(1-\zeta)/2, P\right) + 2\mathbb{P}\left(\text{Poi}\left(n(1-\zeta)/2\right) \leq n(1-\zeta)/2\right) + 2(e^{2\varepsilon}\zeta^2 + \xi) \tag{122}$$

$$\leq 2R_B\left(S, n(1-\zeta)/2, P\right) + 2e^{-n(1-\zeta)/8} + 2(e^{2\varepsilon}\zeta^2 + \xi) , \tag{123}$$

where (120) follows from $R_B(S, m, P) \leq 1$, and the last inequality follows from Lemma 1. Taking the supremum of $\mathbb{E}_Q\left[\left(\hat{T} - d_\varepsilon(P\|Q)\right)^2\right]$ over $\mathcal{M}_S(\zeta)$ and using the arbitrariness of $\zeta$, we have

$$R(S, n, P, \zeta) \leq 2R_B\left(S, n(1-\zeta)/2, P\right) + 2e^{-n(1-\zeta)/8} + 2e^{2\varepsilon}\zeta^2 , \tag{124}$$

which is equivalent to

$$R_B\left(S, n(1-\zeta)/2, P\right) \geq \frac{1}{2}R(S, n, Q, \zeta) - e^{-n(1-\zeta)/8} - e^{2\varepsilon}\zeta^2 . \tag{125}$$

It follows from [15, Lemma 16] that $R_B\left(S, n, P\right) \leq 2R\left(S, n/2, P, 0\right)$. Hence,

$$R(S, n(1-\zeta)/4, P, 0) \geq \frac{1}{2}R_B(S, n(1-\zeta)/2, Q) \tag{126}$$

$$\geq \frac{1}{4}R(S, n, P, \zeta) - \frac{1}{2}e^{-n(1-\zeta)/8} - \frac{1}{2}e^{2\varepsilon}\zeta^2 . \tag{127}$$

### C.4.2 Proof of Lemma 19

By [21, Lemma 31], there are two probability measures $\nu_1^{\eta,a}$ and $\nu_0^{\eta,a}$ on $[\eta, 1]$ such that

1.
$$\int t^l \nu_1^{\eta,a}(dt) = \int t^l \nu_0^{\eta,a}(dt), \text{ for all } l = 0, 1, 2, \ldots, L .$$

2.
$$\int \frac{|e^\varepsilon x - a| - a}{e^\varepsilon x}\nu_1^{\eta,a}(dt) - \int \frac{|e^\varepsilon x - a| - a}{e^\varepsilon x}\nu_0^{\eta,a}(dt) = 2\Delta_L\left[\frac{|e^\varepsilon x - a| - a}{e^\varepsilon x}; [\eta, 1]\right] ,$$

which is equivalent to

$$\int \frac{2[a - e^\varepsilon x]^+ - 2a}{e^\varepsilon x}\nu_1^{\eta,a}(dt) - \int \frac{2[a - e^\varepsilon x]^+ - 2a}{e^\varepsilon x}\nu_0^{\eta,a}(dt) = 2\Delta_L\left[\frac{[a - e^\varepsilon x]^+ - a}{e^\varepsilon x}; [\eta, 1]\right] .$$

The two desired measures are constructed.

### C.4.3 Proof of Lemma 20

Let $g(x;a) = \frac{[a-x]^+ - a}{x}$. We have

$$g(x;a) \quad = \quad \frac{|x-a|-a}{2x} - \frac{1}{2} . \tag{128}$$

By the definition of best polynomial approximation error, for $L > 1$, we have

$$
\begin{aligned}
\Delta_L \left[ g(x;a); [\frac{a}{D}, 1] \right] &= \inf_{h(x) \in \mathrm{Poly}_L} \sup_{x \in [\frac{a}{D}, 1]} \left| \frac{|x-a|-a}{2x} - \frac{1}{2} - h(x) \right| \\
&= \inf_{h(x) \in \mathrm{Poly}_L} \sup_{x \in [\frac{a}{D}, 1]} \left| \frac{|x-a|-a}{x} - h(x) \right| \\
&= \Delta_L \left[ \frac{|x-a|-a}{x}; [\frac{a}{D}, 1] \right] \\
&\gtrsim \begin{cases} \frac{1}{L\sqrt{a}} & \frac{1}{L^2} \le a \le \frac{1}{2} \\ 1 & 0 < a < \frac{1}{L^2} \end{cases} ,
\end{aligned}
$$

where $D$ is from [21, Lemma 30].

Now we consider $f(x;a) = \frac{[e^{-\varepsilon}a - x]^+ - e^{-\varepsilon}a}{x}$, where $a \in (0, 1/2]$. As $e^{-\varepsilon}a \in (0, \frac{1}{2}]$. there exists $D > 0$ such that

$$\Delta_L \left[ f(x;a); [\frac{a}{D}, 1] \right] \ge \Delta_L \left[ f(x;a); [\frac{e^{-\varepsilon}a}{D}, 1] \right] \gtrsim \left( 1 \wedge \frac{1}{L\sqrt{e^{-\varepsilon}a}} \right) \ge \left( 1 \wedge \frac{1}{L\sqrt{a}} \right) \tag{129}$$

### C.5 Proof of Theorem 4

Define good events, where our choice of regimes are correct as

$$E_1 = \left\{ \left\{ i : \hat{p}_{i,1} - e^{\varepsilon}\hat{q}_{i,1} > \sqrt{\frac{(c_1+c_2)\ln n}{n}} (\sqrt{\hat{p}_{i,1}} + \sqrt{e^{\varepsilon}\hat{q}_{i,1}}) \right\} \subseteq \left\{ i : e^{\varepsilon}q_i \le p_i \right\} \right\} , \tag{130}$$

$$E_2 = \left\{ \left\{ i : \hat{p}_{i,1} - e^{\varepsilon}\hat{q}_{i,1} < -\sqrt{\frac{(c_1+c_2)\ln n}{n}} (\sqrt{\hat{p}_{i,1}} + \sqrt{e^{\varepsilon}\hat{q}_{i,1}}) \right\} \subseteq \left\{ i : e^{\varepsilon}q_i \ge p_i \right\} \right\} , \tag{131}$$

$$E_3 = \left\{ \left\{ i : e^{\varepsilon}\hat{q}_{i,1} + \hat{p}_{i,1} < \frac{c_1\ln n}{n} \right\} \subseteq \left\{ i : (p_i, e^{\varepsilon}q_i) \in [0, \frac{2c_1\ln n}{n}]^2 \right\} \right\} , \tag{132}$$

and

$$E_4 = \left\{ \left\{ i : (\hat{p}_{i,1}, e^{\varepsilon}\hat{q}_{i,1}) \in U(c_1, c_1), \hat{p}_{i,1} + e^{\varepsilon}\hat{q}_{i,1} \ge \frac{c_1\ln n}{n} \right\} \right.$$
$$\left. \subseteq \left\{ i : (p_i, e^{\varepsilon}q_i) \in U(c_1, c_1), p_i + e^{\varepsilon}q_i \ge \frac{c_1\ln n}{2n}, \hat{p}_{i,1} + e^{\varepsilon}\hat{q}_{i,1} \ge \frac{p_i + e^{\varepsilon}q_i}{2} \right\} \right\} . \tag{133}$$

Denote the overall good event as

$$E = E_1 \cap E_2 \cap E_3 \cap E_4 . \tag{134}$$

Decompose the error under good events as

$$\mathcal{E}_1 \triangleq \sum_{i \in I_1} \left\{ \hat{p}_{i,2} - e^{\varepsilon}\hat{q}_{i,2} - [p_i - e^{\varepsilon}q_i]^+ \right\} , \tag{135}$$

$$\mathcal{E}_2 \triangleq \sum_{i \in I_2} \left\{ \tilde{D}_K^{(1)}(\hat{p}_{i,2}, \hat{q}_{i,2}) - [p_i - e^{\varepsilon}q_i]^+ \right\} , \tag{136}$$

$$\mathcal{E}_3 \triangleq \sum_{i \in I_3} \left\{ \tilde{D}_K^{(2)}(\hat{p}_{i,2}, \hat{q}_{i,2}; \hat{p}_{i,1}, \hat{q}_{i,1}) - [p_i - e^{\varepsilon}q_i]^+ \right\} , \tag{137}$$

where the indices of those regimes under the good events are

$$I_1 \triangleq \left\{ i : \hat{p}_{i,1} - e^\varepsilon \hat{q}_{i,1} > \sqrt{\frac{(c_1 + c_2)\ln n}{n}} (\sqrt{\hat{p}_{i,1}} + \sqrt{e^\varepsilon \hat{q}_{i,1}}), e^\varepsilon q_i \leq p_i \right\} , \qquad (138)$$

$$I_2 \triangleq \left\{ i : \hat{p}_{i,1} + e^\varepsilon \hat{q}_{i,1} \leq \frac{c_1 e^\varepsilon \ln n}{n}, (p_i, e^\varepsilon q_i) \in \left[0, \frac{2c_1 \ln n}{n}\right]^2 \right\} , \qquad (139)$$

$$I_3 \triangleq \left\{ i : \hat{p}_{i,1} + e^\varepsilon \hat{q}_{i,1} \geq \frac{c_1 \ln n}{n}, (\hat{p}_{i,1}, e^\varepsilon \hat{q}_{i,1}) \in U(c_1, c_2), \right.$$

$$\left. (p_i, e^\varepsilon q_i) \in U(c_1, c_1), p_i + e^\varepsilon q_i \geq \frac{c_1 \ln n}{2n}, \hat{p}_{i,1} + e^\varepsilon \hat{q}_{i,1} \geq \frac{p_i + e^\varepsilon q_i}{2} \right\} . \qquad (140)$$

We can bound the squared error as

$$\mathbb{E}\left[ \left( \hat{d}_{\varepsilon,K,c_1,c_2}(P_n \| Q_n) - d_\varepsilon(P \| Q) \right)^2 \right] \leq \mathbb{E}\left[ \left( \hat{d}_{\varepsilon,K,c_1,c_2}(P_n \| Q_n) - d_\varepsilon(P \| Q) \right)^2 \mathbb{I}(E) \right] + \mathbb{P}(E^c)$$

$$\leq \mathbb{E}[(\mathcal{E}_1 + \mathcal{E}_2 + \mathcal{E}_3)^2] + \mathbb{P}(E^c)$$

$$\leq 3\mathbb{E}[\mathcal{E}_1^2] + 3\mathbb{E}[\mathcal{E}_2^2] + 3\mathbb{E}[\mathcal{E}_3^2] + \mathbb{P}(E^c) . \qquad (141)$$

The last term on the bad event is bounded by $15S/n^\beta$ as shown in following lemma, with a proof in Appendix C.5.1.

**Lemma 23.** *Assuming* $\frac{c_2}{c_1}$ $<$ $\frac{8}{(\sqrt{2}+1)^2}$ $-$ $1$ $\approx$ $0.373$, *and let* $\beta$ $=$ $\min\left\{ \frac{c_1}{6}, \frac{(c_1-c_2)^2}{96c_1}, \frac{1}{3}\left(\sqrt{2c_1} - \frac{\sqrt{2}+1}{2}\sqrt{c_1+c_2}\right)^2 \right\}$, *we have*

$$\mathbb{P}(E^c) \leq \frac{15S}{n^\beta} , \qquad (142)$$

*where good event E is defined in Eq.* (134).

For the first term in Eq. (141), we have

$$\mathbb{E}[\mathcal{E}_1^2] = \mathbb{E}[\mathrm{Var}(\mathcal{E}_1 | I_1) + (E[\mathcal{E}_1 | I_1])^2] = \mathbb{E}[\mathrm{Var}(\mathcal{E}_1 | I_1)] \leq \sum_{i \in \{i : e^\varepsilon q_i \leq p_i\}} \frac{p_i + e^{2\varepsilon} q_i}{n} \lesssim \frac{e^\varepsilon}{n} (143)$$

where we use the fact that $\mathbb{E}[\mathcal{E}_1 | I_1] = 0$ with probability one and the fact that $\hat{p}_{i,2}$ and $\hat{q}_{i,2}$ are independent for indices in $I_1$.

The second term in Eq. (141) is bounded by following lemma, with a proof in Appendix C.5.3.

**Lemma 24.** *Suppose* $(p, e^\varepsilon q) \in \left[0, \frac{2c_1 \ln n}{n}\right]^2$, $(n\hat{p}, n\hat{q}) \sim \mathrm{Poi}(np) \times \mathrm{Poi}(nq)$. *Then,*

$$\left| \mathbb{E}\tilde{D}_K^{(1)}(\hat{p}, \hat{q}) - [p - e^\varepsilon q]^+ \right| \lesssim \frac{1}{K}\sqrt{\frac{c_1 \ln n}{n}}(\sqrt{p} + \sqrt{e^\varepsilon q}) + \frac{1}{K^2}\frac{c_1 \ln n}{n} , \qquad (144)$$

*and*

$$\mathrm{Var}\left(\tilde{D}_K^{(1)}(\hat{p}, \hat{q})\right) \lesssim \frac{B^K c_1 c_3^4 \ln^5 n}{n}(p + e^\varepsilon q) , \qquad (145)$$

*for some constant* $B > 0$. *The estimator* $\tilde{D}_K^{(1)}$ *is introduced in Eq.* (13) *and* $K = c_3 \ln n$, $c_3 e^\varepsilon < c_1$.

We have

$$\mathbb{E}[\mathcal{E}_3^2] \qquad (146)$$

$$\lesssim \sum_{i=1}^S \frac{B^K c_1 c_3^4 \ln^5 n}{n}(p_i + e^\varepsilon q_i) + \left( \sum_{i=1}^S \frac{1}{K}\sqrt{\frac{c_1 \ln n}{n}}(\sqrt{p_i} + \sqrt{e^\varepsilon q_i}) + \frac{1}{K^2}\frac{c_1 \ln n}{n} \right)^2 (147)$$

$$\lesssim \frac{c_1 c_3^4 \ln^5 n}{n^{1-c_3 \ln B}}(e^\varepsilon + 1) + \frac{c_1(e^\varepsilon + 1)S}{c_3^2 n \ln n} \vee \left( \frac{c_1 S}{c_3^2 n \ln n} \right)^2 . \qquad (148)$$

The third term in Eq. (141) is bounded by following lemma, with a proof in Appendix C.5.4.

**Lemma 25.** *Suppose* $(p, e^\varepsilon q) \in U(c_1, c_1)$, $p + e^\varepsilon q \geq \frac{c_1 \ln n}{2n}$, $x + e^\varepsilon y \geq \frac{p + e^\varepsilon q}{2}$, $x \in [0, 1]$, $y \in [0, 1]$. *Suppose* $(n\hat{p}, n\hat{q}) \sim \mathrm{Poi}(np) \times \mathrm{Poi}(nq)$. *Then,*

$$\left| \mathbb{E}\tilde{D}_K^{(2)}(\hat{p}, \hat{q}; x, y) - [p - e^\varepsilon q]^+ \right| \lesssim \frac{1}{K}\sqrt{\frac{c_1 \ln n}{n}}(\sqrt{x} + \sqrt{e^\varepsilon y}), \tag{149}$$

*and*

$$\mathrm{Var}\left(\tilde{D}_K^{(2)}(\hat{p}, \hat{q}; x, y)\right) \lesssim \frac{B^K c_1 \ln n}{n}(x + e^\varepsilon y), \tag{150}$$

*for some constant* $B > 0$, *and* $K = c_3 \ln n$, $c_3 e^\varepsilon < c_1$.

We have

$$\mathbb{E}\left[\mathcal{E}_4^2 | \hat{p}_{i,1}, \hat{q}_{i,1} : 1 \leq i \leq S\right] \tag{151}$$

$$\lesssim \sum_{i=1}^S \frac{B^K c_1 \ln n}{n}(\hat{p}_{i,1} + e^\varepsilon \hat{q}_{i,1}) + \left(\sum_{i=1}^S \frac{1}{K}\sqrt{\frac{c_1 \ln n}{n}}(\sqrt{\hat{p}_{i,1}} + \sqrt{e^\varepsilon \hat{q}_{i,1}})\right)^2, \tag{152}$$

where $B$ is the larger constant defined in both Lemma 24 and Lemma 25.

Taking expectation with respect to $\{\hat{p}_{i,1}, \hat{q}_{i,1} : 1 \leq i \leq S\}$, we have

$$\mathbb{E}\left[\mathcal{E}_4^2\right] \tag{153}$$

$$\lesssim \sum_{i=1}^S \frac{c_1 \ln n}{n^{1-c_3 \ln B}} \mathbb{E}(\hat{p}_{i,1} + e^\varepsilon \hat{q}_{i,1}) + \mathbb{E}\left(\sum_{i=1}^S \frac{\sqrt{c_1 (\hat{p}_{i,1} + e^\varepsilon \hat{q}_{i,1})}}{\sqrt{c_3^2 n \ln n}}\right)^2 \tag{154}$$

$$\lesssim \frac{c_1 \ln n}{n^{1-c_3 \ln B}}(e^\varepsilon + 1) + \sum_{i=1}^S \mathbb{E}\frac{c_1 (\hat{p}_{i,1} + e^\varepsilon \hat{q}_{i,1})}{c_3^2 n \ln n} + \tag{155}$$

$$\sum_{1 \leq i,j \leq S, i \neq j} \sqrt{\frac{\mathbb{E}[c_1 (\hat{p}_{i,1} + e^\varepsilon \hat{q}_{i,1})]}{c_3^2 n \ln n}}\sqrt{\frac{\mathbb{E}[c_1 (\hat{p}_{j,1} + e^\varepsilon \hat{q}_{j,1})]}{c_3^2 n \ln n}} \tag{156}$$

$$\lesssim \frac{c_1 \ln n}{n^{1-c_3 \ln B}}(e^\varepsilon + 1) + \frac{c_1}{c_3^2 n \ln n}(e^\varepsilon + 1) + \sum_{1 \leq i,j \leq S}\frac{c_1 (p_i + e^\varepsilon q_i + p_j + e^\varepsilon q_j)}{c_3^2 n \ln n} \tag{157}$$

$$\lesssim \frac{c_1 \ln n}{n^{1-c_3 \ln B}}(e^\varepsilon + 1) + \frac{c_1}{c_3^2 n \ln n}(e^\varepsilon + 1). \tag{158}$$

Combing everything together, we have

$$\mathbb{E}\left[\left(\widehat{d}_{\varepsilon,K,c_1,c_2}(P_n\|Q_n) - d_\varepsilon(P\|Q)\right)^2\right] \tag{159}$$

$$\lesssim \frac{e^\varepsilon}{n} + \frac{(c_1 c_3^4 + c_1)\ln^5 n}{n^{1-c_3 \ln B}}(e^\varepsilon + 1) + \frac{c_1(e^\varepsilon + 1)S}{c_3^2 n \ln n} \vee \left(\frac{c_1 S}{c_3^2 n \ln n}\right)^2 + \frac{S}{n^\beta}. \tag{160}$$

If $\ln n \lesssim \ln S$, as we assume $\frac{c_2}{c_1} < 0.373$, we can take $c_2$ small enough and $c_1, c_3$ large enough to guarantee that $\frac{S}{n^\beta} \lesssim \frac{S}{n \ln n}$, $\frac{\ln^5 n}{n^{1-c_3 \ln B}} \lesssim \frac{S}{n \ln n}$. We have,

$$\mathbb{E}\left[\left(\widehat{d}_{\varepsilon,K,c_1,c_2}(P_n\|Q_n) - d_\varepsilon(P\|Q)\right)^2\right] \lesssim \frac{e^\varepsilon S}{n \ln n}. \tag{161}$$

### C.5.1 Proof of Lemma 23

The following lemma shows that non-smooth region $U(c_1, c_2)$ contains the region $U(p; c_1, c_2)$ defined previously, which will be later used to bound the probability of bad events.

**Lemma 26.** *The two-dimensional set* $U(c_1, c_1)$ *defined in Eq.* (5) *satisfies*

$$\cup_{x = e^\varepsilon y, x, y \in [0,1]} U(e^\varepsilon x; c_1, c_1) \times U(e^{2\varepsilon} y; c_1, c_1) \subset U(c_1, c_1). \tag{162}$$

**1) Analysis of $\mathbb{P}(E_1^c)$:**

It follows from Lemma 26 that

$$
\begin{aligned}
\mathbb{P}(E_1^c) \quad &= \quad \mathbb{P}\left(\bigcup_{i=1}^{S}\left\{e^\varepsilon q_i < p_i, e^\varepsilon \hat{q}_{i,1} - \hat{p}_{i,1} > \sqrt{\frac{(c_1+c_2)\ln n}{n}}(\sqrt{e^\varepsilon \hat{q}_{i,1}} + \sqrt{\hat{p}_{i,1}})\right\}\right) \\[2mm]
&\leq \quad S \max_{i\in[S]} \mathbb{P}\left(e^\varepsilon q_i < p_i, e^\varepsilon \hat{q}_{i,1} - \hat{p}_{i,1} > \sqrt{\frac{(c_1+c_2)\ln n}{n}}(\sqrt{e^\varepsilon \hat{q}_{i,1}} + \sqrt{\hat{p}_{i,1}})\right) \\[2mm]
&\leq \quad S \max_{i\in[S]} \mathbb{P}\left(e^\varepsilon q_i = p_i, e^\varepsilon \hat{q}_{i,1} - \hat{p}_{i,1} > \sqrt{\frac{(c_1+c_2)\ln n}{n}}(\sqrt{e^\varepsilon \hat{q}_{i,1}} + \sqrt{\hat{p}_{i,1}})\right) \\[2mm]
&\leq \quad S \max_{i\in[S]} \mathbb{P}\left(e^\varepsilon q_i = p_i, (\hat{p}_{i,1}, e^\varepsilon \hat{q}_{i,1}) \notin U(c_1,c_2)\right) \\[2mm]
&\leq \quad S \max_{i\in[S]} \mathbb{P}\left(e^\varepsilon q_i = p_i, (\hat{p}_{i,1}, e^\varepsilon \hat{q}_{i,1}) \notin U(e^\varepsilon p_i; \frac{c_1+c_2}{2}, \frac{c_1+c_2}{2}) \times \right. \\[2mm]
&\qquad \left. U(e^{2\varepsilon} q_i; \frac{c_1+c_2}{2}, \frac{c_1+c_2}{2})\right) \\[2mm]
&\leq \quad S \max_{i\in[S]} \left(1 - \mathbb{P}(\hat{p}_{i,1} \in U(e^\varepsilon p_i; \frac{c_1+c_2}{2}, \frac{c_1+c_2}{2})) \times \right. \\[2mm]
&\qquad \left. \mathbb{P}(e^\varepsilon q_i = p_i, e^\varepsilon \hat{q}_{i,1} \in U(e^{2\varepsilon} q_i; \frac{c_1+c_2}{2}, \frac{c_1+c_2}{2}))\right) \\[2mm]
&\leq \quad S\left(1 - \left(1 - \frac{2}{n^{\frac{c_1+c_2}{6}}}\right)^2\right) \leq \frac{4S}{n^{\frac{c_1+c_2}{6}}},
\end{aligned}
$$

where we have applied Lemma 6 in the last inequality.

**2) Analysis of $\mathbb{P}(E_2^c)$:** Similarly, we have

$$
\mathbb{P}(E_2^c) \quad \leq \quad \frac{4S}{n^{\frac{c_1+c_2}{6}}} . \tag{163}
$$

**3) Analysis of $\mathbb{P}(E_3^c)$:**

$$
\begin{aligned}
\mathbb{P}(E_3^c) \quad &= \quad \mathbb{P}\left(\bigcup_{i=1}^{S}\left\{(p_i, e^\varepsilon q_i) \notin \left[0, \frac{2c_1\ln n}{n}\right]^2, \hat{p}_{i,1} + e^\varepsilon \hat{q}_{i,1} < \frac{c_1\ln n}{n}\right\}\right) \\[2mm]
&\leq \quad \mathbb{P}\left(\bigcup_{i=1}^{S}\left\{p_i + e^\varepsilon q_i > \frac{2c_1\ln n}{n}, \hat{p}_{i,1} + e^\varepsilon \hat{q}_{i,1} < \frac{c_1\ln n}{n}\right\}\right) \\[2mm]
&\leq \quad S \max_{i\in[S]} \mathbb{P}\left(p_i + e^\varepsilon q_i > \frac{2c_1\ln n}{n}, \hat{p}_{i,1} + e^\varepsilon \hat{q}_{i,1} < \frac{c_1\ln n}{n}\right) \\[2mm]
&\leq \quad \frac{S}{n^{c_1/4}},
\end{aligned}
$$

where we have applied Lemma 2 in the last inequality.

**4) Analysis of $\mathbb{P}(E_4^c)$:**

$$
\begin{aligned}
\mathbb{P}(E_4^c) \quad &\leq \quad S \max_{i\in[S]} \mathbb{P}\left((p_i, e^\varepsilon q_i) \notin U(c_1,c_1), (\hat{p}_{i,1}, e^\varepsilon \hat{q}_{i,1}) \in U(c_1,c_2)\right) + \\[2mm]
&\qquad S \max_{i\in[S]} \mathbb{P}\left(\hat{p}_{i,1} + e^\varepsilon \hat{q}_{i,1} > \frac{c_1\ln n}{n}, p_i + e^\varepsilon q_i < \frac{c_1\ln n}{2n}\right) + \\[2mm]
&\qquad S \max_{i\in[S]} \mathbb{P}\left(\hat{p}_{i,1} + e^\varepsilon \hat{q}_{i,1} \geq \frac{c_1\ln n}{n}, p_i + e^\varepsilon q_i \geq 2(\hat{p}_{i,1} + e^\varepsilon \hat{q}_{i,1})\right) .
\end{aligned}
$$

Using Lemma 2 again, we have

$$S \max_{i \in [S]} \mathbb{P}\left(\hat{p}_{i,1} + e^\varepsilon \hat{q}_{i,1} > \frac{c_1 \ln n}{n}, p_i + e^\varepsilon q_i < \frac{c_1 \ln n}{2n}\right) \ \leq \ \frac{S}{n^{c_1/6}} \ ,$$

and

$$S \max_{i \in [S]} \mathbb{P}\left(\hat{p}_{i,1} + e^\varepsilon \hat{q}_{i,1} \geq \frac{c_1 \ln n}{n}, p_i + e^\varepsilon q_i \geq 2(\hat{p}_{i,1} + e^\varepsilon \hat{q}_{i,1})\right)$$

$$\leq \quad S \max_{i \in [S]} \mathbb{P}\left(p_i + e^\varepsilon q_i \geq \frac{2c_1 \ln n}{n}, p_i + e^\varepsilon q_i \geq 2(\hat{p}_{i,1} + e^\varepsilon \hat{q}_{i,1})\right)$$

$$\leq \quad \frac{S}{n^{c_1/4}} \ .$$

It suffices to show that for $p, q \in [0,1]$, there exists some constant $c > 0$ such that

$$\left(\bigcup_{(p,e^\varepsilon q) \notin U(c_1,c_1)} U(e^\varepsilon p; c, c) \times U(e^{2\varepsilon} q; c, c)\right) \bigcap U(c_1, c_2) = \emptyset. \tag{164}$$

Indeed, we have

$$S \max_{i \in [S]} \mathbb{P}\left((p_i, e^\varepsilon q_i) \notin U(c_1, c_1), (\hat{p}_{i,1}, e^\varepsilon \hat{q}_{i,1}) \in U(c_1, c_2)\right)$$

$$\leq \quad S \max_{i \in [S]} \mathbb{P}\left((\hat{p}_{i,1}, e^\varepsilon \hat{q}_{i,1}) \notin U(e^\varepsilon p_i; c, c) \times U(e^{2\varepsilon} q_i; c, c)\right)$$

$$\leq \quad S \max_{i \in [S]} \left(1 - \mathbb{P}\left(\hat{p}_{i,1} \in U(e^\varepsilon p_i; c, c)\right) \mathbb{P}\left(e^\varepsilon \hat{q}_{i,1} \in U(e^{2\varepsilon} q_i; c, c)\right)\right)$$

$$\leq \quad S\left(1 - \left(1 - \frac{2}{n^{\frac{c}{3}}}\right)^2\right) \ \leq \ \frac{4S}{n^{c/3}} \ ,$$

where the last inequality follows from Lemma 6.

Now we work out a $c$ that satisfies (164). We prove the case when $\sqrt{p} - \sqrt{e^\varepsilon q} \geq \sqrt{\frac{2c_1 \ln n}{n}}$. The other case can be proved in a similar way. Assume $c < c_1$. In this case $p \geq \frac{2c_1 \ln n}{n}$. We will show that for any point $(x, e^\varepsilon y) \in U(e^\varepsilon p; c, c) \times U(e^{2\varepsilon} q; c, c)$, we have $\sqrt{x} - \sqrt{e^\varepsilon y} \geq \sqrt{\frac{(c_1 + c_2) \ln n}{n}}$.

If $q \leq \frac{ce^{-\varepsilon} \ln n}{n}$, for any $(x, e^\varepsilon y) \in U(e^\varepsilon p; c, c) \times U(e^{2\varepsilon} q; c, c)$, we have

$$\sqrt{x} - \sqrt{e^\varepsilon y} \quad \geq \quad \sqrt{p - \sqrt{\frac{cp \ln n}{n}}} - \sqrt{\frac{2c \ln n}{n}}$$

$$\geq \quad \sqrt{\frac{2c_1 \ln n}{n} - \sqrt{2cc_1}\frac{\ln n}{n}} - \sqrt{\frac{2c \ln n}{n}}$$

$$= \quad \sqrt{\frac{\ln n}{n}}\left(\sqrt{2c_1 - \sqrt{2cc_1}} - \sqrt{2c}\right) \ ,$$

where in the second step, we use the fact that $x - \sqrt{ax}$, $a > 0$ is a monotonically increasing function when $x \geq a/4$ and the fact that $p \geq \frac{2c_1 \ln n}{n}$. Let $c = \frac{(c_1 - c_2)^2}{32c_1}$, we can verify that

$$\sqrt{x} - \sqrt{e^\varepsilon y} \quad \geq \quad \sqrt{\frac{\ln n}{n}}\left(\sqrt{2c_1 - \sqrt{2cc_1}} - \sqrt{2c}\right) \ \geq \ \sqrt{\frac{\ln n}{n}}\sqrt{c_1 + c_2} \ . \tag{165}$$

If $q > \frac{ce^{-\varepsilon} \ln n}{n}$, for any $(x, e^\varepsilon y) \in U(e^\varepsilon p; c, c) \times U(e^{2\varepsilon} q; c, c)$, we have

$$\sqrt{x} - \sqrt{e^\varepsilon y} \;\geq\; \sqrt{p - \sqrt{\frac{cp\ln n}{n}}} - \sqrt{e^\varepsilon q + \sqrt{\frac{ce^\varepsilon q \ln n}{n}}}$$

$$= \;\frac{p - \sqrt{\frac{cp\ln n}{n}} - e^\varepsilon q - \sqrt{\frac{ce^\varepsilon q \ln n}{n}}}{\sqrt{p - \sqrt{\frac{cp\ln n}{n}}} + \sqrt{e^\varepsilon q + \sqrt{\frac{ce^\varepsilon q \ln n}{n}}}}$$

$$= \;\frac{(\sqrt{p} - \sqrt{e^\varepsilon q})(\sqrt{p} + \sqrt{e^\varepsilon q}) - \sqrt{\frac{c\ln n}{n}}(\sqrt{e^\varepsilon q} + \sqrt{p})}{\sqrt{p - \sqrt{\frac{cp\ln n}{n}}} + \sqrt{e^\varepsilon q + \sqrt{\frac{ce^\varepsilon q \ln n}{n}}}}$$

$$\geq \;(\sqrt{2c_1} - \sqrt{c})\sqrt{\frac{\ln n}{n}} \frac{\sqrt{p} + \sqrt{e^\varepsilon q}}{\sqrt{p - \sqrt{\frac{cp\ln n}{n}}} + \sqrt{e^\varepsilon q + \sqrt{\frac{ce^\varepsilon q \ln n}{n}}}} \;.$$

Further, since $e^\varepsilon q > \frac{c\ln n}{n}$,

$$\frac{\sqrt{p} + \sqrt{e^\varepsilon q}}{\sqrt{p - \sqrt{\frac{cp\ln n}{n}}} + \sqrt{e^\varepsilon q + \sqrt{\frac{ce^\varepsilon q \ln n}{n}}}} \;\geq\; \frac{\sqrt{p} + \sqrt{e^\varepsilon q}}{\sqrt{p} + \sqrt{2e^\varepsilon q}}$$

$$\geq \;\frac{\sqrt{e^\varepsilon q} + \sqrt{\frac{2c_1\ln n}{n}} + \sqrt{e^\varepsilon q}}{\sqrt{2e^\varepsilon q} + \sqrt{e^\varepsilon q} + \sqrt{\frac{2c_1\ln n}{n}}}$$

$$\geq \;\frac{2}{\sqrt{2} + 1}\;,$$

where in the second inequality, we used the fact that $\frac{x + \sqrt{e^\varepsilon q}}{x + \sqrt{2e^\varepsilon q}}$ is a monotonically increasing function of $x$ when $x \geq 0$, and in the third inequality, we used the fact that $\frac{2x+a}{(\sqrt{2}+1)x+a}$ is a monotonically decreasing function of $x$ when $a > 0$, $x > 0$. To guarantee that $\sqrt{x} - \sqrt{e^\varepsilon y} \geq \sqrt{\frac{(c_1+c_2)\ln n}{n}}$, we need

$$\frac{2}{\sqrt{2}+1}\left(\sqrt{2c_1} - \sqrt{c}\right) \;\geq\; \sqrt{c_1 + c_2}\;, \tag{166}$$

which is equivalent to

$$c \leq \left(\sqrt{2c_1} - \frac{\sqrt{2}+1}{2}\sqrt{c_1 + c_2}\right)^2\;, \tag{167}$$

with the constraint that $\frac{c_2}{c_1} < \frac{8}{(\sqrt{2}+1)^2} - 1 \approx 0.373$.

### C.5.2    Proof of Lemma 26

If $x \leq \frac{c_1 e^\varepsilon \ln n}{n}$ and thus $y \leq \frac{c_1\ln n}{n}$, it suffices to show $[0, \frac{2c_1\ln n}{n}]^2 \subset U(c_1, c_1)$. For $(u, e^\varepsilon v) \in [0, \frac{2c_1\ln n}{n}]^2$, we have

$$|\sqrt{u} - \sqrt{e^\varepsilon v}| \;\leq\; \sqrt{\frac{2c_1\ln n}{n}}\;. \tag{168}$$

For $x > \frac{c_1 e^\varepsilon \ln n}{n}$ and thus $y > \frac{c_1\ln n}{n}$, it suffices to show $\left[x - \sqrt{\frac{c_1 x\ln n}{n}}, x + \sqrt{\frac{c_1 x\ln n}{n}}\right]^2 \subset U(c_1, c_1)$. It is shown in [21, Lemma 3] that for any $(u, e^\varepsilon v) \in \left[y - \sqrt{\frac{c_1 y\ln n}{n}}, y + \sqrt{\frac{c_1 y\ln n}{n}}\right]^2$, we have

$$|\sqrt{u} - \sqrt{e^\varepsilon v}| \;\leq\; \sqrt{\frac{2c_1\ln n}{n}}\;. \tag{169}$$

### C.5.3 Proof of Lemma 24

We first analyze the bias. Let $\Delta = \frac{c_1 \ln n}{n}$. As we have applied unbiased estimator $\tilde{D}_K^{(1)}(x, y)$ of $D_K^{(1)}(x, y)$, the bias is entirely due to the functional approximation. We show that for $(x, y) \in [0, 1]^2$, $|u_K(x, y)v_K(x, y) - [x - y]^+| \lesssim \left( \frac{\sqrt{x} + \sqrt{y}}{K} + \frac{1}{K^2} \right)$. Indeed, we have

$$
\begin{aligned}
& \left| u_K(x, y)v_K(x, y) - [x - y]^+ \right| \\
= \quad & \left| u_K(x, y)v_K(x, y) - u_K(x, y)[\sqrt{x} - \sqrt{y}]^+ + u_K(x, y)[\sqrt{x} - \sqrt{y}]^+ - [x - y]^+ \right| \\
\leq \quad & |u_K(x, y)||v_K(x, y) - [\sqrt{x} - \sqrt{y}]^+| + [\sqrt{x} - \sqrt{y}]^+ |u_K(x, y) - \sqrt{x} - \sqrt{y}| \\
\leq \quad & |u_K(x, y) - \sqrt{x} - \sqrt{y}||v_K(x, y) - [\sqrt{x} - \sqrt{y}]^+| + \\
& |\sqrt{x} + \sqrt{y}||v_K(x, y) - [\sqrt{x} - \sqrt{y}]^+| + [\sqrt{x} - \sqrt{y}]^+ |u_K(x, y) - \sqrt{x} - \sqrt{y}| .
\end{aligned}
$$

It follows from Lemma 7 and Lemma 9 that

$$
\Delta_K \left[ \sqrt{x}; [0, 1] \right] \lesssim \frac{1}{K} , \tag{170}
$$

which implies

$$
|u_K(x, y) - \sqrt{x} - \sqrt{y}| \lesssim \frac{1}{K} . \tag{171}
$$

It follows from Lemma 7, Lemma 9 and the fact that $[\sqrt{b} - \sqrt{a}]^+ \leq [\sqrt{b} - \sqrt{c}]^+ + [\sqrt{c} - \sqrt{a}]^+$, we have

$$
|v_K(x, y) - [\sqrt{x} - \sqrt{y}]^+| \lesssim \frac{1}{K} . \tag{172}
$$

Together with Eq. (171), we have

$$
\left| u_K(x, y)v_K(x, y) - [x - y]^+ \right| \lesssim \frac{1}{K^2} + \frac{\sqrt{x} + \sqrt{y} + [\sqrt{x} - \sqrt{y}]^+}{K} \tag{173}
$$

$$
\lesssim \frac{1}{K^2} + \frac{\sqrt{x} + \sqrt{y}}{K} , \tag{174}
$$

which implies there exists a constant $M > 0$ such that

$$
\left| u_K(x, y)v_K(x, y) - u_K(0, 0)v_K(0, 0) - [x - y]^+ \right| \leq M \left( \frac{1}{K^2} + \frac{\sqrt{x} + \sqrt{y}}{K} \right) . \tag{175}
$$

Let $x = p/(2\Delta)$ and $y = e^\varepsilon q/(2\Delta)$. We have

$$
\begin{aligned}
& \sup_{(p, e^\varepsilon q) \in [0, 2\Delta]^2} \left| \mathbb{E}\tilde{D}_K^{(1)}(\hat{p}, \hat{q}) - [p - e^\varepsilon q]^+ \right| \\
= \quad & \sup_{(p, e^\varepsilon q) \in [0, 2\Delta]^2} \left| D_K^{(1)}(p, q) - [p - e^\varepsilon q]^+ \right| \\
= \quad & \sup_{(p, e^\varepsilon q) \in [0, 2\Delta]^2} 2\Delta \left| h_{2K}(\frac{p}{2\Delta}, \frac{e^\varepsilon q}{2\Delta}) - [\frac{p}{2\Delta} - \frac{e^\varepsilon q}{2\Delta}]^+ \right| \\
= \quad & \sup_{(x, y) \in [0, 1]^2} 2\Delta \left| h_{2K}(x, y) - [x - y]^+ \right| \\
= \quad & \sup_{(x, y) \in [0, 1]^2} 2\Delta \left| u_K(x, y)v_K(x, y) - u_K(0, 0)v_K(0, 0) - [x - y]^+ \right| \\
\leq \quad & 2\Delta M \left( \frac{1}{K^2} + \frac{\sqrt{x} + \sqrt{y}}{K} \right) \\
\lesssim \quad & \frac{1}{K} \sqrt{\frac{c_1 \ln n}{n}} (\sqrt{p} + \sqrt{e^\varepsilon q}) + \frac{1}{K^2} \frac{c_1 \ln n}{n} .
\end{aligned}
$$

We now analyze the variance. Express the polynomial $h_{2K}(x, y)$ explicitly as

$$h_{2K}(x, y) \quad = \quad \sum_{0 \leq i \leq 2K, 0 \leq j \leq 2K, i+j \geq 1} h_{ij} x^i y^j \tag{176}$$

$$= \quad \sum_{0 \leq i \leq 2K} \left( \sum_{0 \leq j \leq 2K, i+j \geq 1} h_{ij} y^j \right) x^i . \tag{177}$$

For any fixed value of $y$, $h_{2K}(x^2, y^2)$ is a polynomial of $x$ with degree no more than $4K$ that is uniformly bounded by a universal constant on $[0, 1]$. It follows from Lemma 10 that for any fixed $y \in [-1, 1]$,

$$\left| \sum_{0 \leq j \leq 2K} h_{ij} y^{2j} \right| \quad \leq \quad M(\sqrt{2} + 1)^{4K} , \tag{178}$$

which together with Lemma 10, implies that

$$|h_{ij}| \quad \leq \quad M(\sqrt{2} + 1)^{8K} . \tag{179}$$

Since $\tilde{D}_K^{(1)}$ is the unbiased estimator of $2\Delta h_{2K}(\frac{p}{2\Delta}, \frac{e^\varepsilon q}{2\Delta})$, we know

$$\tilde{D}_K^{(1)}(\hat{p}, \hat{q}) \quad = \quad \sum_{0 \leq i,j \leq 2K, i+j \geq 1} h_{ij} (2\Delta)^{1-i-j} e^{j\varepsilon} g_{i,0}(\hat{p}) g_{j,0}(\hat{q}) , \tag{180}$$

where $g_{i,0}(\hat{p}) = \prod_{k=0}^{i-1}(\hat{p} - \frac{k}{m})$ introduced by Lemma 11.

Denote $\|X\|_2 = \sqrt{\mathbb{E}(X - \mathbb{E}X)^2}$ for random variable $X$, and $M_1 = 2K \vee 2n\Delta$, $M_2 = 2K \vee 2ne^{-\varepsilon}\Delta$. Using triangle inequality of the norm $\|\cdot\|_2$ and Lemma 11, we know

$$\|\tilde{D}_K^{(1)}(\hat{p}, \hat{q})\|_2 \quad \leq \quad \sum_{0 \leq i,j \leq 2K, i+j \geq 1} |h_{ij}| (2\Delta)^{1-i-j} e^{j\varepsilon} \|g_{i,0}(\hat{p})\|_2 \|g_{j,0}(\hat{q})\|_2$$

$$\leq \quad \sum_{0 \leq i,j \leq 2K, i+j \geq 1} M(\sqrt{2}+1)^{8K} (2\Delta) \left( \frac{1}{2\Delta} \sqrt{\frac{2M_1 p}{n}} \right)^i \left( \frac{e^\varepsilon}{2\Delta} \sqrt{\frac{2M_2 q}{n}} \right)^j$$

$$\leq \quad (\sqrt{2}+1)^{8K} \frac{c_1 \ln n}{n} \sum_{0 \leq i,j \leq 2K, i+j \geq 1} \left( \sqrt{\frac{p}{2\Delta}} \right)^i \left( \sqrt{\frac{e^\varepsilon q}{2\Delta}} \right)^j .$$

Since for any $x \in [0, 1]$. $y \in [0, 1]$,

$$\left| \sum_{0 \leq i,j \leq 2K, i+j \geq 1} x^i y^j \right| \quad \leq \quad \left| \sum_{j=1}^{2K} y^j \right| + \left| \sum_{i=1}^{2K} x^i \right| + xy \left| \sum_{0 \leq i,j \leq 2K-1} x^i y^j \right|$$

$$\leq \quad y(2K) + x(2K) + xy(2K)^2$$

$$\leq \quad 2(2K)^2 (x + y) ,$$

we know

$$\|\tilde{D}_K^{(1)}(\hat{p}, \hat{q})\|_2 \quad \lesssim \quad (\sqrt{2}+1)^{8K} \frac{c_1 K^2 \ln n}{n} \left( \sqrt{\frac{p}{2\Delta}} + \sqrt{\frac{e^\varepsilon q}{2\Delta}} \right) \tag{181}$$

$$\lesssim \quad \sqrt{B^K \frac{c_1 c_3^5 \ln^5 n}{n} (p + e^\varepsilon q)} , \tag{182}$$

for some constant $B > 0$.

### C.5.4 Proof of Lemma 25

We first analyze the bias. As we apply the unbiased estimator $\tilde{D}_K^{(2)}(\hat{p}, \hat{q}; x, y)$ of $D_K^{(2)}(p, q; x, y)$. The bias is entirely due to the functional approximation error. Namely,

$$\mathbb{E}\left[\tilde{D}_K^{(2)}(\hat{p}, \hat{q}; x, y)|x, y\right] \quad = \quad D_K^{(2)}(p, q; x, y) \tag{183}$$

$$= \quad \frac{1}{2}\sum_{j=0}^{K} r_j W^{-j+1}(e^\varepsilon q - p)^j + \frac{p - e^\varepsilon q}{2}, \tag{184}$$

where $W = \sqrt{\frac{8c_1 \ln n}{n}}(\sqrt{e^\varepsilon x + y})$, and $r_j$ is defined as the coefficient of best polynomial approximation $R_K(t)$ of $|t|$ over $[-1, 1]$ with order $K$: $R_K(t) = \sum_{j=0}^{K} r_j t^j$.

Since $(p, e^\varepsilon q) \in U(c_1, c_1)$, we know

$$|p - e^\varepsilon q| \quad \leq \quad \sqrt{\frac{2c_1 \ln n}{n}}\left(\sqrt{p} + \sqrt{e^\varepsilon q}\right)$$

$$\leq \quad \sqrt{\frac{2c_1 \ln n}{n}}\sqrt{2}\left(\sqrt{p + e^\varepsilon q}\right)$$

$$\leq \quad \sqrt{\frac{2c_1 \ln n}{n}}\sqrt{2}\left(\sqrt{2(x + e^\varepsilon y)}\right)$$

$$\leq \quad W$$

where we have used the fact that $\sqrt{p} + \sqrt{e^\varepsilon q} \leq \sqrt{2(p + e^\varepsilon q)}$ and the assumption that $p + e^\varepsilon q \leq 2(x + e^\varepsilon y)$.

It is known in [48, Chapter 9, Theorem 3.3] that

$$|R_k(t) - |t|| \quad \lesssim \quad \frac{1}{K}, \tag{185}$$

for all $t \in [-1, 1]$.

We show $D_K^{(2)}(p, q; x, y)$ is best polynomial approximation of $[p - e^\varepsilon q]^+$. We have

$$\left|D_K^{(2)}(p, q; x, y) - [p - e^\varepsilon q]^+\right| \quad = \quad \frac{1}{2}\left|\sum_{j=0}^{K} r_j W^{-j+1}(e^\varepsilon q - p)^j - |p - e^\varepsilon q|\right|$$

$$= \quad \frac{W}{2}\left|R_K\left(\frac{e^\varepsilon q - p}{W}\right) - \left|\frac{e^\varepsilon q - p}{W}\right|\right|$$

$$\lesssim \quad \frac{W}{K}$$

$$\lesssim \quad \frac{1}{K}\sqrt{\frac{c_1 \ln n}{n}}(\sqrt{x} + \sqrt{e^\varepsilon y}).$$

Now we analyze the variance.

Let $a_j = r_j$ for $j = 0, 2, 3, \ldots, K$ and $a_1 = r_1 - 1$ we can write $D_K^{(2)}(p, q; x, y)$ as

$$D_K^{(2)}(p, q; x, y) \quad = \quad \frac{1}{2}\sum_{j=0}^{K} a_j W^{-j+1}(e^\varepsilon q - p)^j. \tag{186}$$

It was shown in [49, Lemma 2] that $r_j \leq 2^{3K}$, $0 \leq j \leq K$. So we have $|a_j| \leq 2 \cdot 2^{3K}$. Denote the unique uniformly minimum unbiased estimator (MVUE) of $(e^\varepsilon q - p)^j$ by $\hat{A}_j(\hat{p}, \hat{q})$. Then the unbiased estimator $\tilde{D}_K^{(2)}$ of polynomial function $D_K^{(2)}$ is

$$\tilde{D}_K^{(2)}(\hat{p}, \hat{q}; x, y) \quad = \quad \frac{1}{2}\sum_{j=0}^{K} a_j W^{-j+1}\hat{A}_j(\hat{p}, \hat{q}). \tag{187}$$

Denote $\|X\|_2 = \sqrt{\mathbb{E}(X - \mathbb{E}X)^2}$ for random variable $X$. It follows from triangle inequality of $\|\cdot\|_2$ and Lemma 12 that

$$
\|\tilde{D}_K^{(2)}(\hat{p},\hat{q};x,y)\|_2 \quad \leq \quad \frac{1}{2}\sum_{j=1}^{K}|a_j|W^{-j+1}\|\hat{A}_j\|_2 \tag{188}
$$

$$
\leq \quad 2^{3K}W\sum_{j=1}^{K}\left(\frac{\sqrt{2}|e^\varepsilon q - p|}{W} \vee \frac{\sqrt{8j(e^{2\varepsilon}q \vee p)}}{\sqrt{n}W}\right)^j, \tag{189}
$$

where

$$
\frac{\sqrt{2}|e^\varepsilon q - p|}{W} \vee \frac{\sqrt{8j(e^{2\varepsilon}q \vee p)}}{\sqrt{n}W}
$$

$$
\leq \quad \frac{\sqrt{2}\sqrt{\frac{2c_1\ln n}{n}}(\sqrt{e^\varepsilon q} + \sqrt{p})}{\sqrt{\frac{8c_1\ln n}{n}}\sqrt{e^\varepsilon y + x}} \vee \frac{\sqrt{8K(e^{2\varepsilon}q + p)}}{\sqrt{n}\sqrt{\frac{8c_1\ln n}{n}}\sqrt{e^\varepsilon y + x}}
$$

$$
\leq \quad \frac{\sqrt{e^\varepsilon q} + \sqrt{p}}{\sqrt{2}\sqrt{e^\varepsilon y + x}} \vee \sqrt{\frac{c_3}{c_1}}\frac{\sqrt{e^{2\varepsilon}q + p}}{\sqrt{e^\varepsilon y + x}}
$$

$$
\leq \quad \sqrt{2} \vee \sqrt{\frac{2c_3 e^\varepsilon}{c_1}}
$$

$$
\leq \quad \sqrt{2}\,.
$$

Consequently,

$$
\|\tilde{D}_K^{(2)}(\hat{p},\hat{q};x,y)\|_2 \quad \leq \quad 2^{3K}WK(\sqrt{2})^K
$$

$$
\leq \quad 2^{3K}\sqrt{\frac{8c_1\ln n}{n}}\sqrt{x + e^\varepsilon y}K(\sqrt{2})^K
$$

$$
\leq \quad \sqrt{B^K\frac{(x + e^\varepsilon y)c_1\ln n}{n}}\,,
$$

where $B > 0$ is some constant.