[Reviews · NeurIPS 2019]

Reviewer 1



I have read the rebuttal. The results are technically interesting, in particular I found the result on sample size amplification from 1/n to 1/(n ln n) pleasantly surprising. However, the paper does not provide an end-to-end solution to the problem of detecting DP violation. --------------- The paper investigates the trade-off between accuracy and sample size in estimating differential privacy guarantees from a black-box access to a purportedly private mechanism. Let A be a mechanism and D and D’ be two neighboring datasets. Let P denote the output distribution A(D) and Q denote the output distribution A(D’). Now checking whether A is (eps,delta)-differentially private can be reduced to a estimating an appropriately defined divergence d_eps(P||Q). Let A be a discrete valued mechanism A whose range is over an alphabet set size S. The first result in this paper is to show that with a simple estimator it is necessary and sufficient for the sample size n to grow as e^eps S to achieve an arbitrary desired error rate (this result marginally improves an earlier upper bound of [12]). The main result of this paper is to show a minimax optimal estimator whose error rate is e^eps S/(n ln n), thus improving the dependence on the sample size from 1/n to 1/(n ln n). The results relies on a line of recent work on property estimation where the general recipe is to identify the regime where the property to be estimated is not smooth, and use functional approximation to estimate a smoothed version of the property. The phenomena of effective sample size amplification is technically interesting. My main concern is the results are narrow in their significance. In a differential privacy verification application the fact that you have to fix the neighboring dataset D’ is very restrictive, and therefore I do not see any practical consequence of this result for differential privacy (as suggested by the title and introduction).

Reviewer 2



Originality: The techniques seem to be build upon polynomial-approximation based techniques provided in [25]. If that is the case, the contribution of that work needs to be better acknowledged. In the current version, it's only introduced among several other papers on entropy estimation. Quality and Clarity: The results are clean and well presented. The reviewer enjoyed reading this work. Significance: The work is very interesting to the reviewer. However, I'm not sure about its significance, since I'm not an expert in this area. ----------------------------------------------------------------------------------------------------- Post Rebuttal: The authors have provided detailed responses and addressed my comments. The reviewer recommends acceptance.

Reviewer 3



The authors study the rates of estimating approximate differential privacy (aDP). They do so by reformulating it as a property estimation problem. I find this reduction fairly novel and ties DP to a large body of work on polynomial estimation. In property estimation, it is known that carefully trading off the bias and variance via polynomial approximation, particularly in regions of low probability, allows for obtaining the optimal min max rates. The authors follow the same recipe and show that the min max error scales as Se^\epsilon / n \log n. This result is cute, and the \log n factor is particularly interesting. This is in contrast to previous work where the only guarantees provided are empirical in nature. The paper is also overall well-written. I liked the fact that authors first prove the result where P is known, and then proceed to estimating both P and Q. I am also satisfied with their simulation and code submission. I recommend that the paper be accepted, I have some minor concerns that I will point out below. First, some typos - 89: ... improved ... 182: The method ... corresponding ... detail in ... 211: ... is non smooth .., In the proof of Theorem 1, it is not clear to me why Poisssonisation is required. Please elaborate on this. While I realize that a naive analysis of the empirical estimate will lead to a factor of e^{2\epsilon}S/n, an elaboration on how Poissonisation allows one to overcome this exponent of 2 might be helpful. On the other hand if it is simply a case of an easier analysis, it might be instructive to mention the same. Some intuition on why the degree of the polynomial K, must scale as \log n might be helpful. This may be presented near Theorem 2 or Theorem 4. Under case 1 and 2 under 2.2.1, it might help to write a line or two why the particular ranges of (x,e^epislon y) are considered. In its present state these seem to appear out of the blue. Post Rebuttal: The authors have addressed my concerns. If the paper is still on the borderline I would definitely recommend an accept.

[Author Response · NeurIPS 2019]

We thank all the reviewers for their constructive feedback and address each one below.

**Response to Reviewer 1:** We emphasize the **novelty** and the **motivation** of our paper below.

**1. Re: novelty of our technique:** The most relevant (and state-of-the-art) previous work on detecting violation of DP
is [11]. We improve over [11] as follows. $(i)$ The method proposed in [11] involves heuristically enumerating over a
large number of subsets $E \subseteq [S]$ that is extremely inefficient, whereas our estimator is efficient with complexity linear
in $|S|$. $(ii)$ Only we have a theoretical guarantee, and further achieve statistically optimality. $(iii)$ We can estimate
more general $(\varepsilon, \delta)$-DP, whereas [11] can only check a special case of $(\varepsilon, 0)$-DP.

**2. Re: motivating application:** One major motivation of this paper is to **detect violation of DP**, whose importance is
acknowledged by the best paper award given to [11] in 2018 ACM CCS conference, which study the same problem. We
significantly improve the detection, by proposing a principled method as we discussed in the above paragraph. There
are several points of failure to designing/implementing DP mechanisms, and a number of published algorithms are
incorrect. In this paper, we propose a new approach to finding bugs that cause algorithms to violate differential privacy,
and generating counterexamples that illustrate these violations. Such a counterexample generator would be useful
in the development cycle in detecting errors and fixing them. This does not necessarily require checking all pairs of
neighboring databases, which is infeasible.

Regarding running our estimator on one or a small number of paired neighboring databases, we would like to emphasize
the following points. $(i)$ If you have some side information, then this might significantly reduce the search space. For
example, if your mechanism is noise adding, then you only need to check two data bases whose true query output is
at maximum difference, i.e. the sensitivity. Heuristics on choosing those databases to check have been proposed, for
example, in [11], and have been proven effective on real-world mechanisms (which we also demonstrate in Figure 1).
Such data driven methods for checking DP guarantees were successfully used in reverse engineering the privacy loss in
Apple's DP mechanisms in [ "Privacy loss in Apple's implementation of differential privacy on MacOS 10.12," , J. Tang,
A. Korolova, X. Bai, X. Wang, and X. Wang, 2017]. $(ii)$ [12] showed that with a relaxed definition of approximate
differential privacy called "random approximate differential privacy", we only need to test on randomly selected pairs
of databases (non-adaptively) to guarantee privacy. Our estimator can be readily applied to such a scenario.

**Response to Reviewer 2:** There is a long line of research on estimating functionals of a single discrete distribution,
which use similar techniques summarized in [25] for generic functionals. As we build upon similar polynomial
approximation techniques, we will better acknowledge [25] in the final revision. However, we want to emphasize that
our work diverges from [25] in the sense that we care about a divergence between two distributions, which requires a
more careful design of the polynomial approximation. In that sense, our work is more closely related to [31], which we
will better acknowledge in the revision as well.

**Response to Reviewer 3:** We will fix the typos as suggested, and discuss major comments below.

(Re: Poissonization) The choice of Poissonization makes the analysis relatively simpler, and we will state this explicitly.
Without Poissonization, the marginal distribution will change from Poisson to Binomial. The minimum variance
unbiased estimator should be changed accordingly (for example see ["Bias reduction by taylor series", C.S.Withers,
1987]). With this modified approximation, we believe that the same guarantee might be achievable, but requires more
careful analysis on the covariance, as we do not have independence. Getting more samples on symbol $i$ implies getting
less samples of other symbols, if we fix the sample size $n$ (as opposed to choosing it from Poisson).

(Re: degree $K$) We will add the explanation that "Bias scales as $(1/K)\sqrt{(p_i \ln n)/n}$ and variance scales as
$(B^K p_i \ln n)/n$, and the optimal trade-off is achieved for $K = c \ln n$ with an appropriate choice of the constant.".

(Re: experiments) We will add more results comparing the plug-in and proposed estimators. For example, the following
show results for a different value of $\varepsilon = 0.2$ (left) and different distributions of Zipf and mixture of uniform (right).

MSE

sample size $n$

sample size $n$

(Re: algorithm 2) We will move some proofs to the appendix and expand the explanation of the Algorithm 2. We will
also explain that the division into case 1 and 2 in the range of $(x, e^\varepsilon y)$ is necessary. Case 2 is the standard approximation,
but when $(p, q) \in [0, 2c_1 \ln n/n]^2$ this approximation fails to provide the desired bias, as in Eq. (149). This is because
as both $p$ and $q$ get small, the desired level of bias also gets small, and the standard approximation is no longer sufficient.

[Meta-Review · NeurIPS 2019]

The reviews are broadly positive, but the authors should take the following into account for the camera-ready version: - Change the title as the paper does not currently deliver what the title promises. In particular, the end-to-end problem of detecting DP-violation is not completely solved by this paper. - Further emphasize the relation to previous work. The authors should of course also follow other reviewer comments to improve the write-up of the paper.